# ALIGN YOUR STRUCTURES: GENERATING TRAJECTORIES WITH STRUCTURE PRETRAINING FOR MOLECULAR DYNAMICS

**Aniketh Iyengar**[1*] **Jiaqi Han**[1*] **Pengwei Sun**[1*] **Mingjian Jiang**[1] **Jianwen Xie**[2] **Stefano Ermon**[1]
[1] Stanford University    [2] Lambda, Inc

## ABSTRACT

Generating molecular dynamics (MD) trajectories using deep generative models has attracted increasing attention, yet remains inherently challenging due to the limited availability of MD data and the complexities involved in modeling high-dimensional MD distributions. To overcome these challenges, we propose a novel framework that leverages structure pretraining for MD trajectory generation. Specifically, we first train a diffusion-based structure generation model on a large-scale conformer dataset, on top of which we introduce an interpolator module trained on MD trajectory data, designed to enforce temporal consistency among generated structures. Our approach effectively harnesses abundant structural data to mitigate the scarcity of MD trajectory data and effectively decomposes the intricate MD modeling task into two manageable subproblems: structural generation and temporal alignment. We comprehensively evaluate our method on the QM9 and DRUGS small-molecule datasets across unconditional generation, forward simulation, and interpolation tasks, and further extend our framework and analysis to tetrapeptide and protein monomer systems. Experimental results confirm that our approach excels in generating chemically realistic MD trajectories, as evidenced by remarkable improvements of accuracy in geometric, dynamical, and energetic measurements.

## 1 INTRODUCTION

Molecular Dynamics (MD) is a computational method used to model the physical motions of atoms and molecules over time (Alder & Wainwright, 1959; Verlet, 1967). Numerically integrating Newton's equations of motion, MD simulates the temporal evolution of molecular systems at atomic resolution. It has become a widely adopted tool in biology (McCammon et al., 1977), chemistry (Rahman, 1964), and materials science (Antalík et al., 2024). However, MD can be computationally demanding, often requiring long simulation times and many small integration steps, especially for physio-realistic dynamics. This cost has motivated extensive work on accelerating MD and improving sampling efficiency (Shaw et al., 2009; Darden et al., 1993; Laio & Parrinello, 2002). Moreover, advances in biomolecular engineering increasingly leverage machine learning to design molecular systems (Jumper et al., 2021; Passaro et al., 2025; Powers et al., 2025), highlighting its importance in drug discovery. In this context, deep generative models—especially diffusion models (Noé et al., 2019; Jing et al., 2024a; Klein et al., 2023)—have emerged as effective surrogates for capturing the complex and diverse distributions observed in MD simulations.

Despite their promise, we identify a factor that poses remarkable limitations on their utility. The MD generative models are typically optimized on a single or limited number of molecular systems (Noé et al., 2019; Han et al., 2024; Jing et al., 2024c), making it a fundamental challenge for them to generalize across arbitrary molecules. Two main factors contribute to this issue. *Data scarcity*: Constructing large-scale, physio-realistic MD datasets spanning diverse molecular systems is prohibitively expensive due to the high computational cost of running MD simulations at scale. As a result, available training data is insufficient for capturing the full diversity of MD distributions.

---

*Equal contribution. Correspondence to `jiaqihan@stanford.edu`. Code is available at `https://github.com/ani11452/Align_Your_Structures`.

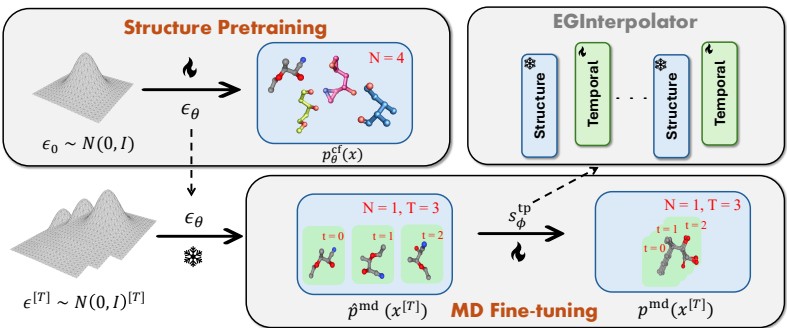

Figure 1: The overall two-stage framework of EGINTERPOLATOR. *Structure pretraining:* We first pretrain a conformer model $\epsilon_\theta$ on a large-scale conformer dataset. *MD fine-tuning:* The model is then combined with additional temporal interpolator $\mathbf{s}_\phi^{\text{tp}}$ to approach the MD distribution $p^{\text{md}}(\mathbf{x}^{[T]})$.

*Modeling complexity*: MD data extends the molecular structure space with an additional temporal dimension, making it inherently high-dimensional. This significantly increases modeling difficulty, especially when models must preserve both structural fidelity and realistic dynamical behavior.

In this work, we propose a novel approach named EGINTERPOLATOR that addresses the challenges through *structure pretraining*. Specifically, we decompose the MD modeling problem into two sequential subtasks. First, we train a conformer diffusion model to generate conformers—*i.e.*, plausible molecular structures corresponding to frames along an MD trajectory—using large-scale conformer datasets. Building on this pretrained structure model, we then initialize additional temporal layers and integrate structural and temporal information through a novel module called the equivariant temporal interpolator. We theoretically show that the temporal interpolator implicitly models a transition from a temporally independent structural distribution to the fully correlated MD distribution. This formulation alleviates optimization difficulty by decoupling spatial and temporal learning, which enables (1) more efficient learning of dynamics from limited MD data through the temporal interpolator, and (2) generation of higher-fidelity, physically realistic molecular poses implicitly constrained by the pretrained structure module.

Our approach directly addresses three central challenges. First, it mitigates MD data scarcity by leveraging large-scale conformer datasets with diverse molecular structures, complementing small-scale MD data and improving generalization to unseen molecules. Second, it ensures structural and energetic fidelity by grounding trajectory generation in a pretrained conformer model, which provides a foundation for downstream dynamics. Third, the two-stage pipeline decomposes the complexity of modeling high-dimensional MD distributions into two manageable tasks: learning the distribution of independent frames and subsequently capturing their temporal dependencies.

**Contributions. 1.** We identify key challenges in the generalization of MD diffusion models and propose structure pretraining as a remedy. **2.** We develop a principled training framework based on structure pretraining and validate it on small molecular systems. **3.** We introduce the equivariant temporal interpolator, a module for learning temporal dependencies across frames. **4.** We evaluate our framework on unconditional generation, forward simulation, and interpolation, showing accurate modeling of MD distributions while preserving conformer generation quality across small molecules, tetrapeptide, and protein monomer systems.

## 2 RELATED WORK

**Geometric diffusion models.** Generative models for geometric data have garnered increasing attention across multiple domains. In molecular generation, GeoDiff (Xu et al., 2022) pioneered for conformer generation while EDM (Hoogeboom et al., 2022b) operates on both continuous coordinates and categorical atom types. Subsequent works (Xu et al., 2023; 2024a) introduced structured latent spaces to enhance scalability and controllability. For larger molecules, GCDM (Morehead & Cheng, 2024) incorporated geometry-complete local frames and chirality-sensitive features into SE(3)-equivariant networks. EBD (Park & Shen, 2024) performs hierarchically by first sampling scaffolds before refining atom positions through blurring-based denoising. Yet, they only model static structures while in this work we study the problem of their temporal correlation in MD.

**Molecular Structure Datasets & Sampling.** Large-scale structural datasets are central to molecular modeling. Some, like the Protein Data Bank (PDB) (Berman et al., 2000), archive experimentally resolved biomolecular structures, while others, such as GEOM (QM9 and Drugs) (Axelrod & Gomez-Bombarelli, 2022) and OMol (Levine et al., 2025), provide computationally derived conformer ensembles at scale. The latter can utilize accelerated sampling strategies that emphasize structural diversity while reducing computational cost. For instance, OMol reports many protein–ligand simulations at elevated temperatures, while GEOM employs CREST (Berman et al., 2000), coupling the semiempirical GFN2-xTB method (Bannwarth et al., 2019) with metadynamics and geometry optimization. Such approaches broaden structural coverage but trade dynamic accuracy for diversity, highlighting the complementary role of generative models in capturing physio-realistic dynamics.

**ML-based Molecular Dynamics.** Modeling molecular dynamics is challenging due to complex multi-body interactions, data scarcity, and high-dimensional state spaces. Equivariant architectures such as EGNN (Satorras et al., 2021b) and SE(3)-Transformer (Fuchs et al., 2020) improve generalization by embedding physical symmetries (Brandstetter et al., 2022; Xu et al., 2024b), while autoregressive approaches like Timewarp (Klein et al., 2023) and EquiJump (dos Santos Costa et al., 2024) capture temporal transitions but suffer from error compounding and limited design flexibility. Diffusion-based methods address these issues by modeling trajectories holistically: GeoTDM (Han et al., 2024) enforces equivariance but requires molecule-specific training, and MDGen (Jing et al., 2024b) extends to peptide torsions with flow-based modeling but relies on key-frame conditioning. In contrast, our method generalizes more readily across arbitrary molecular systems.

**Video Generation from Image Models.** Blattmann et al. (2023) highlighted extending image diffusion models to videos by adding temporal layers, an idea motivating our spatial–temporal decoupling. Related work in latent image diffusion (Rombach et al., 2021) and holistic video generation (Brooks et al., 2024) further demonstrate the scalability of spatiotemporal diffusion.

## 3 PRELIMINARIES

**Geometric representation of molecular dynamics.** In this work, we represent each molecular dynamics trajectory as a collection of *static structures*, or equivalently *conformers* that evolve through time. Each frame of conformer at timestep $t$ is viewed as a geometric graph $\mathcal{G}^{(t)} := (\mathbf{h}, \mathbf{x}^{(t)}, \mathcal{E})$ where each row $\mathbf{h}_i \in \mathbb{R}^H$ is the node feature of atom $i$ such as its atomic number, $\mathbf{x}_i^{(t)} \in \mathbb{R}^3$ is the Euclidean coordinate of atom $i$ at timestep $t$, and $\mathcal{E}$ is the set of edges induced by the chemical bonds between atoms. The trajectory with length $T$ is correspondingly represented as $\mathbf{x}^{[T]} := \mathbf{x}^{(0:T-1)} \in \mathbb{R}^{T \times N \times 3}$.

**Geometric diffusion model for conformer generation.** Geometric diffusion models (Xu et al., 2022; Hoogeboom et al., 2022a; Xu et al., 2023) are a family of diffusion-based generative models (Sohl-Dickstein et al., 2015; Ho et al., 2020a; Song & Ermon, 2019; Song et al., 2021) dedicated to capture the distribution of static conformer structures $p(\mathbf{x}|\mathbf{h}, \mathcal{E})$, given the configuration of the molecular graph specified by the node feature $\mathbf{h}$ and edge connectivity $\mathcal{E}$. Inheriting the framework of diffusion models, they feature a Markovian forward noising process that gradually perturbs $\mathbf{x}_0$ toward $\mathbf{x}_{\mathcal{T}}$ through $\mathcal{T}$ diffusion steps, with the Gaussian transition kernel $q(\mathbf{x}_\tau|\mathbf{x}_{\tau-1}) = \mathcal{N}(\mathbf{x}_\tau; \sqrt{1-\beta_\tau}\mathbf{x}_{\tau-1}, \beta_\tau \mathbf{I})$, where $\beta_\tau$ is the noise schedule such that $\mathbf{x}_{\mathcal{T}}$ is close to the Gaussian prior $\mathcal{N}(\mathbf{0}, \mathbf{I})$. The reverse process denoises toward the clean data using $p_\theta(\mathbf{x}_{\tau-1}|\mathbf{x}_\tau) = \mathcal{N}(\mathbf{x}_{\tau-1}; \boldsymbol{\mu}_\theta(\mathbf{x}_\tau; \tau), \sigma_\tau^2 \mathbf{I})$. The model is optimized via Ho et al. (2020a):

$$\mathcal{L}_{\text{conf}} = \mathbb{E}_{\mathbf{x}_0 \sim \mathcal{D}_{\text{conf}}, \tau \sim \text{Unif}(1, \mathcal{T}), \boldsymbol{\epsilon} \sim \mathcal{N}(\mathbf{0}, \mathbf{I})} \|\boldsymbol{\epsilon} - \boldsymbol{\epsilon}_\theta(\mathbf{x}_\tau, \tau)\|_2^2, \tag{1}$$

where $\mathcal{D}_{\text{conf}}$ is the conformer dataset, $\mathbf{x}_\tau = \sqrt{\bar{\alpha}_\tau}\mathbf{x}_0 + \sqrt{1-\bar{\alpha}_\tau}\boldsymbol{\epsilon}$ with $\bar{\alpha}_\tau$ being certain noise schedule and $\boldsymbol{\epsilon}_\theta$ parameterizes the mean by $\boldsymbol{\mu}_{\boldsymbol{\theta}}(\mathbf{x}_\tau, \tau) = \frac{1}{\sqrt{\alpha_\tau}}(\mathbf{x}_\tau - \frac{\beta_\tau}{\sqrt{1-\bar{\alpha}_\tau}}\boldsymbol{\epsilon}_\theta(\mathbf{x}_\tau, \tau))$. A critical property of geometric diffusion models lies in the SE(3)-invariance of their marginal[1], *i.e.*, $p_\theta(\mathbf{x}_0) = g \cdot p_\theta(\mathbf{x}_0), g \in \text{SE}(3)$, where $g$ is an arbitrary group action in SE(3) that consists of all 3D rotations and translations, and $p_\theta(\mathbf{x}_0) = p(\mathbf{x}_{\mathcal{T}}) \prod_{\tau=1}^{\mathcal{T}} p_\theta(\mathbf{x}_{\tau-1}|\mathbf{x}_\tau)$. This is achieved by parameterizing $\boldsymbol{\epsilon}_\theta$ with an equivariant graph neural network (Satorras et al., 2021b;a) such that $g \cdot \boldsymbol{\epsilon}_\theta(\mathbf{x}_\tau, \tau) = \boldsymbol{\epsilon}_\theta(g \cdot \mathbf{x}_\tau, \tau)$ which guarantees the SE(3)-equivariance of the transition kernel $p_\theta(\mathbf{x}_{\tau-1}|\mathbf{x}_\tau)$ at each step $\tau$.

---

[1]For conciseness we henceforth omit the conditions $\mathbf{h}, \mathcal{E}$ in $p(\mathbf{x}_0|\mathbf{h}, \mathcal{E})$ unless otherwise specified.

**Problem definition.** In this work, we seek to design a diffusion model that captures the distribution of molecular dynamics $p^{\mathrm{md}}(\mathbf{x}^{[T]})$ given node features $\mathbf{h}$ and edges $\mathcal{E}$. Based on this goal, we are additionally interested in two relevant subtasks, namely *forward simulation*, which models the conditional distribution $p^{\mathrm{md}}(\mathbf{x}^{(1:T-1)}|\mathbf{x}^{(0)})$ given the initial structure $\mathbf{x}^{(0)}$, and *interpolation*, which models $p^{\mathrm{md}}(\mathbf{x}^{(1:T-2)}|\mathbf{x}^{(0)},\mathbf{x}^{(T-1)})$ given both the initial frame $\mathbf{x}^{(0)}$ and final frame $\mathbf{x}^{(T-1)}$.

## 4 METHOD

In this section, we present our approach for generating MD trajectories by temporally aligning structural distributions. § 4.1 introduces the overall framework of conformer pretraining and temporal alignment; § 4.2 describes the temporal interpolator that couples conformer and temporal layers; and § 4.3 details the implementation of EGINTERPOLATOR.

### 4.1 TRAJECTORY GENERATION BY ALIGNING STRUCTURE MODEL

**Motivation.** While substantial research has advanced the modeling of empirical conformer data distribution $p^{\mathrm{cf}}(\mathbf{x})$, generalizing this paradigm to molecular dynamics trajectories remains inherently challenging for two primary reasons. **1.** *Data scarcity.* Unlike conformer modeling, which benefits from extensive datasets (Ramakrishnan et al., 2014; Axelrod & Gomez-Bombarelli, 2022), molecular dynamics simulations incur prohibitive computational costs. Consequently, existing MD datasets (Chmiela et al., 2017; Meersche et al., 2024) are typically constrained to limited molecular classes, significantly restricting generalizeability across more arbitrarily defined molecular types. **2.** *Modeling complexity.* MD trajectories inhabit high-dimensional spaces with an additional temporal dimension. The inherent complexity of the joint distribution $p^{\mathrm{md}}(\mathbf{x}^{[T]})$ is further exacerbated by data scarcity, as insufficient training samples create greater sparsity in the high-dimensional data support, thereby complicating accurate density estimation.

**Our solution.** We propose to leverage a pretrained conformer diffusion model and transform it into an MD generation model, by stacking additional trainable temporal layers to enforce temporal consistency along each MD trajectory. Formally, given a pretrained conformer diffusion model $\epsilon_\theta$ inducing the marginal $p_\theta^{\mathrm{cf}}(\mathbf{x})$, we devise $\epsilon_{\theta,\phi}^{\mathrm{md}}$ for modeling the MD distribution $p_{\theta,\phi}^{\mathrm{md}}(\mathbf{x}^{[T]})$, where $\phi$ represents parameters in the additional temporal layers, indicating that the MD generative model with parameter set $\{\theta,\phi\}$ is partially initialized from the pretrained structure model $\theta$. The MD diffusion model is then optimized on the MD trajectory dataset with the diffusion loss

$$\mathcal{L}_{\mathrm{md}} = \mathbb{E}_{\mathbf{x}_0^{[T]}\sim\mathcal{D}_{\mathrm{md}},\tau\sim\mathrm{Unif}(1,\mathcal{T}),\boldsymbol{\epsilon}^{[T]}\sim\mathcal{N}(\mathbf{0},\mathbf{I})}\|\boldsymbol{\epsilon}^{[T]} - \boldsymbol{\epsilon}_{\theta,\phi}^{\mathrm{md}}(\mathbf{x}_\tau^{[T]},\tau)\|_2^2, \tag{2}$$

where $\mathbf{x}_\tau^{[T]} = \sqrt{\bar{\alpha}_\tau}\mathbf{x}_0^{[T]} + \sqrt{1-\bar{\alpha}_\tau}\boldsymbol{\epsilon}^{[T]}$ and $\boldsymbol{\epsilon}^{[T]} \in \mathbb{R}^{T\times N\times 3}$ is the Gaussian noise and $\mathcal{D}_{\mathrm{md}}$ is the MD dataset. Our proposal effectively addresses the core challenges. We mitigate MD data scarcity by initializing with a conformer model trained on large-scale conformer datasets, transferring generalization capability to unseen molecules. Furthermore, our two-stage pipeline decomposes the complex modeling of $p^{\mathrm{md}}(\mathbf{x}^{[T]})$ into manageable subproblems: conformer pretraining first models each frame independently, yielding an intermediate trajectory-level distribution $\hat{p}_\theta^{\mathrm{md}}(\mathbf{x}^{[T]}) := \prod_{t=0}^{T-1} p_\theta^{\mathrm{cf}}(\mathbf{x}^{(t)})$ that does not incorporate any temporal correlation. The second stage introduces additional parameters $\phi$ to capture the temporal dependency across different frames, leading to the joint distribution $p_{\theta,\phi}^{\mathrm{md}}(\mathbf{x}^{[T]})$. This approach efficiently offloads the complexity by using $\hat{p}_\theta^{\mathrm{md}}(\mathbf{x}^{[T]})$ as an anchor. The flowchart of our proposed framework is depicted in Fig. 1.

### 4.2 TEMPORAL INTERPOLATOR

With the proposed framework, it is still yet unrevealed how to allocate the additional parameters $\phi$ to capture the temporal dependency across frames for aligning the structures into an MD trajectory. To this end, we introduce a novel temporal interpolator module that entangles the pretrained structure denoiser $\epsilon_\theta^{\mathrm{cf}}$ with the additional temporal network $\epsilon_\phi^{\mathrm{tp}}$ through a linear interpolation:

$$\boldsymbol{\epsilon}_{\theta,\phi}^{\mathrm{md}}(\mathbf{x}_\tau^{[T]},\tau) = \alpha\hat{\boldsymbol{\epsilon}}^{\mathrm{md}} + (1-\alpha)\boldsymbol{\epsilon}_\phi^{\mathrm{tp}}(\mathbf{x}_\tau^{[T]},\hat{\boldsymbol{\epsilon}}^{\mathrm{md}},\tau), \qquad \text{s.t.} \quad \hat{\boldsymbol{\epsilon}}^{\mathrm{md}} = [\boldsymbol{\epsilon}_\theta^{\mathrm{cf}}(\mathbf{x}_\tau^{(t)},\tau)]_{t=0}^{T-1}, \tag{3}$$

where $\alpha \in \mathbb{R}$ is the interpolation coefficient, and $[\epsilon_\theta(\mathbf{x}_\tau^{(t)}, \tau)]_{t=0}^{T-1}$ is the concatenation along the temporal axis for the outputs $\epsilon_\theta^{\text{cf}}(\mathbf{x}_\tau^{(t)})$ at frames $0 \leq t \leq T-1$, and $\epsilon_\phi^{\text{tp}}(\mathbf{x}_\tau^{[T]}, \hat{\epsilon}^{\text{md}}, \tau) = \mathbf{s}_\phi^{\text{tp}}(\mathbf{x}_\tau^{[T]} + \hat{\epsilon}^{\text{md}}, \tau) - \mathbf{x}_\tau^{[T]}$ where $\mathbf{s}_\phi^{\text{tp}}$ is an equivariant temporal attention network (Han et al., 2024).

Intuitively, Eq. 3 mixes the output from the structure model $\epsilon_\theta^{\text{cf}}$ together with the the temporal model $\epsilon_\phi^{\text{tp}}$ as the final output $\epsilon_{\theta,\phi}^{\text{md}}$, making it both structural and temporal-aware. Notably, compared with other mixing strategies, our design has several unique benefits, as we analyzed below.

We start by showing that the interpolation mechanism in Eq. 3 implicitly induces an intermediate distribution for the temporal network to learn. We reveal such insight in the following theorem.

**Theorem 4.1.** *Suppose $\epsilon_\theta^{\text{cf}}$ perfectly models $p^{\text{cf}}(\mathbf{x})$ and $\epsilon_{\theta,\phi}^{\text{md}}$ perfectly models $p^{\text{md}}(\mathbf{x}^{[T]})$, then the interpolation in Eq. 3 implicitly induces the distribution $\tilde{p}^{\text{md}}(\mathbf{x}^{[T]}) \propto p^{\text{md}}(\mathbf{x}^{[T]})^\beta \hat{p}^{\text{md}}(\mathbf{x}^{[T]})^{1-\beta}$ for $\epsilon_\phi$, where $\beta = \frac{1}{1-\alpha}$ and $\hat{p}^{\text{md}} = \prod_{t=0}^{T-1} p^{\text{cf}}(\mathbf{x}^{(t)})$.*

**Temporal interpolator reduces training overhead.** Instead of directly matching the highly complex MD distribution $p^{\text{md}}(\mathbf{x}^{[T]})$, the temporal network is now expected to model an intermediate transition between the frame-independent distribution $\hat{p}^{\text{md}}(\mathbf{x}^{[T]})$ obtained from the structure model and the target MD distribution $p^{\text{md}}(\mathbf{x}^{[T]})$, with $\beta = \frac{1}{1-\alpha}$ defining the weight. By this means, we relieve from the optimization difficulty for learning the MD distribution by leveraging the interpolation $\tilde{p}^{\text{md}}(\mathbf{x}^{[T]})$ as the stepping stone, while also effectively taking advantage from the conformer pretraining by incorporating $p^{\text{cf}}(\mathbf{x}^{(t)})$ using $\hat{p}^{\text{md}}(\mathbf{x}^{[T]})$ as the bridge. The effectiveness of our design is also supported by the ablation study in Sec. A.8 which shows clear advantage of our approach compared against a naive two-stage separate training.

**The parameterization of $\epsilon_\phi^{\text{tp}}$.** Another core design lies in that we inherit the output from the structure model, $\hat{\epsilon}^{\text{md}}$, as the input to the temporal model, instead of only feeding in the original noised trajectory $\mathbf{x}_\tau^{[T]}$. This is beneficial in terms of facilitates the optimization for $\epsilon_\phi^{\text{tp}}$. Consider the extreme case that the frame-independent distribution is close to the MD distribution, $\hat{p}^{\text{md}}(\mathbf{x}^{[T]}) \approx p^{\text{md}}(\mathbf{x}^{[T]})$. According to Theorem 4.1, we have that the implicit distribution for the temporal model to approach would be $\tilde{p}^{\text{md}}(\mathbf{x}^{[T]}) \approx \hat{p}^{\text{md}}(\mathbf{x}^{[T]})$. Therefore, equivalently the temporal model only needs to satisfy $\epsilon_\phi^{\text{tp}}(\mathbf{x}_\tau^{[T]}, \hat{\epsilon}^{\text{md}}, \tau) \approx \hat{\epsilon}^{\text{md}}$, which can be simply realized by $\mathbf{s}_\phi^{\text{tp}}$ being an identity mapping, according to Eq. 3. Therefore, negligible optimization effort is required for $\mathbf{s}_\phi^{\text{tp}}$.

**Interpolation coefficient $\alpha$.** To further enhance thr training flexibility, empirically we adopt the parameterization of $\alpha = \sigma(k)$ where $\sigma(\cdot)$ is the Sigmoid function to ensure a smooth interpolation, where $k$ is a *learnable* parameter optimized during training.

**Temporal interpolator enables flexible inference.** Our design enables two inference modes. Setting $\alpha = 1$ suppresses the temporal network, reducing output to $\hat{\epsilon}^{\text{md}}$, equivalent to independent conformer generation for each frame with batch size $T$ and preserving conformer capability. Using the learned $\alpha^\star$ restores the full dynamics sampler. Shown in Appendix A.9.2, perturbations of $\alpha$ between these modes also yield meaningful inference behaviors, underscoring the flexibility of our approach.

**Temporal interpolator preserves equivariance.** Importantly, the linear interpolation rule for our temporal interpolator preserves the $\text{SE}(3)$-equivariance (proof in Appendix D.2), given the $\text{SE}(3)$-equivariance of both the structure and the temporal models. This property is vital for ensuring the $\text{SE}(3)$-invariance of the marginal, a critical inductive bias to promote data efficiency.

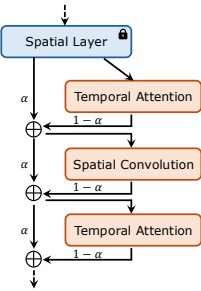

Figure 2: Cascaded temporal interpolator block.

**Cascaded temporal interpolator.** Given the justifications for the interpolator, we further explore an extension of our approach by performing such operation in a *block-wise manner*, enabling more expressive information fusion between the pretrained structure model and the additional temporal module. Specifically, we perform the interpolation for the output from the structure and temporal model at the $l$-th block with $\alpha^{(l)} \in \mathbb{R}$ being the coefficient. Furthermore, we also incorporate the interpolation between each layer in the temporal block and the output from the structure block. Detailed flowchart can be found in Fig. 2.

**A.** Coverage and Matching Results on QM9 and GEOM-Drugs

| | Method | COV-R (%) ↑ | | MAT-R (Å) ↓ | | COV-P (%) ↑ | | MAT-P (Å) ↓ | |
|---|---|---|---|---|---|---|---|---|---|
| | | Mean | Med. | Mean | Med. | Mean | Med. | Mean | Med. |
| **QM9** | CONFGF | 88.49 | 94.31 | 0.2673 | 0.2685 | 46.43 | 43.41 | 0.5224 | 0.5124 |
| | GEODIFF-A | 90.54 | 94.61 | 0.2104 | 0.2021 | 52.35 | 50.10 | 0.4539 | 0.4399 |
| | BASICES | 87.62 | 92.03 | 0.2574 | 0.2613 | 58.12 | 53.24 | 0.4451 | 0.4445 |
| **Drugs** | CONFGF | 62.15 | 70.93 | 1.1629 | 1.1596 | 23.42 | 15.52 | 1.7219 | 1.6863 |
| | GEODIFF-A | 88.36 | 96.09 | 0.8704 | 0.8628 | 60.14 | 61.25 | 1.1864 | 1.1391 |
| | BASICES | 92.35 | 100.00 | 0.8340 | 0.8245 | 65.59 | 70.87 | 1.1389 | 1.0973 |

**B.** Generated Conformers

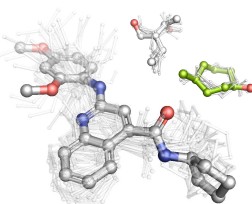

Figure 3: (**A**) reports performance of BASICES with borrowed numbers from Xu et al. (2022) on SOTA baselines; (**B**) Example conformers from BASICES on both QM9 & Drugs

Such design inherits the benefits of the interpolator while permitting a much denser information flow between the network that evidently improves optimization. We henceforth coin the original design SIMPLE and the cascaded version CASC.

### 4.3 INSTANTIATION OF EGINTERPOLATOR

Based on the dedicated design of the temporal interpolator in § 4.2, we describe the overall instantiation of our framework following the paradigm depicted in § 4.1.

**Conformer pretrainings stage.** The first stage of our pipeline is the structure pretraining using the large scale conformer dataset $\mathcal{D}_{cf}$. For the conformer model $\epsilon_\theta^{cf}$, we resort to Equivariant Graph Convolution Layer (EGCL) (Satorras et al., 2021b) as the basic building block with the update:

$$\mathbf{x}', \mathbf{h}' = f_{ES}(\mathbf{x}, \mathbf{h}, \mathcal{E}), \tag{4}$$

where ES is shorthand for Equivariant Structure layer. The denoiser $\epsilon_\theta$ consists of $L$ layers of $f_{ES}$ stacked sequentially, and is optimized using the loss in Eq. 1 for structure pretraining.

**MD training stage.** With the pretrained conformer model, we conduct the second stage, the MD training stage with the limited-size MD dataset $\mathcal{D}_{md}$, with the additionally initialized temporal network parameterized by $\mathbf{s}_\phi^{md}$. For the temporal network, we utilize the Equivariant Temporal Attention Layer introduced in Han et al. (2024) to capture the temporal dependency with attention:

$$\mathbf{x}'^{[T]}, \mathbf{h}'^{[T]} = f_{ET}(\mathbf{x}^{[T]}, \mathbf{h}^{[T]}, \mathcal{E}), \tag{5}$$

where ET refers to Equivariant Temporal layer. Each temporal block is a stack of three layers—ET at the top and bottom, with an ES layer in the middle—a design that promotes dense entanglement of structural and temporal features. For every ES layer in the pretrained model, we initialize one temporal block; together, these form $L$ interpolator blocks. The model is trained with the trajectory denoising loss (Eq. 2), freezing the pretrained ES layers. This yields a performant MD generative model without degrading conformer generation performance—an assurance not achieved in prior work. Appendix A.8.4 details the contribution of the temporal module and MD training, while Appendix A.9, E.8 interpret the learned $\alpha$ values.

**Forward simulation and interpolation.** Our model naturally supports structure-conditioned MD generation: forward simulation conditions on the first frame $\mathbf{x}^{(0)}$, and interpolation on both $\mathbf{x}^{(0)}$ and $\mathbf{x}^{(T-1)}$. Conditioning frames are treated as control signals, passed with noisy frames through the interpolator, and removed before loss computation to ensure the loss is applied only to noisy frames.

## 5 EXPERIMENTS

We refer to our framework as EGINTERPOLATOR and evaluate its ability to generate realistic MD trajectories for unseen molecules under practical data constraints with limited MD simulations and diverse static structural data. We focus first on small organic molecules and then further extend our analysis and framework to tetrapeptides and protein monomers.

### 5.1 CONFORMER PRETRAINING

**Datasets.** We use GEOM-QM9 (Ramakrishnan et al., 2014) and GEOM-Drugs (Axelrod & Gomez-Bombarelli, 2022) following prior work in conformer generation (Xu et al., 2022; Ganea et al., 2021).

Table 1: Performance Comparison on QM9 Unconditional Generation and Drugs Forward Simulation.

| | Method | JSD (Mean — Median) (↓) | | | | | | | | | |
|---|---|---|---|---|---|---|---|---|---|---|---|
| | | Bond Angle | | Bond Length | | Torsion | | TICA_0 | | TICA_0,1 | |
| QM9 | MD ORACLE | 0.042 | 0.028 | 0.032 | 0.031 | 0.192 | 0.134 | 0.318 | 0.291 | 0.413 | 0.394 |
| | AR + EGNN | 0.702 | 0.677 | 0.770 | 0.780 | 0.702 | 0.761 | 0.770 | 0.788 | 0.820 | 0.824 |
| | AR + ET | 0.705 | 0.746 | 0.680 | 0.721 | 0.553 | 0.586 | 0.568 | 0.562 | 0.783 | 0.786 |
| | AR + GEOTDM | 0.752 | 0.746 | 0.699 | 0.694 | 0.466 | 0.506 | 0.456 | 0.463 | 0.714 | 0.719 |
| | GEOTDM | 0.691 | 0.690 | 0.676 | 0.670 | 0.489 | 0.527 | 0.449 | 0.453 | 0.691 | 0.694 |
| | EGINTERPOLATOR-SIMPLE | 0.357 | 0.350 | 0.263 | 0.246 | 0.381 | 0.405 | 0.426 | 0.423 | 0.652 | 0.655 |
| | EGINTERPOLATOR-CASC | **0.305** | **0.292** | **0.210** | **0.188** | **0.363** | **0.380** | **0.417** | **0.406** | **0.636** | **0.642** |
| Drugs | MD ORACLE | 0.036 | 0.023 | 0.030 | 0.028 | 0.215 | 0.131 | 0.484 | 0.494 | 0.610 | 0.630 |
| | AR + EGNN | 0.663 | 0.655 | 0.748 | 0.784 | 0.723 | 0.741 | 0.716 | 0.731 | 0.806 | 0.821 |
| | AR + ET | 0.765 | 0.766 | 0.733 | 0.745 | 0.526 | 0.533 | 0.565 | 0.558 | 0.791 | 0.795 |
| | AR + GEOTDM | 0.608 | 0.611 | 0.613 | 0.613 | 0.509 | 0.497 | 0.504 | 0.505 | 0.727 | 0.725 |
| | GEOTDM | 0.640 | 0.645 | 0.643 | 0.645 | 0.498 | 0.503 | 0.531 | 0.550 | 0.712 | 0.720 |
| | EGINTERPOLATOR-SIMPLE | 0.208 | 0.192 | 0.258 | 0.244 | 0.385 | 0.399 | 0.462 | 0.465 | 0.660 | 0.662 |
| | EGINTERPOLATOR-CASC | **0.173** | **0.153** | **0.1419** | **0.112** | **0.377** | **0.388** | **0.454** | **0.441** | **0.650** | **0.644** |

Our spatial model is pretrained separately on each dataset, using the same train/validation splits as (Xu et al., 2022) and a preprocessing pipeline similar to (Ganea et al., 2021) (Appendix B.1.1). This results in 37.7K/4.7K training/validation molecules with 188.6K/23.7K conformers for QM9 and 38.0K/4.8K training/validation molecules with 190.0K/23.7K conformers for Drugs. We then use the same test sets from (Xu et al., 2022; Shi et al., 2021a), consisting of 200 distinct molecules, with 22.4K conformers for QM9 and 14.3K for Drugs.

**Experimental Setup & Baselines** We train our base BASICES model on this conformer generation task up to 800K steps for both QM9 and Drugs, learning 1000 denoising steps over only heavy atom coordinates. We compare the performance of our pretrained spatial models to that reported in (Xu et al., 2022), namely GEODIFF-A as well as CONFGF (Shi et al., 2021a).

**Metrics.** Per prior work in the space, we utilize the **Cov**erage and **Mat**ching metrics (Ganea et al., 2021; Xu et al., 2022) (Appendix B.1.3). We report both the Recall (R) to measure diversity and Precision (P) to measure accuracy. We use default $\delta$ **Cov**erage values, 0.5Å / 1.25Å (QM9/Drugs).

**Results & Discussion.** Results are summarized in Figure 3. Our pretrained BASICES model performs competitively with prior SOTA methods. For QM9, we prioritize precision-based metrics relevant to MD pretraining, which leads to slightly lower COV/MAT-R scores but superior fidelity in conformer bond angle and bond length distributions (see Appendix A.2).

## 5.2    MOLECULAR DYNAMICS FINETUNING

To generate MD data for diverse organic and drug-like molecules, we subsample from GEOM, resulting in 1109/1018/240 train/validation/test splits for QM9 and 1137/1044/100 for Drugs. We then perform five, all-atom (including hydrogens), explicit-solvent simulations of 5 ns per molecule. In the test set, four trajectories are used as reference data and the fifth serves as an oracle baseline (MD ORACLE). Full simulation and force field details are provided in the Appendix B.2.

**Experimental Setup & Baselines.** Unless otherwise noted, all models are trained with trajectory time-steps $\Delta t = 5.2$ ps. We learn across heavy atoms and use 1000 denoising steps. We compare our EGINTERPOLATOR framework against several representative approaches. First, we evaluate against GEOTDM (Han et al., 2024), a recent all-atom trajectory diffusion model. We also implement Markovian autoregressive baselines using EGNN (Hoogeboom et al., 2022a) and the Equivariant Transformer (Thölke & Fabritiis, 2022) as push-forward networks, denoted AR + EGNN and AR + ET, respectively. Finally, inspired by dos Santos Costa et al. (2024), we include a autoregressive diffusion baseline that adopts GeoTDM's architecture, denoted AR + GEOTDM.

## 5.3    UNCONDITIONAL GENERATION

In the *unconditional generation* setting, we train models to generate 2.6 ns trajectories with no reliance on a reference frame. For evaluation, we sample ten unconditional generations per molecule, resulting in 26 ns of generated trajectories. We focus on QM9 for this setting given the smaller

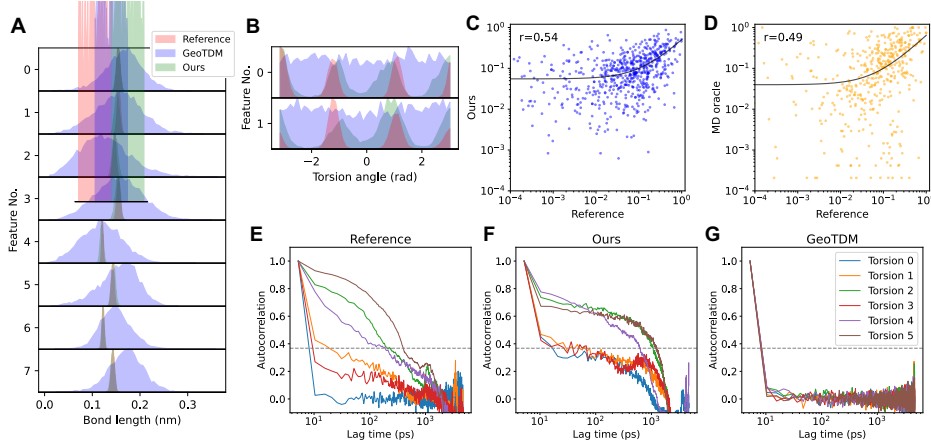

Figure 4: (**A**) Bond length and (**B**) torsion angle distributions from reference (red), our generations (green), and GeoTDM (blue). MSM occupancies from reference versus (**C**) our generations and (**D**) MD oracles. Autocorrelations of torsion angles for an example molecule from (**E**) reference, (**F**) our generations, and (**G**) GeoTDM. Gray dashed line marks the 1/e decorrelation threshold.

memory footprint of these molecules. In Appendix A.5, we also highlight block diffusion roll-outs for GEOM-Drugs in an unconditional manner.

**Distributional & Energetic Results.** A prerequisite to good molecular dynamics, we evaluate similarity between generated and reference trajectories using average Jensen–Shannon divergence (JSD) across key collective variable distributions: bond lengths and angles (energetically constrained features), as well as torsions. As shown in Table 1, EGINTERPOLATOR consistently outperforms baselines, with the CASC variant further improving over SIMPLE. Figure 4A,B examples illustrate gains over GeoTDM (Han et al., 2024) , with near-perfect alignment to ground-truth bond-length distributions, closely matched tri-modal torsion profiles, and similar trends across additional collective variables in Figure 11. Moreover, important potential energy analyses are reported in Appendix A.7, E.5 , where EGINTERPOLATOR shows markedly improved agreement over GEOTDM.

## 5.4 FORWARD SIMULATION

In the *forward simulation* setting, models are trained to generate 1.3 ns trajectories conditioned on a reference frame. We then extend these to 5.2 ns using successive block diffusion roll-outs, sampling five such trajectories per molecule. This setting focuses on GEOM-Drugs, targeting larger molecules.

**Distributional & Energetic Results.** Across all metrics in Table 1, EGINTERPOLATOR outperforms baselines and approaches the distributional fidelity of the replicate MD ORACLE on torsions. We once again see that the CASC variant further improves SIMPLE. Additionally, complementary potential energy analyses, including quantifying error propagation in short (4-block) and long (16-block) diffusion roll-outs, are reported in Appendix A.7, E.5 and further support our methods.

**Dynamical Results.** Assessing the dynamical consistency of our model, Table 1 shows that our method outperforms baselines and approaches the MD oracle in the distribution of the leading *time-lagged independent component analysis* (TICA) components, which capture the system's slow dynamics. We evaluate torsional dynamics via decorrelation time and find that EGINTERPOLATOR better captures distinct relaxation behaviors within molecules compared to GeoTDM (Fig. 4E,F,G), although certain fast relaxations seem to be a challenge. Furthermore, by constructing Markov State Models (MSMs) from torsion angles and clustering into 10 metastates, we observe agreement in metastate occupancy between generated and reference trajectories (Fig. 4C). Our model even surpasses MD oracle baselines in capturing coarse-grained dynamical distributions (Fig. 4D).

## 5.5 INTERPOLATION

In the *interpolation* (or *transition path sampling*) task, models generate 0.52 ns trajectories conditioned on both start and end frames. As this setting requires endpoint conditioning, we compare only to the ML baseline GeoTDM (Han et al., 2024). Results are reported for Drugs (QM9 in Appendix

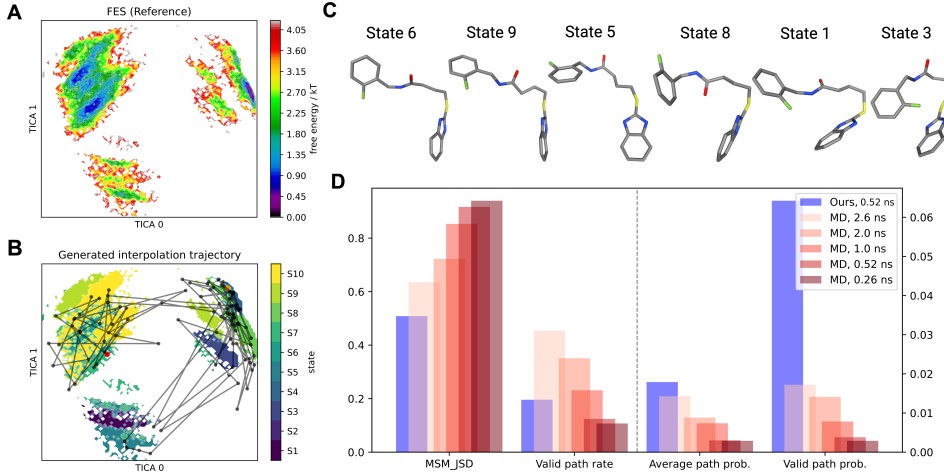

Figure 5: (**A**) Reference free energy surface along the top two TICA components. (**B**) Generated interpolation trajectory projected onto the reference surface (red = start, orange = end). Surface is colored by metastate assignment. (**C**) Key frames from intermediate metastates. (**D**) Statistics comparing JSD, valid path rate, average path probability, and valid path probability for generated trajectories and replicate MD oracles.

A.3), using the MSM pipeline from Jing et al. (2024c) to benchmark against MD oracles of varying lengths. Given prior stronger empirical performance, we use the CASC variant for this task.

**Evaluation.** Following Jing et al. (2024c), we frame interpolation as transition path sampling. An MSM built from reference trajectories defines two distant metastates as start and end states, from which we sample 900 frame pairs. Our model generates 900 corresponding trajectories, evaluated against reference and MD oracles using JSD over metastate occupancies. Owing to the high barrier and rare transitions, we also report valid path rate, average path probability, and valid path probability.

**Results.** As shown in Fig. 5D, our $0.52$ ns trajectories yield the lowest JSD and highest average path probability, outperforming MD oracles of equal length and matching longer ones in path quality. Although long oracles achieve higher valid path rates, our model excels at generating high-probability valid transitions. Fig. 5A,B further show a generated trajectory traversing key metastates on the reference FES, efficiently reaching the target end states.

## 5.6 ABLATION STUDY

We present main ablations here, with additional in Appendix A.8.

**Structural Pretraining.** We evaluate EGINTERPOLATOR-**N**aive, trained without conformer pretraining. In Table 2, this yields degraded bond length, angle, torsion fidelity, and diminished TICA_0,1.

Table 2: Ablation on QM9 Unconditional Generation and Drugs Forward Simulation.

| | Method | JSD (Mean — Median) ($\downarrow$) | | | | | | | |
|---|---|---|---|---|---|---|---|---|---|
| | | Bond Angle | | Bond Length | | Torsion | | TICA_0,1 | |
| | | Mean | Median | Mean | Median | Mean | Median | Mean | Median |
| QM9 | EGINTERPOLATOR-N | 0.538 | 0.538 | 0.583 | 0.580 | 0.441 | 0.494 | 0.680 | 0.685 |
| | **EGINTERPOLATOR** | **0.305** | **0.292** | **0.210** | **0.188** | **0.363** | **0.380** | **0.636** | **0.642** |
| Drugs | EGINTERPOLATOR-S | 0.325 | 0.330 | 0.330 | 0.321 | 0.414 | 0.419 | 0.673 | 0.672 |
| | EGINTERPOLATOR-N | 0.332 | 0.332 | 0.386 | 0.383 | 0.455 | 0.466 | 0.698 | 0.703 |
| | **EGINTERPOLATOR** | **0.173** | **0.153** | **0.142** | **0.112** | **0.377** | **0.388** | **0.650** | **0.644** |

**Interpolation and Architecture** EGINTERPOLATOR-**STACK** removes (1) our cascaded layer design and (2) interpolation, using a residual stack of temporal modules atop pretrained spatial layers as a finetuned head. In Table 2, this variant underperforms EGINTERPOLATOR, underscoring the importance of our interpolation architecture.

## 5.7 TETRAPEPTIDES

We extend our evaluation to tetrapeptides using the Timewarp dataset (Klein et al., 2023), which comprises 1500/400/433 train/validation/test sequences simulated for up to 50 ns (train) and 500 ns (validation/test) under all-atom implicit-solvent MD. Models are trained with $\Delta t = 10$ ps and generate 2.5 ns rollouts conditioned on a reference frame, which are iteratively composed into 10

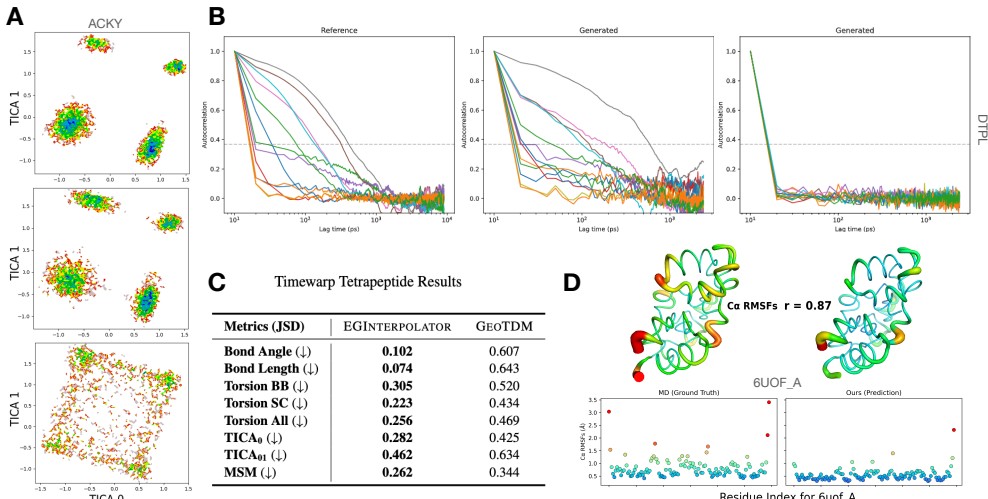

Figure 6: (**A**) Free energy surface along top two TICA components for a reference (**top**), ours (**top**), and GeoTDM (**bottom**) tetrapeptide trajectory. (**B**) Torsion auto-correlations from ours reference (**left**), ours (**middle**), and GeoTDM (**right**) (**C**) Key collective variable distribution JSD metrics. (**D**) C-$\alpha$ RMSF analysis and visualization for selected protein dynamics generation.

ns trajectories, with five samples per peptide. We compare EGINTERPOLATOR against GEOTDM, evaluating all methods against 50 ns reference simulations from the test set. As no conformer dataset exists for tetrapeptides, we construct one directly from the Timewarp training frames, detailed in Appendix B.1.1. This yields 1057/200 training and validation peptides (10.5K/2.7K conformers).

### 5.7.1 RESULTS

As shown in Figure 6C, our method significantly lowers the JSD of both backbone and side-chain torsions relative to GeoTDM. This advantage is reflected in the free-energy landscapes (Figure 6A), where EGINTERPOLATOR exhibits sharper, better-resolved basins, while GeoTDM remains diffuse and unstructured. These gains also carry over to potential-energy metrics reported in Appendix A.7. Beyond per-frame fidelity, our model attains markedly improved dynamical consistency, achieving lower JSD in the leading TICA components and MSM occupancies (Figure 6C), as well as well-aligned de-correlation times (Figure 6B).

### 5.8 PROTEIN SIMULATION

We extend our framework to protein monomer dynamics using the ATLAS dataset (Meersche et al., 2024), training a forward-simulation model and following the data splits of Jing et al. (2024a). Building on Boltz1 (Wohlwend et al., 2024), we incorporate a temporal module—pair-biased sliding-window spatial attention, per Boltz1, combined with interleaved RoPE-based temporal attention layers (Su et al., 2023) across atoms/tokens—to enable trajectory generation. During training, we apply random rigid-body augmentations and superpose trajectories to a zero-reference frame. Our experiments trained on 200 proteins, with 30/50 for validation/testing, generating 250-frame segments at 100 ps and composing four such blocks for 100 ns rollouts. Preliminary results on the example protein from Jing et al. (2024b) are shown in Figure 6. Using 100 diffusion steps, generations take 0.12 (s) per frame.

## 6 CONCLUSION

We have introduced a diffusion model for modeling MD distributions by pretraining a structure model on conformer dataset and then finetuning on trajectory dataset. At the core of our approach is an module named EGINTERPOLATOR that mixes the output from the pretrained structure model and the temporal model to captures the temporal dependency. Our approach demonstrates strong performance in terms of producing realistic MD trajectories on diverse benchmarks and tasks.

## ACKNOWLEDGMENT

We thank the anonymous reviewers for their feedback on improving the manuscript. This work was supported by ARO (W911NF-21-1-0125), ONR (N00014-23-1-2159), and the CZ Biohub.

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

# A  EXPERIMENTS CONTINUED

## A.1  COMPARISON TO MDGEN ON THE TETRAPEPTIDE DATASET

We benchmark our model against MDGen (Jing et al., 2024b), which parameterizes tetrapeptide conformations explicitly through backbone and sidechain torsional angles. Using the subsampling procedure described in Appendix B.1.1, we convert MDGen training trajectories into a peptide conformer dataset and augment it with a pruned TimeWarp-derived conformer set to avoid data leakage. We then fine-tune our conformer model on this combined dataset, initializing from GEOM-DRUGS pretrained weights. As shown in Table 3, our downstream peptide EGINTERPOLATOR preserves fine-grained structural information and superior fidelity in bond lengths and bond angles, which are essential for accurate all-heavy-atom molecular dynamics and modeling small molecules. However, it underperforms MDGen on torsional distributions and torsion-derived dynamical metrics. This gap suggests that MDGen's torsion-centric representation confers an advantage in capturing peptide rotational behavior, where our model's strengths are geometric consistency at an atomic level.

## A.2  OPTIMIZING FOR CONFORMER PRECISION METRICS

As discussed in Section 5.1, we prioritize precision-based conformer quality metrics when selecting our base structure model. While this may come at the cost of lower COV/MAT-R scores, we observe superior fidelity in bond length, bond angle, and torsion angle distributions—an aspect we consider more critical for a pretrained structure module.

We highlight this point using two checkpoints of the BASICES model trained on QM9. In Table 4 we can see that while *539* lacks in COV-R, it does substantially better than *99* in COV/MAT-P metrics. In Figure 10, we then see that *539* reflects high quality bond angle, length, and torsion distributions, as compared to *99*. We select checkpoint *539* for the conformer results reported in Section 5.1 and for training the downstream trajectory models.

Table 3: Results on MDGen Tetrapeptide Dataset

| Metrics (JSD) | EGINTERPOLATOR | MDGEN |
|---|---|---|
| **Bond Angle** ($\downarrow$) | 0.092 | N/A |
| **Bond Length** ($\downarrow$) | 0.056 | N/A |
| **Torsion BB** ($\downarrow$) | 0.378 | 0.130 |
| **Torsion SC** ($\downarrow$) | 0.189 | 0.093 |
| **Torsion All** ($\downarrow$) | 0.265 | 0.109 |
| **TICA$_0$** ($\downarrow$) | 0.409 | 0.230 |
| **TICA$_{01}$** ($\downarrow$) | 0.568 | 0.316 |
| **MSM** ($\downarrow$) | 0.312 | 0.235 |

Table 4: Conformer metrics on QM9 compared between two checkpoints.

| Checkpoint | COV-R (%) $\uparrow$ | | MAT-R (Å) $\downarrow$ | | COV-P (%) $\uparrow$ | | MAT-P (Å) $\downarrow$ | |
|---|---|---|---|---|---|---|---|---|
| | Mean | Med. | Mean | Med. | Mean | Med. | Mean | Med. |
| *99* | **90.18** | **94.59** | 0.2969 | 0.3049 | 55.23 | 51.36 | 0.4932 | 0.4823 |
| *539* | 87.62 | 92.03 | **0.2574** | **0.2613** | **58.12** | **53.24** | **0.4451** | **0.4445** |

## A.3 QM9 INTERPOLATION

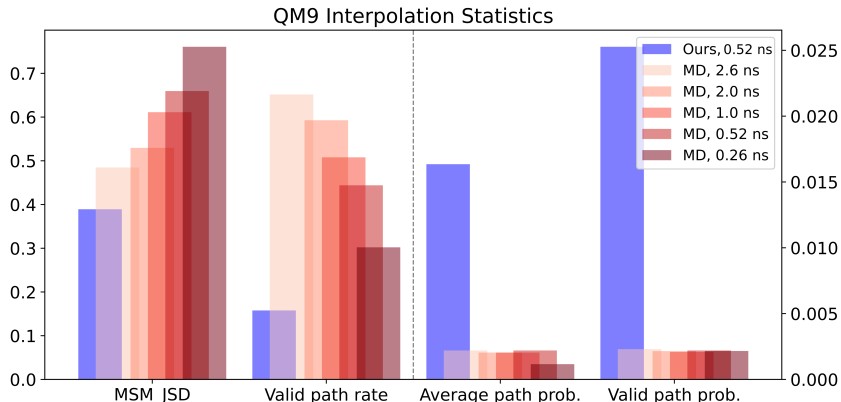

Figure 7: Statistics evaluating the JSD with the reference trajectories, valid path rate, average path probability, and valid path probability of our generated trajectories and replicate MD oracles.

For the interpolation task on QM9 dataset, as shown in Figure 7, our 0.52 ns trajectories from CASC consistently achieve the lowest Jensen-Shannon Divergence (JSD) and the highest average path probability, outperforming MD oracles of the same duration. It reveals that our method can samples transition paths between far metastates more efficiently. While the MD oracles exhibit higher valid path rates in this setting, our model still performs competitively in generating high-probability valid transitions.

Figure 14 illustrates several free energy surfaces (FES) and corresponding metastate assignments for representative molecules. We observe that the generated trajectories successfully traverse key intermediate states and reach the appropriate end states, demonstrating the model's ability to perform efficient and meaningful transition path sampling.

## A.4 DRUGS UNCONDITIONAL GENERATION

Since the molecules in the Drugs dataset are more challenging systems than those in QM9, we further ablate the reliance on the starting reference frame by conducting an unconditional generation

Table 5: Performance comparison on Drugs Forward Simulation versus Unconditional Generation. Reported values are JSD (Mean — Median) ↓.

| Method | Bond Angle | Bond Length | Torsion | $TICA_0$ | $TICA_{0,1}$ |
|---|---|---|---|---|---|
| GEOTDM | 0.640 0.645 | 0.643 0.645 | 0.498 0.503 | 0.531 0.550 | 0.712 0.720 |
| EGINTERPOLATOR-SIMPLE | 0.208 0.192 | 0.258 0.244 | 0.385 0.399 | 0.462 0.465 | 0.660 0.662 |
| EGINTERPOLATOR-CASC | 0.173 0.153 | 0.142 0.112 | 0.377 0.388 | 0.454 0.441 | 0.650 0.644 |
| EGINTERPOLATOR-CASC-U | 0.220 0.202 | 0.195 0.168 | 0.414 0.429 | 0.499 0.496 | 0.689 0.697 |

experiment (U). Specifically, we retain the same experimental set-up but remove conditioning of the first block on a ground-truth frame, and retrain a new unconditional generation model. As shown in Table 5, while performance does not match our EGINTERPOLATOR-CASC trained with forward simulation, the unconditional variant still surpasses GEOTDM trained with forward simulation by a significant margin in terms of bond angle, bond length, and torsion distribution fidelity.

## A.5 DRUGS LONG SIMULATION

To more rigorously evaluate generation quality, we repeat the forward-simulation experiments on DRUGS using a long 16-block roll-out of 20.8 ns, generating one trajectory per molecule. Although this extended roll-out exhibits some degradation relative to the parallelized version—likely due to accumulated error propagation quantified in Section A.7—our method still outperforms all baselines, including their parallel 4-block configurations, shown in Table 6.

## A.6 MULTITASK/MODAL LEARNING

To further demonstrate our framework's ability to generalize across molecular dynamics regimes, we first pre-train a conformer model and then train a dynamics interpolator jointly on QM9 and DRUGs for both forward simulation and unconditional generation. We benchmark this unified model against single-task counterparts on each dataset. As shown in Table 6, the unified model consistently outperforms all baselines, and notably achieves improved performance on QM9—indicating that pretraining on more diverse and chemically complex systems can enhance dynamics generation quality even on previously unseen molecules.

Table 6: Additional Performance Comparisons on QM9 Unconditional Generation and Drugs Forward Simulation.

| | Method | JSD (Mean — Median) (↓) | | | | | | | | | |
|---|---|---|---|---|---|---|---|---|---|---|---|
| | | Bond Angle | | Bond Length | | Torsion | | $TICA_0$ | | $TICA_{0,1}$ | |
| QM9 | MD ORACLE | 0.042 | 0.028 | 0.032 | 0.031 | 0.192 | 0.134 | 0.318 | 0.291 | 0.413 | 0.394 |
| | GEOTDM | 0.691 | 0.690 | 0.676 | 0.670 | 0.489 | 0.527 | 0.449 | 0.453 | 0.691 | 0.694 |
| | EGINTERPOLATOR | 0.305 | 0.292 | 0.210 | 0.188 | 0.363 | 0.380 | 0.417 | 0.406 | 0.636 | 0.642 |
| | EGINTERPOLATOR-BOTH | **0.231** | **0.219** | **0.168** | **0.158** | **0.348** | **0.367** | **0.393** | **0.390** | **0.623** | **0.631** |
| Drugs | MD ORACLE | 0.036 | 0.023 | 0.030 | 0.028 | 0.215 | 0.131 | 0.484 | 0.494 | 0.610 | 0.630 |
| | GEOTDM | 0.640 | 0.645 | 0.643 | 0.645 | 0.498 | 0.503 | 0.531 | 0.550 | 0.712 | 0.720 |
| | EGINTERPOLATOR | **0.173** | **0.153** | **0.142** | **0.112** | **0.377** | **0.388** | **0.454** | **0.441** | **0.650** | **0.644** |
| | EGINTERPOLATOR-BOTH | 0.212 | 0.197 | 0.216 | 0.195 | 0.417 | 0.434 | 0.488 | 0.506 | 0.681 | 0.679 |
| | EGINTERPOLATOR-LONG | 0.180 | 0.155 | 0.147 | 0.116 | 0.404 | 0.411 | 0.484 | 0.484 | 0.685 | 0.680 |

## A.7 ENERGY-BASED ANALYSIS

In addition to evaluating collective variable distributions and MSM metrics as measures of trajectory fidelity, we further assess model rigor by examining the energy profiles of generated trajectories. Per-frame energies are estimated using TorchANI2x (Gao et al., 2020) and reported in Hartrees. Alongside the results presented in this section, we also provide energy comparisons to ground truth trajectories for representative molecules from both datasets in Table 17.

Table 7: **Top:** Average Wasserstein-1 (W1) distance between predicted and ground-truth (GT) energy profiles for EGINTERPOLATOR-CASC and GEOTDM across dataset test sets. **Bottom:** Per-block W1 analysis in forward simulation roll-outs for Drugs.

| Dataset | EGInterpolator vs GT W1 ↓ | GeoTDM vs GT W1 ↓ |
|---|---|---|
| QM9 | **0.8127** | 2.9201 |
| Drugs | **0.7728** | 12.7664 |

| Block | EGInterpolator vs GT W1 ↓ | GeoTDM vs GT W1 ↓ |
|---|---|---|
| 1 | **0.2454** | 11.2398 |
| 2 | **0.3654** | 12.8999 |
| 3 | **0.3656** | 13.0270 |
| 4 | **0.3702** | 13.1235 |

Table 8: **Top:** Average Wasserstein-1 (W1) distance between predicted and ground-truth (GT) energy profiles for EGINTERPOLATOR and GEOTDM across tetrapeptides. **Bottom:** Per-block W1 analysis of forward simulation roll-outs.

| Metric | EGINTERPOLATOR vs GT W1 ↓ | | | |
|---|---|---|---|---|
| Overall Energy W1 | **0.3806** | | | |
| GEOTDM Energy W1 | 12.8636 | | | |

| Block | 1 | 2 | 3 | 4 |
|---|---|---|---|---|
| EGINTERPOLATOR W1 | **0.2638** | **0.3912** | **0.4196** | **0.4494** |
| GEOTDM W1 | 12.4955 | 12.9417 | 13.0007 | 13.0681 |

### A.7.1 OVERALL RESULTS

In Table 7, 8, we report the Wasserstein-1 (W1) distance between the energy distributions of generated trajectories and the ground-truth (GT) trajectories, averaged across the test sets of all datasets. Our framework achieves substantially lower W1 distances than the GEOTDM baseline, demonstrating much closer correspondence to the GT energy profiles.

### A.7.2 BLOCK DIFFUSION DETERIORATION

In Tables 7 and 8, we address a central concern in forward roll-outs using block diffusion: the potential for error accumulation and degradation in sample fidelity over time. To quantify this, we conduct a block-wise analysis of the generated trajectories and observe that our framework remains well aligned with ground-truth energy distributions, exhibiting only mild deterioration—most notably between Blocks 1 and 2. Extending this analysis to longer roll-outs, we find that unlike GeoTDM, our model does not collapse to degenerate energy states, though errors begin to compound beyond approximately 8 frames. Mitigating this effect is a promising direction for future work, where incorporating force or energy-based guidance during training or inference may further improve long-horizon stability.

### A.8 TRAJECTORY MODEL ABLATIONS

### A.8.1 FROZEN BASICES

As mentioned in Section 5.6, we assess the benefit of fine-tuning the frozen spatial encoder by training a fully end-to-end version of EGINTERPOLATOR, called EGINTERPOLATOR-**F**, on the Drugs forward simulation task. In Figure 8, we see that performance remains largely unchanged across metrics, indicating that the pretrained spatial model generalizes well without task-specific tuning, while the temporal layers effectively capture the necessary dynamic information.

Table 9: **Block-wise Wasserstein-1 Progression.** Mean W1 error across 250-frame blocks during forward simulation.

| Block | Frame Range | Mean W1 ($\downarrow$) |
|---|---|---|
| 1 | 1–250 | 0.2303 |
| 2 | 251–500 | 0.2778 |
| 3 | 501–750 | 0.3693 |
| 4 | 751–1000 | 0.2769 |
| 5 | 1001–1250 | 0.4541 |
| 6 | 1251–1500 | 0.3962 |
| 7 | 1501–1750 | 0.3830 |
| 8 | 1751–2000 | 0.3805 |
| 9 | 2001–2250 | 0.5765 |
| 10 | 2251–2500 | 0.5861 |
| 11 | 2501–2750 | 0.6666 |
| 12 | 2751–3000 | 0.3979 |
| 13 | 3001–3250 | 0.4276 |
| 14 | 3251–3500 | 0.5328 |
| 15 | 3501–3750 | 0.4830 |
| 16 | 3751–4000 | 0.5192 |

| Temporal Region | Mean W1 ($\downarrow$) |
|---|---|
| Early (Blocks 1–4; Frames 1–1000) | 0.3460 |
| Mid (Blocks 5–12; Frames 1001–3000) | 0.5568 |
| Late (Blocks 13–16; Frames 3001–4000) | 0.4906 |

### A.8.2 GENERALIZATION TO AN EXTENDED TEST SET

To further assess the robustness of our QM9 unconditional generation model, we evaluate performance on an extended test set of 959 molecules, which includes the original test set from Section 5.2. As shown in Table 10, we compare GEOTDM (Han et al., 2024), EGINTERPOLATOR-**N** (without structure pretraining), and our full EGINTERPOLATOR model. While all models perform comparably on this larger evaluation set, EGINTERPOLATOR consistently outperforms the baselines, underscoring its strong generalization and the value of structural pretraining.

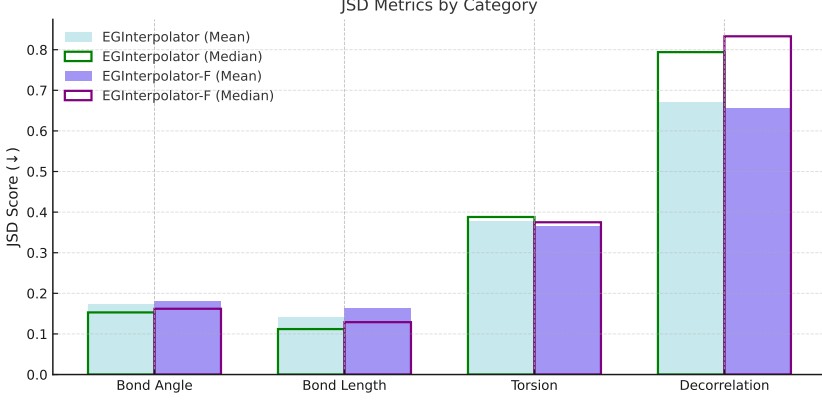

Figure 8: JSD metrics computed for Bond Angles, Bond Lengths, Torsions, and Decorrelation Times. Compared between EGINTERPOLATOR (green) and EGINTERPOLATOR-**F** (purple).

Table 11: JSD metrics for bond angle, bond length, and torsion across QM9 and DRUGS datasets with and without temporal layers.

| Model | Bond Angle ($\downarrow$) | Bond Length ($\downarrow$) | Torsion ($\downarrow$) |
|---|---|---|---|
| EGINTERPOLATOR NORMAL (QM9) | 0.305 / 0.292 | 0.210 / 0.188 | 0.363 / 0.380 |
| EGINTERPOLATOR $\alpha = 1$ (QM9) | 0.398 / 0.391 | 0.618 / 0.613 | 0.358 / 0.372 |
| EGINTERPOLATOR NORMAL (DRUGS) | 0.173 / 0.153 | 0.142 / 0.112 | 0.377 / 0.388 |
| EGINTERPOLATOR $\alpha = 1$ (DRUGS) | 0.435 / 0.445 | 0.580 / 0.091 | 0.378 / 0.382 |

Table 10: JSD Metric ($\downarrow$) for QM9 Unconditional Generation. Top: Trained on **Standard** Train, evaluated on **Enlarged** Test. Bottom: Trained on **Enlarged** Train, evaluated on **Standard** Test.

| Train $\rightarrow$ Test | Method | Bond Angle | | Bond Length | | Torsion | | TICA_0 | | TICA_0,1 | |
|---|---|---|---|---|---|---|---|---|---|---|---|
| | | Mean | Med. | Mean | Med. | Mean | Med. | Mean | Med. | Mean | Med. |
| | GEOTDM | 0.690 | 0.690 | 0.674 | 0.668 | 0.488 | 0.529 | 0.452 | 0.451 | 0.695 | 0.699 |
| **Standard $\rightarrow$ Enlarged** | EGINTERPOLATOR-N | 0.539 | 0.538 | 0.584 | 0.582 | 0.447 | 0.492 | 0.438 | 0.440 | 0.678 | 0.685 |
| | **EGINTERPOLATOR** | **0.307** | **0.293** | **0.214** | **0.194** | **0.361** | **0.385** | **0.416** | **0.409** | **0.633** | **0.639** |
| | GEOTDM | 0.757 | 0.757 | 0.782 | 0.793 | 0.488 | 0.533 | 0.454 | 0.453 | 0.697 | 0.703 |
| **Enlarged $\rightarrow$ Standard** | EGINTERPOLATOR-N | 0.470 | 0.460 | 0.540 | 0.544 | 0.433 | 0.481 | 0.443 | 0.440 | 0.681 | 0.691 |
| | **EGINTERPOLATOR** | **0.296** | **0.286** | **0.261** | **0.247** | **0.370** | **0.388** | **0.405** | **0.394** | **0.636** | **0.638** |

### A.8.3 CONTRIBUTION OF AN EXTENDED TRAIN SET

While our framework is motivated by the scarcity of trajectory data, we also evaluate model performance under increased supervision. We train on an enlarged dataset—4× larger than the original—comprising 4437 molecules, with the original split from Section 5.2 as a subset. As shown in Table 10, while EGINTERPOLATOR-**N** and EGINTERPOLATOR interestingly do not improve substantially with more data, the latter maintains a clear advantage. This highlights the continued value of structural pretraining even in higher-data regimes.

### A.8.4 CONTRIBUTIONS OF THE TEMPORAL MODULE TO NON-TRIVIAL DYNAMICS

To assess the contribution of our temporal module in learning non-trivial dynamics—specifically the fast torsional processes observed in organic small molecules—we compare our framework run with and without the temporal component. We generate trajectories for both QM9 and Drugs with $\alpha = 1$ (i.i.d. conformers, i.e., no temporal interpolation). Additionally, we shuffle the frames of both GT trajectories and our original model generations to establish baselines corresponding to random frame orderings. We then computed torsional decorrelation times for all conditions. While our method does not fully match GT torsional decorrelation times on QM9, we see that it clearly avoids the trivial 5.2 ps baseline (the MD frame rate). This supports that the temporal module learns non-trivial dynamical properties essential for modeling diverse molecule dynamics.

### A.8.5 CONTRIBUTION OF THE TEMPORAL MODULE TO STRUCTURE LEARNING

To asses if our temporal module's spatial update layers refine structural predictions during trajectory generation, we compare the collective variable JSD distributions between the normal and $\alpha = 1$ setting. As demonstrated in Table 11, the full interpolator improves bond lengths, bond angles, and torsions, indicating that the system can correct imperfections in the conformer prior rather than inherit them.

### A.9 $\alpha$ MIXING PARAMETERS: INTERPRETATION & CONTRIBUTION

### A.9.1 EMPIRICALLY LEARNED VALUES

We analyze the ranges of alpha values learned during training and in order to identify consistent patterns and interpretable behaviors in Figure 9 and Figures 16, 17. As context: (1) Positive alpha values assign greater weight to the pretrained spatial model, while negative values emphasize the

Table 12: Mean torsional decorrelation times (ps) across test sets, comparing GT MD data, our original generations, i.i.d. conformer generations ($\alpha = 1$), and shuffled variants. Shuffled data collapse to the frame rate of 5.2 ps, reflecting a lack of temporal structure.

| Dataset | GT MD | Original Gen. | $\alpha = 1$ Gen. | Shuffled GT | Shuffled Gen. |
|---|---|---|---|---|---|
| QM9 Test | 101.0 | 13.59 | 5.2 | 5.2 | 5.2 |
| Drugs Test | 130.1 | 185.64 | 5.2 | 5.2 | 5.2 |

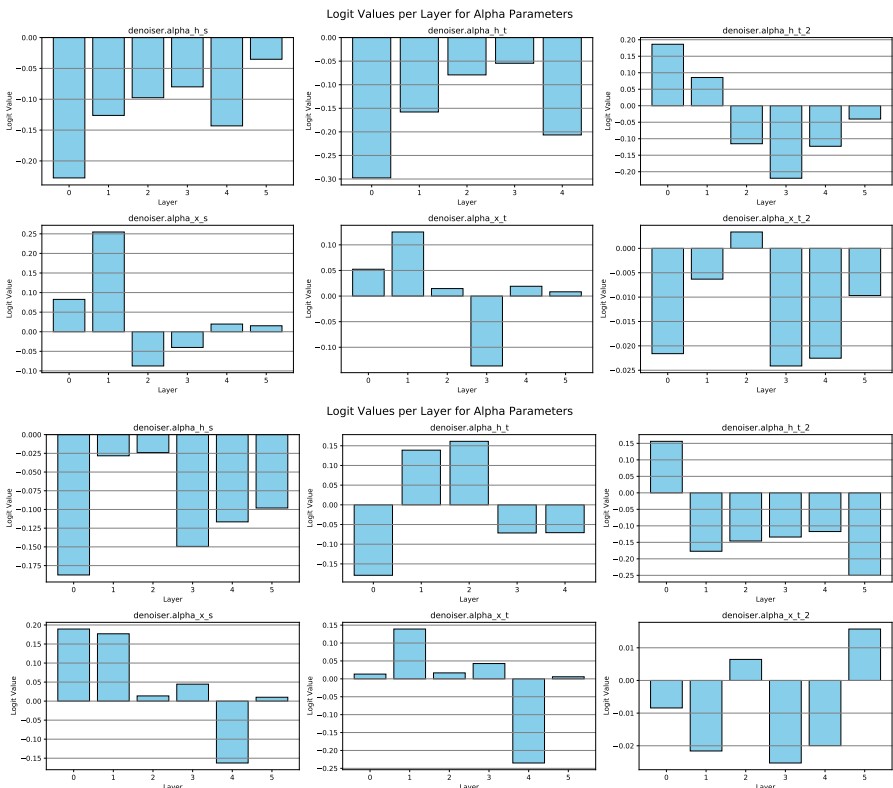

Figure 9: **Top:** Logits of $\alpha$ for each spatial and temporal layer after convergence on the QM9 unconditional generation task. **Bottom:** Logits of $\alpha$ for each spatial and temporal layer after convergence on the DRUGS forward simulation task. **Both:** Results obtained with EGINTERPOLATOR-CASC.

temporal component; (2) alpha_h/x_s correspond to the pretrained spatial layer and the spatial layer in the temporal module, where h and x denote mixing coefficients for invariant and vector features, respectively; (3) Layer 5 does not include an alpha_h_t term, as this output is never used.

Overall, alpha values generally fall within $[-0.25, 0.25]$. From Figure 9, we observe some exciting trends: in the first temporal block (alpha_x_t) and spatial block (alpha_x_s) of the temporal module, earlier layers prefer pretrained information, while later layers favor temporal module information. For the final temporal block (alpha_x_t_2), the model generally relies on newly trained information across layers. This supports our design choices: early layers focus on structural integrity, while later layers prioritize dynamics, with the last temporal block reinforcing dynamic updates.

### A.9.2 CONTRIBUTION TO CONFORMER GENERATION

Although the endpoints $\alpha = 0$ and $\alpha = 1$ yield straightforward and well-defined inference dynamics, we also investigate the inference-time flexibility of this parameter by running EGINTERPOLATOR as a conformer generator on QM9 while perturbing $\alpha$. Specifically, we linearly interpolate the mixing

Table 13: QM9 results across $\lambda$ for CASC and SIMPLE.

| $\lambda$ | COV-R (%) ↑ | | MAT-R (Å) ↓ | | COV-P (%) ↑ | | MAT-P (Å) ↓ | |
|---|---|---|---|---|---|---|---|---|
| | Mean | Med. | Mean | Med. | Mean | Med. | Mean | Med. |
| | | | | CASC | | | | |
| 0.000 | 87.99 | 91.98 | 0.2539 | 0.2600 | 58.30 | 53.62 | **0.4430** | **0.4397** |
| 0.025 | 87.82 | 92.98 | 0.2570 | 0.2598 | 57.70 | 53.17 | 0.4470 | 0.4396 |
| 0.050 | 88.34 | 92.84 | 0.2568 | 0.2556 | 58.29 | 53.78 | 0.4460 | 0.4439 |
| 0.075 | 88.16 | 93.61 | 0.2577 | 0.2610 | 57.38 | 52.86 | 0.4490 | 0.4467 |
| 0.100 | 88.47 | 92.66 | 0.2588 | 0.2654 | 57.14 | 52.51 | 0.4531 | 0.4523 |
| 0.125 | 89.09 | 94.36 | 0.2579 | 0.2589 | 56.67 | 52.21 | 0.4581 | 0.4549 |
| 0.150 | 89.55 | 93.39 | 0.2580 | 0.2637 | 56.37 | 50.83 | 0.4618 | 0.4580 |
| 0.175 | 89.06 | 94.46 | 0.2621 | 0.2612 | 55.88 | 51.06 | 0.4669 | 0.4670 |
| 0.200 | **89.27** | **94.56** | 0.2633 | 0.2604 | 55.23 | 50.95 | 0.4697 | 0.4653 |
| | | | | SIMPLE | | | | |
| 0.000 | 88.11 | 92.47 | 0.2557 | 0.2553 | 59.03 | 54.52 | **0.4413** | **0.4439** |
| 0.025 | 88.54 | 91.28 | 0.2546 | 0.2540 | 58.27 | 53.72 | 0.4472 | 0.4419 |
| 0.050 | 89.71 | 94.13 | 0.2518 | 0.2577 | 58.20 | 54.24 | 0.4492 | 0.4397 |
| 0.075 | 90.34 | 94.20 | 0.2536 | 0.2589 | 57.79 | 53.51 | 0.4539 | 0.4496 |
| 0.100 | 91.11 | 95.36 | 0.2542 | 0.2589 | 57.14 | 52.23 | 0.4598 | 0.4542 |
| 0.125 | 92.11 | 96.36 | 0.2558 | 0.2638 | 57.19 | 53.04 | 0.4647 | 0.4582 |
| 0.150 | 92.05 | 96.07 | 0.2618 | 0.2665 | 57.14 | 54.30 | 0.4681 | 0.4630 |
| 0.175 | 92.21 | 96.39 | 0.2678 | 0.2737 | 55.58 | 51.75 | 0.4795 | 0.4675 |
| 0.200 | **92.63** | **96.08** | 0.2713 | 0.2783 | 54.94 | 50.63 | 0.4885 | 0.4850 |
| | | | | GeoDiff-A | | | | |
| – | 90.54 | 94.61 | 0.2104 | 0.2021 | 52.35 | 50.10 | 0.4539 | 0.4399 |

parameter logits between 1 and the learned value $\alpha^\star$ by introducing a new variable $\lambda \in [0, 1]$, such that $\alpha' = \lambda\alpha + (1 - \lambda)$.

Across both the SIMPLE and CASC variants, we observe a trade-off between precision and diversity metrics as summarized in Table 13. Notably, varying $\lambda$ allows us to recover and surpass the COV-R diversity scores reported by GeoDiff. The SIMPLE variant exhibits a more favorable precision–diversity trade-off curve with respect to $\lambda$, which we attribute to its closer alignment with our theoretical formulation in Theorem 4.1. More broadly, these findings indicate that the temporal module captures aspects of conformational diversity beyond those provided by the pretrained conformer model, and that the $\alpha$ parameters offer a natural mechanism for controlling the balance between precision and conformational dynamics in the generated trajectories.

# B  EXPERIMENTAL DETAILS

## B.1  CONFORMER PRETRAINING

### B.1.1  DATA PREPROCESSING

The datasets obtained from the (Xu et al., 2022; Shi et al., 2021a) codebase are provided as pickle files, each containing a list of PyTorch Geometric data objects representing individual conformers. We apply the following filtering steps to ensure data quality. First, we verify that the saved `RDMol` objects can be successfully sanitized using RDKit. Next, we remove any conformers exhibiting fragmentation in their `RDMol` representations. Following Ganea et al. (2021), we also account for conformers that may have reacted in the original data generation process. Namely, we compare the canonical SMILES strings derived from both the saved SMILES and the corresponding `RDMol`, and discard any conformers where the two do not match. We also exclude any molecules whose saved SMILES cannot be converted into a valid `RDMol` by RDKit. Lastly, specific to our method, we

remove hydrogens from the molecules according to `rdkit.Chem.RemoveHs`[2] and retain heavy atoms. For QM9, this leaves [*C, N, O, F*]. For Drugs, we have [*C, N, O, S, P, F, Cl, Br, I, B, Si*].

For each peptide in the Timewarp and MDen train/validation sets (Klein et al., 2023) (Jing et al., 2024b), we compute per-residue $\phi/\psi$ dihedral features, subsample up to 10,000 frames, cluster them in dihedral space using K-medoids, and select the lowest-energy member of each cluster as a representative conformer, yielding 10 / 20 conformers per peptide.

### B.1.2 TRAINING DETAILS

We train both the QM9, Drugs, and Tetrapeptide conformer models using 4 NVIDIA RTX A4000 GPUs, with an effective batch size of 128 (32 samples per GPU) and a learning rate of $1 \times 10^{-4}$. Training is carried out until convergence, typically around 800K steps. As described in Section 5.1, all models are trained using 1000 diffusion steps. We adopt a DDPM framework (Ho et al., 2020b) with a linear noise schedule. Additionally, we employ an equivariant loss function that leverages optimal Kabsch alignment (Kabsch, 1976), with more details in Section C.4.

### B.1.3 EVALUATION DETAILS

We evaluate the quality of generated conformers using Coverage (COV-P) and Matching (MAT-P), both based on the root mean square deviation (RMSD) computed after Kabsch alignment (Kabsch, 1976).

Let $S_g$ and $S_r$ denote the sets of generated and reference conformers, respectively. The metrics are defined as:

$$\text{COV-P}(S_g, S_r) = \frac{1}{|S_g|} \left| \left\{ \hat{C} \in S_g \,\middle|\, \min_{C \in S_r} \text{RMSD}(\hat{C}, C) \leq \delta \right\} \right|, \tag{6}$$

$$\text{MAT-P}(S_g, S_r) = \frac{1}{|S_g|} \sum_{\hat{C} \in S_g} \min_{C \in S_r} \text{RMSD}(\hat{C}, C), \tag{7}$$

where $\delta$ is a predefined threshold. COV-R and MAT-R, inspired by *Recall*, are defined analogously by swapping $S_g$ and $S_r$.

Following Xu et al. (2022), we set $|S_g| = 2 \times |S_r|$ per molecule. The results reported in Section 5.1 correspond to the average COV-*/MAT-* scores across all test molecules. COV-P reflects precision by measuring the fraction of generated conformers that are sufficiently close to the reference set (within threshold $\delta$), while MAT-P captures the mean deviation of each generated conformer from its closest reference match. High COV and low MAT scores indicate greater fidelity and precision in conformer generation.

### B.2 MOLECULAR DYNAMICS FOR SMALL MOLECULES

### B.2.1 PARAMETERIZATION

We run all-atom molecular dynamics simulations, including hydrogens, using OpenMM (Eastman et al., 2017) and employ `openmmforcefields` to apply small molecule force field parameterizations developed by the Open Force Field Initiative (OpenFF) (Boothroyd et al., 2023). We follow the setup guidelines provided in the `openmmforcefields` GitHub repository. Specifically, we adopt the `openff-2.2.1` (Sage) (McIsaac et al., 2024) small molecule force field in conjunction with a base `amber/protein.ff14SB.xml` protein force field and a combination of `amber/tip3p_standard.xml` and `amber/tip3p_HFE_multivalent.xml` for explicit solvent and ion parameters. Continuing with standard hyperparameters, we set the nonbonded cutoff to 0.9 nm and the switch distance to 0.8 nm. Hydrogen mass repartitioning (HMR) is applied with a mass of 1.5 amu, along with constraints on all hydrogen bonds. Long-range electrostatic interactions are computed using the Particle Mesh Ewald (PME) method under periodic boundary conditions. A padding of 1.5 nm is used for the explicit solvent box.

---

[2]Note that `RemoveHs` does not eliminate all hydrogen atoms and may retain chemically relevant ones (see the RDKit documentation). Our method explicitly incorporates and models such retained hydrogens.

### B.2.2 SIMULATION

All molecular dynamics simulations are performed using a friction coefficient of $1 \, \text{ps}^{-1}$, a temperature of 300 K, and an integration timestep of 4 fs, employing the `LangevinMiddleIntegrator` (Zhang et al., 2019). As described in Section 5.2, five independent trajectories are generated per molecule, each initialized from a conformer assigned to that molecule in the selected data subset. Each trajectory simulation begins with energy minimization, followed by 5000 steps of equilibration under constant volume and temperature (NVT) conditions. This is followed by a 5 ns production run under constant pressure and temperature (NPT) conditions, comprising a total of 1.25M integration steps. Trajectory simulation is parallelized across 32 NVIDIA RTX A4000 GPUs and saved with a frame rate of 400 fs/0.4 ps.

### B.3 TRAJECTORY FINETUNING

### B.3.1 DATASET PREPARATION

As mentioned in Section 5.2, we randomly sample a subset of the molecules from the GEOM-QM9 and Drugs conformer data to generate trajectory data from. As this is quite costly, for Drugs we generate simulations for the standard train/validation/test splits mentioned in Section 5.2. For QM9, we generate data for enlarged train/test sets along with the standard validation set. We then subsample 25% of the enlarged splits to be the standard train/test sets. A summary of the dataset splits is provided below:

- **Drugs:**
    - *Standard splits:* `1137/1044/100` train/validation/test molecules (`5682/5209/496` associated trajectories)
- **QM9:**
    - *Standard splits:* `1109/1018/240` train/validation/test molecules (`5534/5080/1193` associated trajectories)
    - *Enlarged sets:* `4437/959` train/test molecules (`22132/4793` associated trajectories)

As a note, out of the test trajectories, we select 1 out of 5 per molecule to be the MD ORACLE baseline. Moreover, we filter out any molecules over 60 atoms in the Drugs dataset to reduce memory usage variance. Finally, the test set for the interpolation is a subset of the standard test sets mentioned above. We further define this process of selection in Section B.6 and B.3.3.

### B.3.2 TRAINING PROTOCOL

While the compute setup and batch size vary across datasets and generation settings, we consistently employ a DDPM framework with a linear noise schedule and train all models using 1000 diffusion steps. A fixed learning rate of $1 \times 10^{-4}$ is used and training is performed until convergence. Additionally, we adopt an equivariant loss function based on optimal global Kabsch alignment of trajectories, as detailed in Section C.4. Setting-specific training configurations are provided in Sections B.4-B.6.

### B.3.3 EVALUATION METRICS

**Jensen-Shannon Divergence.** We compute the JSD as implemented in `scipy`, where $m = (p + q)/2$:

$$\sqrt{\frac{D(p \parallel m) + D(q \parallel m)}{2}} \tag{8}$$

- Torsions: The 1D JSD is computed over a 100-bin histogram discretized across $[-\pi, \pi]$.
- Bond Angles: The 1D JSD is computed over a 100-bin histogram discretized across $[0, \pi]$.
- Bond Lengths: The 1D JSD is computed over a 100-bin histogram discretized across $[100, 220]$ pm.

- Torsion decorrelation: The 1D JSD is computed over 275-bin histogram discretized across $[5, 1380]$ ps, which are corresponding to the minimum and maximum torsion decorrelation time of molecules across the dataset.

- TICA-0 and TICA-0,1: We reduce the dimensionality of the trajectory by time-lagged independent component analysis (TICA). Then 1D, 2D JSDs are computed over 100-bin histograms on the first TICA component (TICA-0) and the first two components (TICA-0,1), respectively. Since different molecules have totally different TICA projections and values, we use the minimum and maximum values from each molecule as its unique discretization range for TICA-0 and TICA-0,1. We use 10.4 ps (2 steps) lag time for QM9 and 20.8 ps (4 steps) for drugs.

**Markov State Models.** We intensively use Markov State Models (MSM) for interpolation tasks. We featurize reference trajectories with all torsion angles except for those within an aromatic ring. Then TICA is performed on the torsion-based trajectories. After dimensionality reduction, a k-means clustering algorithm is used to discretize the trajectories to 100 clusters. An MSM analysis is performed on the trajectories of 100 states and PCCA+ spectral clustering from PyEMMA package (Scherer et al., 2015) is used to aggregate clusters to 10 coarse metastates. A second MSM analysis is done on the coarse trajectories. We use 52 ps (10 steps) lag time for QM9 and 104 ps (20 steps) for drugs.

To sample the start and end frames used in the interpolation task, we compute the flux matrix over the 10 metastates. To construct a high barrier and rare transition probability, we choose the two states with least flux between them as start and end states. Then we randomly sample 900 start and end frames from the corresponding states, and those frames are used as the conditions in the interpolation inference process. The generated trajectories undergo the same featurization process, and then projected on the TICA components defined by the reference trajectories. They are further discretized according to the reference metastate assignments, and a new MSM is performed on the discretized generation trajectories.

To compare the generation with reference trajectories, we compute the JSD over the metastate occupancy probabilites. To evaluate interpolation sampling quality, we compute the average path probability, valid path rate, and valid path probability as described in Jing et al. (2024c). The average path probability is the average of all paths' likelihood for transitioning from the start to the end. The valid path rate is the fraction of paths that successfully traverse from the start to the end. The valid path probability is the average of all valid paths' likelihood (excluding zero-probability paths). To fairly compare the generation and MD oracle, we truncate the MD oracle trajectories to varying time length, and sample 900 transition paths based on the MSM constructed from the metastates. With the sampled transition paths, we can compute the JSD over metastates, average path probability, valid path rate, and valid path probability of MD oracles.

### B.4 Unconditional Generation Details

**Training.** Training is conducted by denoising randomly sampled 2.6 ns segments (500 frames) from the training trajectories. For QM9, we utilize 8 NVIDIA RTX A4000 GPUs with an effective batch size of 32 (4 samples per GPU), training the models for 400 epochs.

**Evaluation.** For each molecule in the test set, we generate ten independent 2.6 ns segments (500 frames each). Distributional histograms are then computed from these generated trajectories and compared against those derived from four reference 5 ns molecular dynamics (MD) trajectories. Results reported for this model setting for QM9 include both the standard test in Section 5.3 and enlarged test set in Section A.3.2-A.3.3.

### B.5 Forward Simulation Details

**Training.** Training is conducted by randomly sampling 251-frame segments at a 5.2 ps frame rate and denoising the subsequent 250 frames (corresponding to 1.3 ns), conditioned on the initial frame-0. For the Drugs and Timewarp dataset, we utilize 8 NVIDIA RTX A4000 GPUs with an effective batch size of 32 (2 samples per GPU with 2 gradient accumulation steps), training the models for 400 epochs.

**Evaluation.** For each molecule in the test set, we generate five forward roll-outs of 5.2 ns (1,000 frames total), each conditioned on the first frame of a reference trajectory. Distributional histograms are then computed from the generated trajectories and compared against those obtained from four reference 5 ns molecular dynamics (MD) trajectories. For a fair comparison, we truncate our generation trajectories to the same length as the reference trajectories in evaluation. Results reported for this model setting for Drugs are based on the standard test set in Section 5.4.

### B.6 INTERPOLATION DETAILS

**Training.** Training is conducted by randomly sampling 101-frame segments at a 5.2 ps frame rate and denoising the middle 99 frames (corresponding to $\approx$0.52 ns), conditioned on frame-0 and frame-100. For the QM9 dataset, we utilize 2 NVIDIA A100 GPUs with an effective batch size of 128 (64 samples per GPU), training the models for 300 epochs. For the Drugs dataset, we utilize 4 NVIDIA A100 GPUs with an effective batch size of 32 (8 samples per GPU), training the models for 400 epochs.

**Evaluation.** For each molecule in the test set, we perform featurization, dimensionality reduction, and clustering on the reference trajectories. We then construct an MSM on the discretized trajectories and retain only those test molecules for which all microstates from clustering are represented in the MSM. After filtering, this yields 124 QM9 and 36 Drug test molecules. Due to computational constraints, we subsample 80 QM9 molecules while using all 36 Drug molecules for inference and evaluation. For each selected test molecule, we generate 900 interpolation trajectories conditioned on 900 sampled start and end states. For each MD oracle length, we also sample 900 transition paths. We report the average results across all molecules successfully modeled by the MSM, as shown in Section 5.5, Figure 5, as well as Section A.2, Figure 6 (see details in Section B.3.3).

## C  METHOD DETAILS

### C.1  MOLECULE INPUT REPRESENTATION

Throughout our framework, input molecules are represented as 2D heterogeneous graphs. The bonding network includes both the original bond types present in the molecule and additional higher-order edges that we incorporate. Specifically, we include edges up to third-order for both the QM9 and Drug datasets. Following the approach of Shi et al. (2021b), this augmentation is designed to facilitate more effective information transfer between atoms involved in bond angle and torsion angle interactions.

Table 14: Atom and bond embedding specifications.

| Embedding Type | Input | Dimension |
|---|:---:|---|
| Atom Embedding | Atomic Number | 30 |
| Bond Embedding | No Bond, Bond Type, 2nd/3rd-order edge | 4 |

We defined learned embeddings for atom type as well as bond type. Moreover, we also provide input node features per atom, largely based on Ganea et al. (2021). Below, we provide a table with these details. These two information sources, the learned embedding and input features, as combined in our embedding module as described in Section C.2.

### C.2  ARCHITECTURES

**Embeddings.** Across all of our models—both conformer and trajectory—we use a hidden dimension of 128 and a diffusion timestep embedding dimension of 32. For molecular embeddings, we combine atom type embeddings and atom-level features via a single linear projection: $\mathbb{R}^{\texttt{node\_dim+ft\_dim}} \rightarrow \mathbb{R}^{\texttt{node\_dim}}$.

**BASICES.** As introduced in Section 4.3, our BASICES architecture consists of 6 Equivariant Graph Convolution (EGCL) layers, following the formulation in Satorras et al. (2021b). To promote

Table 15: Node feature vector based on atom-level properties.

| | Atom Features | | |
|---|---|---|---|
| **Indices** | **Description** | **Options** | **Type** |
| 0–1 | Aromaticity | true, false | One-hot |
| 2–7 | Hybridization | $sp, sp^2, sp^3, sp^3d, sp^3d^2$, other | One-hot |
| 8 | Partial charge | $\mathbb{R}$ | Value |
| 9–16 | Implicit valence | 0, 1, 2, 3, 4, 5, 6, other | One-hot |
| 17–24 | Degree | 0, 1, 2, 3, 4, 5, 6, other | One-hot |
| 25–28 | Formal charge | -1, 0, 1, other | One-hot |
| 29–35 | In ring of size $x$ | 3, 4, 5, 6, 7, 8, other | k-hot |
| 36–39 | Number of rings | 0, 1, 2, 3+ | One-hot |
| 40–42 | Chirality | CHI_TETRAHEDRAL_CW, CHI_TETRAHEDRAL_CCW, unspecified/other | One-hot |

interaction between invariant and equivariant representations, we insert a Geometric Vector Perceptron (GVP) (Jing et al., 2021) transition layer after each EGCL block. The full model contains approximately 918K parameters.

**EGINTERPOLATOR.** As described in Section 4.3, EGINTERPOLATOR extends BASICES by introducing temporal attention to model dependencies across trajectory frames. Specifically, we incorporate the Equivariant Temporal Attention Layer (ETLayer) from Han et al. (2024) to capture temporal structure through attention mechanisms. The architecture is constructed by stacking an additional sequence of ETLayer + EGCL + ETLayer on top of each pretrained EGCL layer from BASICES, as illustrated in Figure 2. We retain the use of GVP-based transition layers and introduce LayerNorm (Ba et al., 2016) at key interpolation steps to improve numerical stability. The resulting model comprises 6 layers and contains 3.3M parameters in total, with 2.3M trained during trajectory finetuning in the EGINTERPOLATOR framework.

## C.3 CONDITIONAL GENERATION

We control conditional generation by setting appropriate entries of a conditioning mask $\mathbf{m}$ to either 1 or 0. Let $\mathbf{m}[t, a]$ denote the conditioning status for frame $t$ and atom $a$. We define mask:

- Forward simulation:

$$\mathbf{m}[t, :] = \begin{cases} 1 & t = 0 \\ 0 & \text{otherwise} \end{cases}$$

- Interpolation:

$$\mathbf{m}[t, :] = \begin{cases} 1 & t \in \{0, M\} \ (M \text{ is index of the final frame}) \\ 0 & \text{otherwise} \end{cases} .$$

In the unconditional setting, we default to $\mathbf{m}[:, :] = 0$. To incorporate this conditioning information, we use a condition state embedding added to the invariant node features, with the same hidden dimension as the main model. The conditioning mask is also used to restrict the denoising process and loss computation to frames where $\mathbf{m}[t', :] = 0$.

## C.4 KABSCH ALIGNMENT

Inspired by Xu et al. (2022), we propose to use trajectory-level Kabsch alignment to find the optimal rotation and translation between the noisy trajectory $\mathbf{x}_\tau^{[T]}$ and the input trajectory $\mathbf{x}_0^{[T]}$ at diffusion step $\tau$. This corresponds to the following optimization problem:

$$\mathbf{R}^*, \mathbf{t}^* = \underset{\mathbf{R}, \mathbf{t}}{\arg \min} \|\mathbf{R}\mathbf{x}_\tau^{[T]} + \mathbf{t} - \mathbf{x}_0^{[T]}\|_2. \tag{9}$$

In practice, this can be realized by extending the original Kabsch algorithm (Kabsch, 1976) on the set of points with the temporal dimension $T$ combined into the number of points dimension $N$, that forms a point cloud with effective number of points $T \times N$. Afterwards, we re-compute the target noise $\bar{\epsilon}$ based on the aligned $\bar{\mathbf{x}}_\tau^{[T]} = \mathbf{R}^* \mathbf{x}_\tau^{[T]} + \mathbf{t}^*$ and the clean data $\mathbf{x}_0^{[T]}$ by the forward diffusion process, and then match the output of EGINTERPOLATOR towards re-computed noise $\bar{\epsilon}$ after alignment.

### C.5 BASELINES

**Autoregressive Models.** In the autoregressive baseline setup, molecular dynamics trajectories are modeled under the Markov assumption, where the model—EGNN (Satorras et al., 2021b), Equivariant Transformer (Thölke & Fabritiis, 2022), or GeoTDM (Han et al., 2024)—learns the transition distribution $p(x_{t+1}, |, x_t)$. To ensure fair comparison, we keep timestep intervals and frame counts consistent across all datasets during both training and inference, matching the settings used in our proposed methods. For EGNN and ET, we adopt identical configurations with six stacked EGCL or Equivariant Transformer blocks, respectively, to maintain experimental parity. For AR+GeoTDM, the model is trained as a two-frame diffusion process, with the first frame serving as conditioning, effectively reducing it to a next-step forward simulation model.

**GEOTDM.** The training setup and embedding configurations for our implementation of GEOTDM are aligned with those used in our proposed framework. Following the architecture described in Han et al. (2024), the model consists of 6 stacked layers of EGCL and ETLayer blocks, resulting in a total of 1.4M parameters.

## D PROOFS

### D.1 PROOF OF THEOREM 4.1

For better readability we restate Theorem 4.1 below.

**Theorem 4.1.** *Suppose $\epsilon_\theta^{\mathrm{cf}}$ perfectly models $p^{\mathrm{cf}}(\mathbf{x})$ and $\epsilon_{\theta,\phi}^{\mathrm{md}}$ perfectly models $p^{\mathrm{md}}(\mathbf{x}^{[T]})$, then the interpolation in Eq. 3 implicitly induces the distribution $\tilde{p}^{\mathrm{md}}(\mathbf{x}^{[T]}) \propto p^{\mathrm{md}}(\mathbf{x}^{[T]})^\beta \hat{p}^{\mathrm{md}}(\mathbf{x}^{[T]})^{1-\beta}$ for $\epsilon_\phi$, where $\beta = \frac{1}{1-\alpha}$ and $\hat{p}^{\mathrm{md}} = \prod_{t=0}^{T-1} p^{\mathrm{cf}}(\mathbf{x}^{(t)})$.*

*Proof.* Upon perfect optimization, we have the connection between the denoiser and the score of the underlying distribution (Song & Ermon, 2019; Song et al., 2021):

$$\epsilon_\theta^{\mathrm{cf}}(\mathbf{x}_\tau^{(t)}, \tau) = -\sqrt{1-\bar{\alpha}_\tau}\nabla \log p^{\mathrm{cf}}(\mathbf{x}^{(t)}), \quad \forall 0 \leq t \leq T-1, 0 \leq \tau \leq \mathcal{T}, \tag{10}$$

and similarly,

$$\epsilon_{\theta,\phi}^{\mathrm{md}}(\mathbf{x}_\tau^{[T]}, \tau) = -\sqrt{1-\bar{\alpha}_\tau}\nabla \log p^{\mathrm{md}}(\mathbf{x}^{[T]}), \quad \forall 0 \leq \tau \leq \mathcal{T}. \tag{11}$$

By leveraging Eq 10 for all frames $0 \leq t \leq T-1$, we have

$$\hat{\epsilon}^{\mathrm{md}} = [\epsilon_\theta^{\mathrm{cf}}(\mathbf{x}_\tau^{(t)}, \tau)]_{t=0}^{T-1} = -\sqrt{1-\bar{\alpha}_\tau}\nabla \log \hat{p}^{\mathrm{md}}(\mathbf{x}^{[T]}), \tag{12}$$

where $\hat{p}^{\mathrm{md}}(\mathbf{x}^{[T]})$ is the joint of i.i.d. framewise distributions $p(\mathbf{x})$. Combining with the interpolation rule in Eq. 3, we have

$$\epsilon_\phi = \frac{1}{1-\alpha}\epsilon_{\theta,\phi}^{\mathrm{md}} - \frac{\alpha}{1-\alpha}\hat{\epsilon}^{\mathrm{md}}, \tag{13}$$

$$= (-\sqrt{1-\bar{\alpha}_\tau})\left(\frac{1}{1-\alpha}\nabla \log p^{\mathrm{md}}(\mathbf{x}^{[T]}) - \frac{\alpha}{1-\alpha}\nabla \log \hat{p}^{\mathrm{md}}(\mathbf{x}^{[T]})\right), \tag{14}$$

$$= (-\sqrt{1-\bar{\alpha}_\tau})\left(\beta\nabla \log p^{\mathrm{md}}(\mathbf{x}^{[T]}) + (1-\beta)\nabla \log \hat{p}^{\mathrm{md}}(\mathbf{x}^{[T]})\right), \tag{15}$$

where $\beta = \frac{1}{1-\alpha}$. Now, consider the distribution $\tilde{p}^{\mathrm{md}}(\mathbf{x}^{[T]}) \propto p^{\mathrm{md}}(\mathbf{x}^{[T]})^\beta \hat{p}^{\mathrm{md}}(\mathbf{x}^{[T]})^{1-\beta}$, we have

$$\nabla \log \tilde{p}^{\mathrm{md}}(\mathbf{x}^{[T]}) = \beta\nabla \log p^{\mathrm{md}}(\mathbf{x}^{[T]}) + (1-\beta)\nabla \log \hat{p}^{\mathrm{md}}(\mathbf{x}^{[T]}). \tag{16}$$

Therefore, $\boldsymbol{\epsilon}_\phi = -\sqrt{1 - \bar{\alpha}_\tau}\nabla \log \tilde{p}^{\mathrm{md}}(\mathbf{x}^{[T]})$. This verifies that the interpolation rule implicitly induces the distribution $\tilde{p}^{\mathrm{md}}(\mathbf{x}^{[T]})$ with $\boldsymbol{\epsilon}_\phi$ as its score network. Furthermore, the induction is unique, since for any distribution $q(\mathbf{x}^{[T]})$ satisfying $\boldsymbol{\epsilon}_\phi = -\sqrt{1 - \bar{\alpha}_\tau}\nabla \log q(\mathbf{x}^{[T]})$, we have that $\nabla \log \tilde{p}^{\mathrm{md}}(\mathbf{x}^{[T]}) = \nabla \log q(\mathbf{x}^{[T]})$, which gives us $q(\mathbf{x}^{[T]}) = \tilde{p}(\mathbf{x}^{[T]})$ due to the property of Stein score as demonstrated in Hyvärinen & Dayan (2005); Song & Ermon (2019).

$\square$

## D.2 Proof of Equivariance

**Theorem D.2.** EGInterpolator *is SO(3)-equivariant and translation-invariant. Namely,* $\mathbf{R}f_{\mathrm{EGI}}(\mathbf{x}^{[T]}) = f_{\mathrm{EGI}}(\mathbf{R}\mathbf{x}^{[T]} + \mathbf{t})$*, for all rotations* $\mathbf{R}$ *and translations* $\mathbf{t}$ *where* $f_{\mathrm{EGI}}$ *is the mapping defined per* EGInterpolator.

*Proof.* Recall the definition of the interpolator:

$$\boldsymbol{\epsilon}^{\mathrm{md}}_{\theta,\phi}(\mathbf{x}^{[T]}_\tau, \tau) = \alpha\hat{\boldsymbol{\epsilon}}^{\mathrm{md}} + (1 - \alpha)\boldsymbol{\epsilon}^{\mathrm{tp}}_\phi(\mathbf{x}^{[T]}_\tau, \hat{\boldsymbol{\epsilon}}^{\mathrm{md}}, \tau), \qquad \text{s.t.} \quad \hat{\boldsymbol{\epsilon}}^{\mathrm{md}} = [\boldsymbol{\epsilon}^{\mathrm{cf}}_\theta(\mathbf{x}^{(t)}_\tau, \tau)]^{T-1}_{t=0}, \qquad (17)$$

with the parameterization $\boldsymbol{\epsilon}^{\mathrm{tp}}_\phi(\mathbf{x}^{[T]}_\tau, \hat{\boldsymbol{\epsilon}}^{\mathrm{md}}, \tau) = \mathbf{s}^{\mathrm{tp}}_\phi(\mathbf{x}^{[T]}_\tau + \hat{\boldsymbol{\epsilon}}^{\mathrm{md}}, \tau) - \mathbf{x}^{[T]}_\tau$. It suffices to show that the temporal interpolator is rotation-equivariant and translation-invariant, since the equivariance of the structure model $\boldsymbol{\epsilon}^{\mathrm{cf}}_\theta$ directly follows the original work of Satorras et al. (2021b). For any $g \coloneqq (\mathbf{R}, \mathbf{t}) \in \mathrm{SE}(3)$, we have $[\boldsymbol{\epsilon}^{\mathrm{cf}}_\theta(\mathbf{R}\mathbf{x}^{(t)}_\tau + \mathbf{t}, \tau)]^{T-1}_{t=0} = \mathbf{R}[\boldsymbol{\epsilon}^{\mathrm{cf}}_\theta(\mathbf{x}^{(t)}_\tau, \tau)]^{T-1}_{t=0} = \mathbf{R}\hat{\boldsymbol{\epsilon}}^{\mathrm{md}}$. By the proof in Han et al. (2024), we have that the temporal network $\mathbf{s}^{\mathrm{tp}}_\phi$ is SE(3)-equivariant, *i.e.*,

$$\mathbf{s}^{\mathrm{tp}}_\phi(\mathbf{R}(\mathbf{x}^{[T]}_\tau + \hat{\boldsymbol{\epsilon}}^{\mathrm{md}}) + \mathbf{t}, \tau) = \mathbf{R}\mathbf{s}^{\mathrm{tp}}_\phi(\mathbf{x}^{[T]}_\tau + \hat{\boldsymbol{\epsilon}}^{\mathrm{md}}, \tau) + \mathbf{t}. \qquad (18)$$

Therefore, we have

$$\boldsymbol{\epsilon}^{\mathrm{md}}_{\theta,\phi}(\mathbf{R}\mathbf{x}^{[T]}_\tau + \mathbf{t}, \tau) = \alpha\mathbf{R}\hat{\boldsymbol{\epsilon}}^{\mathrm{md}} + (1 - \alpha)\left(\mathbf{s}^{\mathrm{tp}}_\phi(\mathbf{R}(\mathbf{x}^{[T]}_\tau + \hat{\boldsymbol{\epsilon}}^{\mathrm{md}}) + \mathbf{t}, \tau) - \mathbf{R}\mathbf{x}^T_\tau - \mathbf{t}\right), \qquad (19)$$

$$= \alpha\mathbf{R}\hat{\boldsymbol{\epsilon}}^{\mathrm{md}} + (1 - \alpha)\mathbf{R}\boldsymbol{\epsilon}^{\mathrm{tp}}_\phi(\mathbf{x}^{[T]}_\tau, \hat{\boldsymbol{\epsilon}}^{\mathrm{md}}, \tau), \qquad (20)$$

$$= \mathbf{R}\boldsymbol{\epsilon}^{\mathrm{md}}_{\theta,\phi}(\mathbf{x}^{[T]}_\tau, \tau), \qquad (21)$$

which concludes the proof. $\square$

# E ADDITIONAL RESULTS

## E.1 CONFORMER PRETRAINING: QM9

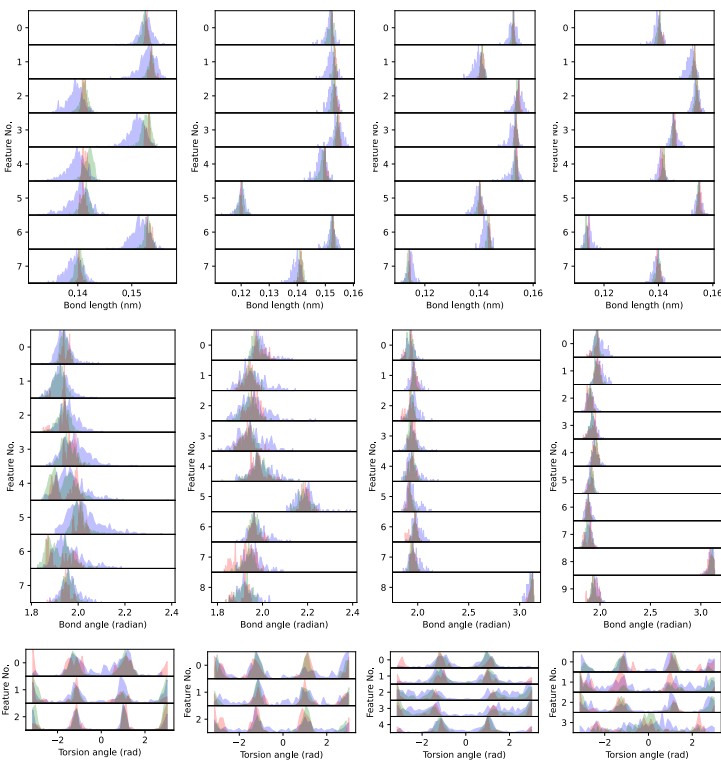

Figure 10: Distributions computed from reference conformers shown in red, Checkpoint 539 in green, and Checkpoint 99 in purple. We see that 539 aligns more closely with reference distributions across all collective variables and shows improved discretization of torsional states.

Above we show the additional plot associated with Section 5.1 and A.1. The plots above correspond to the following molecules (left to right):

N#C[C@](O)(CO)CCO, C[C@@H](O)[C@@H](CO)CC#N,
C[C@@H](O)CCOCCO, CC[C@@H](CC=O)[C@@H](C)O

## E.2 SPEEDUP ANALYSIS

Table 16: Average time (s) taken to generate trajectory

| Dataset & Duration | OpenMM MD | 4x Block Diffusion | Full Diffusion |
|---|---|---|---|
| Drugs (5.2 ns) | 584.52 | 201.70 | 161.08 |
| QM9 (2.6 ns) | 151.38 | - | 60.08 |

## E.3 UNCONDITIONAL GENERATION: QM9

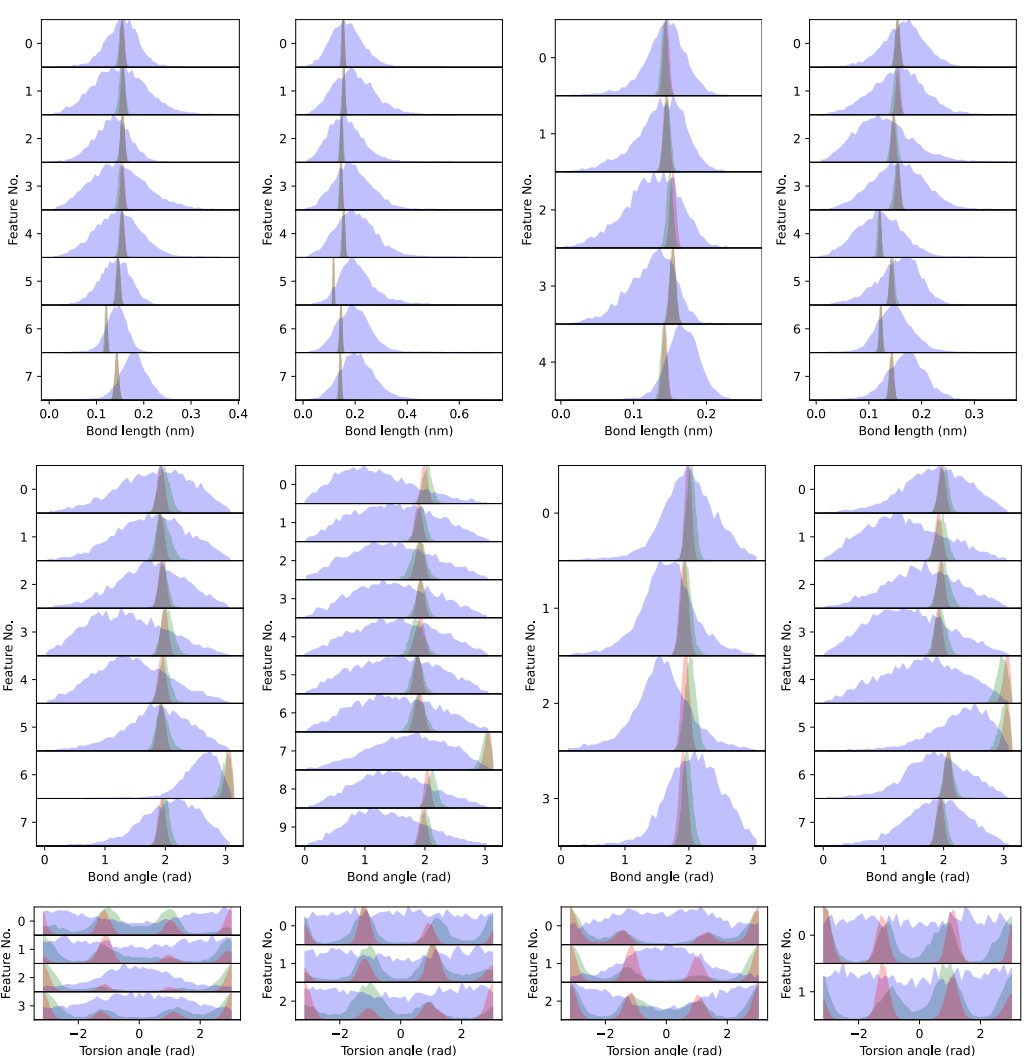

Figure 11: Distributions computed from reference QM9 trajectories (red), EGINTERPOLATOR (green), and GeoTDM (purple). Across all examples, our framework more closely matches the reference distributions across all collective variables and better captures torsional state discretizations than GeoTDM.

The figure above provides additional examples corresponding to the distributional analysis in Section 5.3. The molecule featured in the main paper in Figure 4A and 4B is:

CC[C@H](C#CC=O)CO

The plots above correspond to the following molecules (left to right):

C#CCCC[C@@H](C)CO,  CC[C@@](C#N)(CO)OC,

COCCCO,  CC[C@H](C#CC=O)CO

### E.4 FORWARD SIMULATION: DRUGS

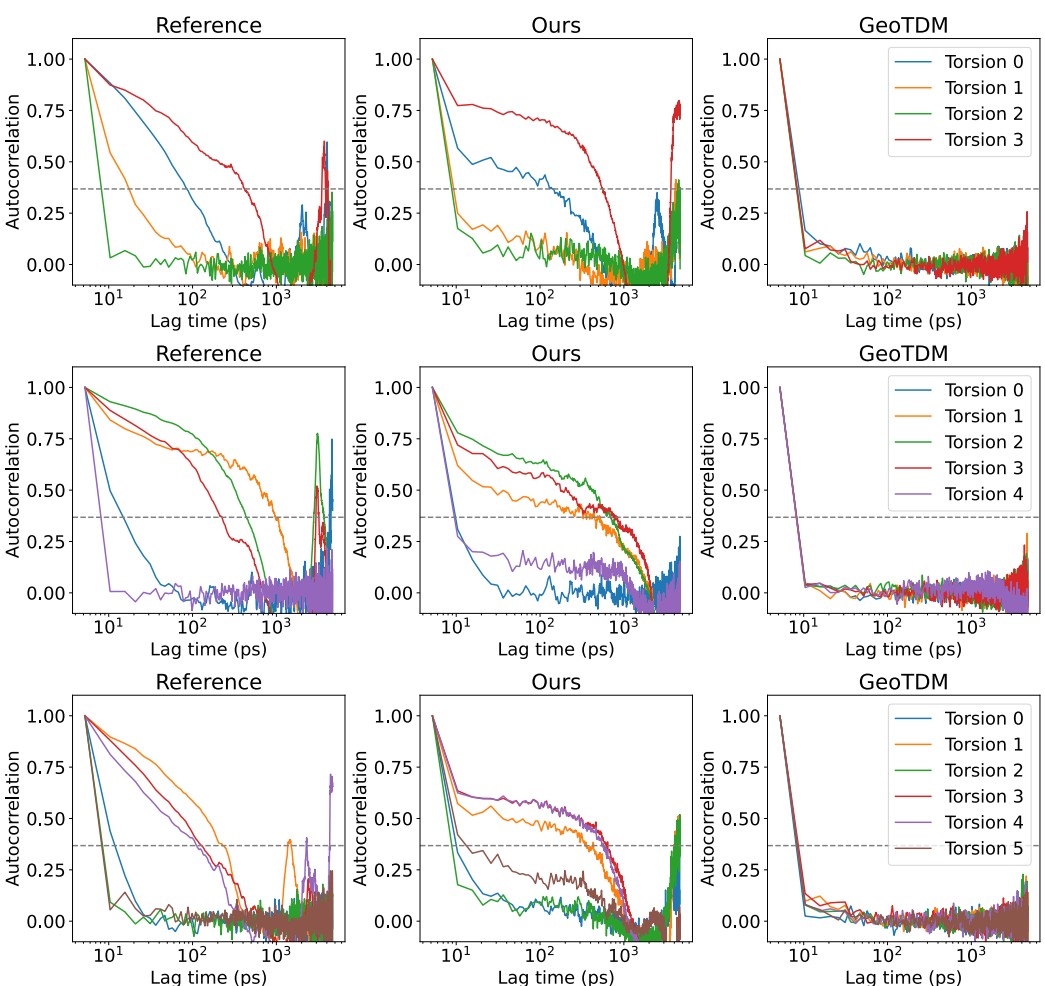

Figure 12: Autocorrelations of individual torsion angles for an example molecule, comparing reference trajectories with generations from EGINTERPOLATOR and GeoTDM. For the challenging task of capturing temporal de-correlation behavior, EGINTERPOLATOR closely follows the reference dynamics, whereas GeoTDM fails to model frame-to-frame correlations effectively.

The figure above provides additional examples corresponding to the dynamical analysis in Section 5.4. The molecule featured in the main paper in Figure 4E-G is:

```
O=C(O)c1[nH]c2ccc(Cl)cc2c1CC(=O)N1CCN(c2ccccc2)CC1
```

The plots above correspond to the following molecules (left to right):

```
Cc1ccc(C)c(CN2C(=O)NC3(CCCCC3)C2=O)c1,
COc1ccc(NS(=O)(=O)c2ccc3c(c2)Cc2ccccc2-3)cn1,
COc1ccc(S(=O)(=O)Nc2c(C(=O)O)[nH]c3ccccc23)c(OC)c1
```

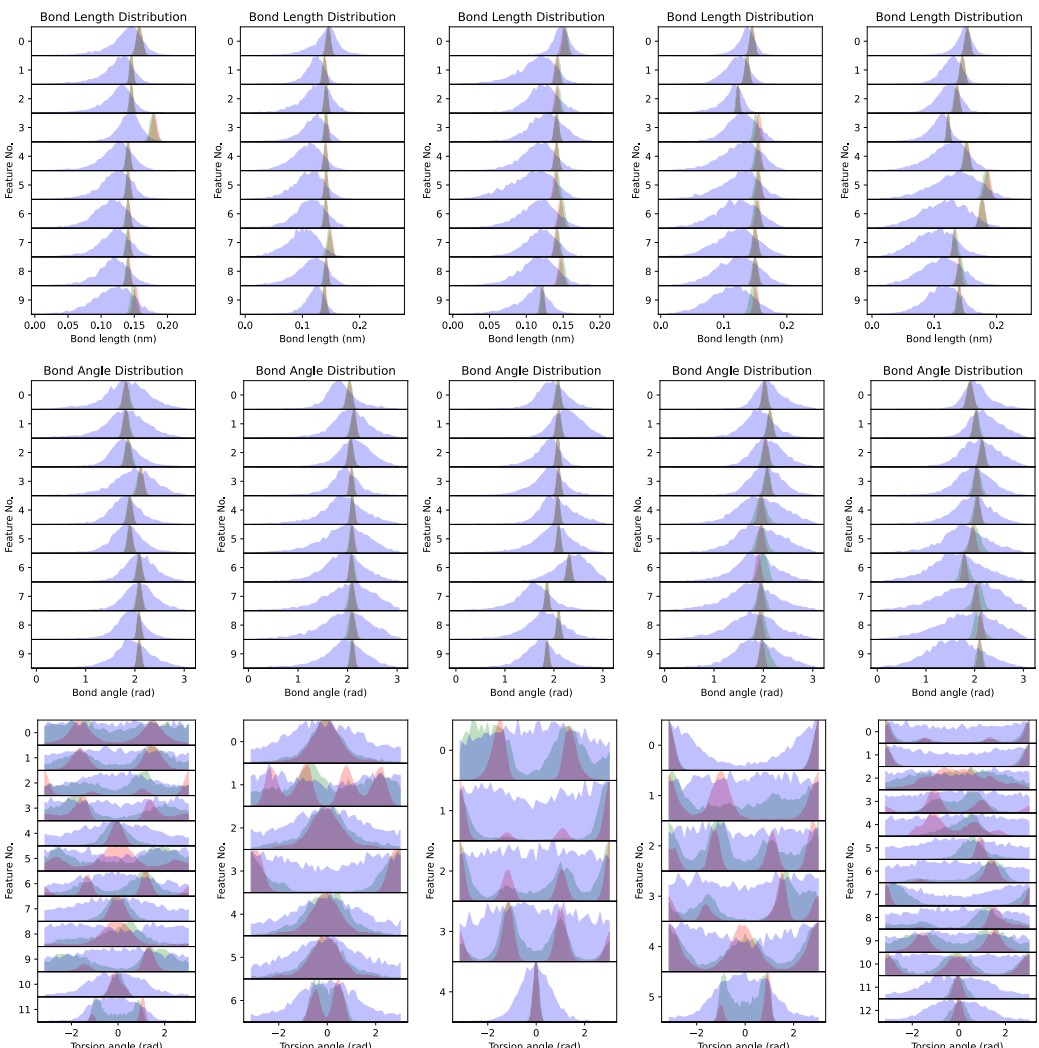

Figure 13: Distributions computed from reference Drugs trajectories (red), EGINTERPOLATOR (green), and GeoTDM (purple). Across all examples, our framework aligns closely with reference distributions across all collective variables and exhibits improved torsional state discretization compared to GeoTDM.

The figure above provides additional examples related to the distributional analysis in Section 5.4.

The plots above correspond to the following molecules (left to right):

NS(=O)(=O)c1ccc(CCNC(=O)COC(=O)CN2C(=O)[C@H]3CCCC[C@H]3C2=O)cc1,

COc1ccc(C(=O)N2CCc3cc(OC)c(OC)cc3C2)cc1OC,

Cc1ccc2c(c1)C(=O)N(CCCCO)C2=O,

COC(=O)C1CCN(Cc2cc(=O)oc3cc(OC)ccc23)CC1,

CCOC(=O)CSC1=Nc2ccccc2C2=N[C@H](CC(=O)NCc3ccc(OC)cc3)C(=O)N12

## E.5 ENERGY EXAMPLES: QM9 AND DRUGS

Table 17: **Top:** Mean and standard deviation (Hartrees) of energies for selected QM9 test molecules, comparing ground-truth (GT), EGINTERPOLATOR, and GEOTDM. **Bottom:** Block-wise energy means and standard deviations for selected Drugs test molecules, showing how EGINTERPOLATOR tracks GT distributions across successive diffusion blocks.

| SMILES | GT | EGInterpolator | GeoTDM |
|---|---|---|---|
| CC(CO)(CO)CC#N | $-440.287 \pm 0.005$ | $-440.249 \pm 0.036$ | $-438.206 \pm 1.813$ |
| COC[C@@]1(CO)N[C@H]1C | $-441.428 \pm 0.008$ | $-441.364 \pm 0.075$ | $-439.407 \pm 3.932$ |
| C#CCCC@HOCC | $-388.299 \pm 0.006$ | $-388.225 \pm 0.117$ | $-385.743 \pm 2.259$ |
| CCOCCCN1CC1 | $-405.525 \pm 0.007$ | $-405.387 \pm 0.426$ | $-402.751 \pm 3.424$ |
| CC(=O)C@HCCO | $-460.165 \pm 0.007$ | $-460.140 \pm 0.021$ | $-458.121 \pm 1.225$ |
| CCCC@@(CC)OC | $-390.780 \pm 0.006$ | $-390.753 \pm 0.026$ | $-387.954 \pm 3.434$ |
| CCC[C@@H]1C@HC[C@@H]1O | $-425.413 \pm 0.009$ | $-425.372 \pm 0.066$ | $-423.323 \pm 5.039$ |
| CCO[C@H]1C@@H[C@H]1CO | $-425.364 \pm 0.008$ | $-425.326 \pm 0.045$ | $-423.153 \pm 4.070$ |
| COCCC[C@H]1CN1C | $-405.507 \pm 0.008$ | $-405.463 \pm 0.047$ | $-403.101 \pm 4.087$ |
| CCC@HCC(C)C | $-389.583 \pm 0.007$ | $-389.547 \pm 0.035$ | $-386.846 \pm 2.405$ |

| SMILES | EGInterpolator Block | Energy (Hartrees) |
|---|---|---|
| Cc1ccc(C)c(CN2C(=O)NC3(CCCCC3)C2=O)c1 | GT | $-960.102 \pm 0.010$ |
| | Block 1 | $-960.062 \pm 0.020$ |
| | Block 2 | $-960.027 \pm 0.167$ |
| | Block 3 | $-959.940 \pm 0.307$ |
| | Block 4 | $-960.037 \pm 0.044$ |
| Cc1ccc(N[C@H]2CCCN(C(=O)c3ccc(-n4ccnc4)cc3)C2)cc1C | GT | $-1185.987 \pm 0.012$ |
| | Block 1 | $-1185.837 \pm 0.241$ |
| | Block 2 | $-1185.846 \pm 0.168$ |
| | Block 3 | $-1185.785 \pm 0.324$ |
| | Block 4 | $-1185.854 \pm 0.133$ |
| CCOC(=O)[C@H]1C@HNC(=O)N[C@@]1(O)C(F)(F)F | GT | $-2171.285 \pm 0.013$ |
| | Block 1 | $-2171.224 \pm 0.062$ |
| | Block 2 | $-2171.212 \pm 0.058$ |
| | Block 3 | $-2171.195 \pm 0.060$ |
| | Block 4 | $-2171.167 \pm 0.105$ |

## E.6 INTERPOLATION: QM9

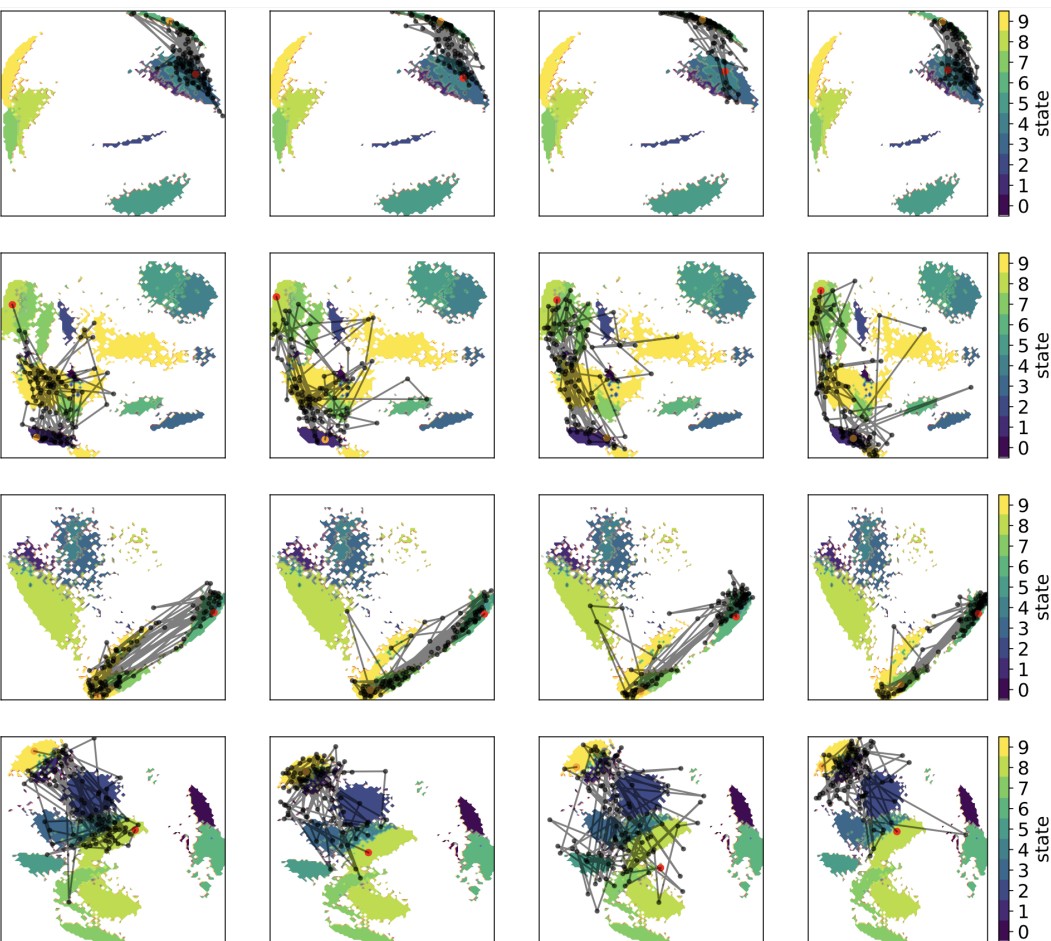

Figure 14: Generated QM9 interpolation trajectories from EGINTERPOLATOR, projected on the reference surface. The red point denotes the start frame, and the orange point denotes the end frame. The reference surface is colored by metastate assignment. Each row corresponds to a different molecule, and each column shows a generated interpolation. These examples illustrate the model's ability to generate efficient and meaningful transition paths.

The figure above provides additional examples related to the analysis in Section A.2.

The trajectories correspond to the following QM9 molecules (top to bottom):

C#C[C@@](O)(CC)COC, N#CC[C@H](O)CCCO,
C[C@H](C=O)NCC=O, CCC[C@@H](O)CC#N

## E.7 INTERPOLATION: DRUGS

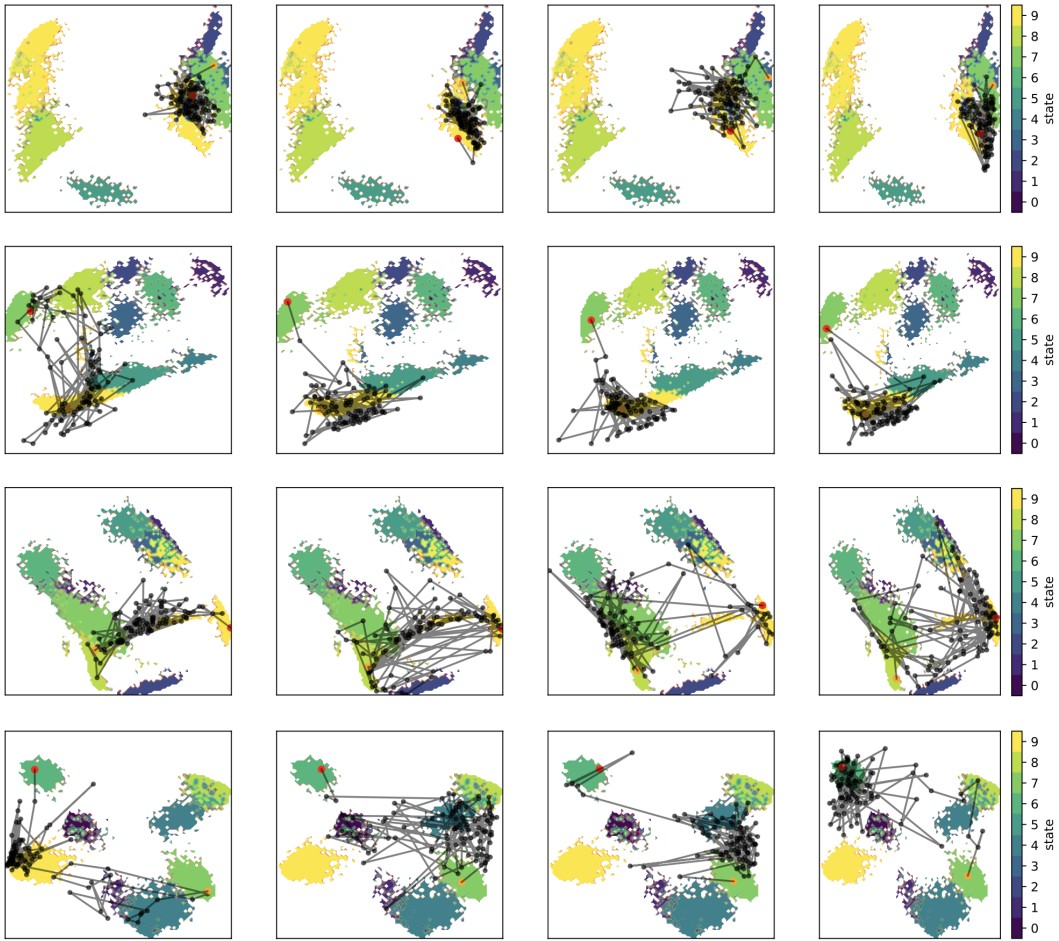

Figure 15: Generated Drug interpolation trajectories from EGINTERPOLATOR, projected onto the reference surface. The red point indicates the start frame, and the orange point indicates the end frame. The reference surface is colored by metastate assignment. Each row corresponds to a different molecule, and each column shows a generated interpolation. These examples highlight the model's ability to generate efficient and meaningful transition paths.

The figure above provides additional examples related to the analysis in Section 5.5. The molecule featured in the main paper in Figure 5B is:

O=C(CCCSc1nc2ccccc2[nH]1)NCc1ccccc1F

The trajectories above correspond to the following Drug molecules (top to bottom):

COc1ccc(S(=O)(=O)Nc2c(C(=O)O)[nH]c3ccccc23)c(OC)c1,

Cn1c(C(=O)NCCN2CCOCC2)cc2c(=O)n(C)c3ccccc3c21,

O=C(c1ccc(Br)o1)N1CCN(c2ccccc2F)CC1,

CCOC(=O)c1c(C)[nH]c(C)c1C(=O)CSc1ncccn1

### E.8 $\alpha$ MIXING PARAMETERS: INTERPOLATION RESULTS & EGINTERPOLATOR-SIMPLE

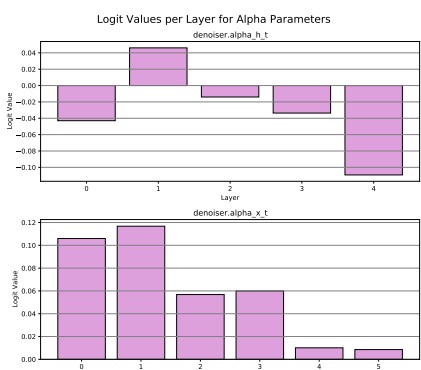

Figure 16: **Top:** Logits of $\alpha$ for each spatial and temporal layer after convergence on QM9. **Bottom:** Logits of $\alpha$ for each spatial and temporal layer after convergence on DRUGS. **Both:** Results obtained with EGINTERPOLATOR-CASC for the interpolation task.

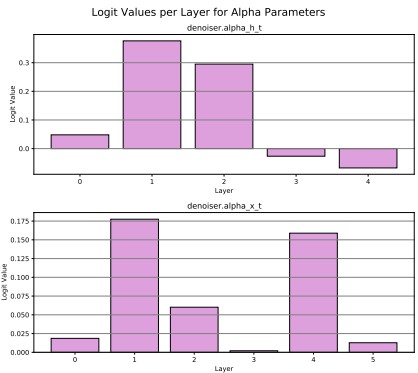

Logits of $\alpha$ for each spatial and temporal layer after convergence on the QM9 unconditional generation task.

Logits of $\alpha$ for each spatial and temporal layer after convergence on the DRUGS forward simulation task.

Figure 17: Results obtained with EGINTERPOLATOR-SIMPLE.

## F  STATEMENTS AND DISCUSSIONS

### F.1  LIMITATIONS CONT. AND FUTURE OPPORTUNITIES

Our results demonstrate that structural pretraining significantly enhances all-atom diffusion models for simulating molecular dynamics trajectories, especially on chemically diverse set of molecules. Nonetheless, our work has limitations that highlight directions for future research. As noted in Section 6, machine learning methods still lag behind ground-truth MD simulations in terms of physical accuracy. Future work may therefore explore improved learning objectives, molecular parameterizations, and the incorporation of physics-based regularization to help bridge this gap.

While our focus was primarily on the challenging domain of organic small molecules and addresses generalizeability in this chemical space, molecular dynamics is broadly applicable to other $N$-body systems, such as a peptides and protein–ligand complexes. While our efforts touched these realms, future work may robustly extend our framework, leveraging pretraining and structure prediction models, along with other physics-based or experimental signals, to enable generative modeling of larger bio-molecular simulations. Moreover, while we have shown promising results, continued efforts should look to unify the unique dynamics of small and large systems, as well as span multiple tasks.

Additionally, although our approach effectively reproduces distributions and dynamics consistent with classical mechanics, it remains subject to the inherent biases of molecular dynamics simulations. Future research may explore aligning both conformer and trajectory generation more closely with Boltzmann-distributed energy landscapes to improve thermodynamic fidelity.

### F.2  ETHICS AND IMPACTS STATEMENT

This work develops generative models for molecular dynamics to advance efficient, accurate simulation in chemistry and biology. While such models can accelerate scientific discovery, they also raise concerns around AI safety and dual-use risks, particularly in the design of harmful chemical or biological agents.

Our goal is to support beneficial applications in drug discovery, materials science, and molecular understanding through data-efficient and physically grounded modeling. All models are trained on publicly available, non-sensitive data and are released under open licenses to promote transparency and responsible use. We encourage continued dialogue on the safe development and deployment of generative AI in the physical and natural sciences.

