# OpenReview forum: "Align Your Structures: Generating Trajectories with Structure Pretraining for Molecular Dynamics"
_ICLR.cc/2026/Conference — ICLR 2026 Poster_

### Official Review · Reviewer_yR8j · 2025-10-18

**Soundness:** 3
**Presentation:** 3
**Contribution:** 3
**Rating:** 6
**Confidence:** 4

**Summary:**

This work introduces EGInterpolator, a novel diffusion-based model designed for modeling MD distributions. To address the data sparsity of MD trajectories, the model adopts a two-stage training strategy: it is first pretrained on a large conformer dataset and subsequently fine-tuned on MD data. A key component of the framework is a temporal interpolator, which captures temporal dependencies by integrating the pretrained unconditional generative model with a temporal network through linear interpolation. Experiments on GEOM-QM9 and GEOM-Drugs demonstrate that EGInterpolator achieves strong consistency with MD reference trajectories across unconditional generation, forward simulation, and interpolation tasks.

**Strengths:**

- The paper presents a clear research motivation and a well-organized logical flow. Considering the scarcity of MD data, the strategy of pretraining on large-scale conformer data followed by fine-tuning on MD data effectively reduces the complexity of spatio-temporal modeling of MD distributions.
- The temporal interpolator is designed in a concise and effective manner, enabling efficient temporal modeling while preserving the structural generation capability acquired during pretraining.
- The experiments on organic molecules thoroughly investigate tasks including unconditional generation, forward simulation, and transition path sampling, providing a comprehensive demonstration of the model’s capability to capture MD distributions.

**Weaknesses:**

- This work is evaluated only on small-molecule systems. However, for such systems with relatively few atoms, MD simulations using empirical force fields can achieve high accuracy at an acceptable computational cost, which may limit the practical advantages of the proposed model. The authors are encouraged to further justify the benefits of their approach over traditional MD methods or provide additional experiments on more complex biomolecular systems.

**Questions:**

1. For small-molecule systems, MD simulations using empirical force fields already offer a good balance between efficiency and accuracy. The authors should further clarify the advantages of their model over traditional MD methods or include additional experiments on more complex biomolecular systems.
2. Line 171 defines the so-called conformer distribution $p^{cf}(x)$, which is a potentially misleading definition. It can only be well-defined if it is based on an empirical data distribution (e.g., a dataset) or a known distribution (e.g., the Boltzmann distribution). The authors should clearly specify the definition.
3. In the MD finetuning stage, are the model parameters $\theta$ pretrained on the conformer dataset fixed or further optimized? This point does not appear to be clearly specified in Equation (3). If the parameters $\theta$ continue to be updated during finetuning, then the premise of Theorem 4.1 may not hold, since it would be unclear whether $\epsilon_{\theta}^{cf}$ remains consistent with the pretrained approximation of $p^{cf}$ during finetuning. This would undermine the practical validity of the theorem.
4. Lines 234–235 assume an extreme case, $\hat{p}^{md} = p^{md}$, to illustrate the advantage of the parametrization. I find this assumption inappropriate, as $\hat{p}^{md}$ assumes all conformations are i.i.d., while there exist temporal dependencies between consecutive MD samples, making it nearly impossible for their joint distribution to coincide with that of an i.i.d. case. Could the authors provide a reasonable justification for this assumption?
5. From the experimental results in Figure 3(A), the recall of the coverage and matching metrics on the QM9 dataset still falls short of SOTA performance. Could the authors provide possible explanations for this gap?
6. I have concerns about the setup of the interpolation task. Even though the model can generate a trajectory from the initial to the target state through the conditioning mask, the actual transition time corresponding to this process may be longer than 0.52 ns, which could result in the generated trajectory failing to capture the true underlying dynamics. Could the authors provide an explanation for this?
7. A typo: line 458 states that the generated MD trajectory spans 0.52 ns, whereas Figure 5D labels it as 1 ns.

**Details Of Ethics Concerns:**

None.

---

> ### Author Response · Authors · 2025-11-26
>
> We thank the reviewer for their detailed review, questions, and suggestions! We provide point-to-point response to the comments as follows.
>
> > **W1 & Q1. This work is evaluated only on small-molecule systems. The authors should further clarify the advantages of their model over traditional MD methods or include additional experiments on more complex biomolecular systems.**
>
> We thank the reviewer for this comment and suggestion. As noted in Sections 5.3–5.5, our small-molecule experiments demonstrate strong generalization to unseen organic compounds with diverse functional groups and topologies. We began in the small-molecule MD regime because it remains the standard, well-controlled benchmark for developing generative dynamics models. Importantly, modeling small-molecule dynamics is not trivial as systems such as QM9 and DRUGS decorrelate an order of magnitude faster (median 4.8 ps / 2 ps), requiring high-fidelity bond, angle, and torsion modeling, and span far chemical diversity.
>
> Our method also enables two capabilities that are not available in prior traditional approaches. First, the diffusion-based temporal interpolator allows interpolation sampling between metastable states, a setting where our method produces meaningful transition paths competitive with MD oracles (Section 5.5). Second, our framework allows a unique angle of inference-time α perturbation, allowing controlled modulation of temporal vs. spatial contributions and enabling increased conformational diversity in single-frame sampling (Section A.6.2). Together, these results highlight both the difficulty and value of the small-molecule regime and the novel modeling capabilities introduced by our approach.
>
> However, To further assess generality, we evaluate a single unified model trained jointly on two tasks (unconditional generation and forward simulation) and across two datasets (QM9 and DRUGS). This model performs substantially better than all baselines and closely matches the performance of specialized models; notably, its accuracy on QM9 slightly exceeds that of the QM9-only model. We hypothesize that the DRUGS dataset, with its greater chemical complexity and diversity, provides richer training signal that enhances generalization to the simpler QM9 regime. Overall, these findings show that a cross-task, cross-dataset model can perform competitively, underscoring the versatility of our approach. This effort is detailed in Appendix A.5 and the results are reported below.
>
> |Model|Bond Angle|Bond Length|Torsion|TICA_0|TICA_01
> |-|-|-|-|-|-|
> |EGInterpolator-Both QM9|**0.231** **0.219**|**0.168** **0.158**|**0.348** **0.367**|**0.393** **0.390**|**0.623** **0.631**|
> |EGInterpolator QM9|0.305 0.292|0.210 0.188|0.363 0.380|0.417 0.406|0.636 0.642|
> |GeoTDM QM9|0.691 0.690|0.676 0.670|0.489 0.527|0.449 0.453|0.691 0.694|
> |EGInterpolator-Both Drugs|0.212 0.197|0.216 0.195|0.417 0.434|0.488 0.506|0.681 0.679|
> |EGInterpolator Drugs|**0.173** **0.153**|**0.142** **0.112**|**0.377** **0.388**|**0.454** **0.441**|**0.650** **0.644**|
> |GeoTDM Drugs|0.640 0.645|0.643 0.645|0.498 0.503|0.531 0.550|0.712 0.720|
>
> We also evaluate our framework on tetrapeptide systems using the Timewarp datasets [2]. As conformer datasets for peptides are limited, we create our own peptide conformer dataset from the frames of the training trajectories, and this process is detailed in Appendix B.1.1. We generate 5 samples of 10 ns of dynamics at a 10 ps frame rate per peptide to compare to 50 ns of reference trajectories. We find that our method yields the strongest performance across distributional JSD metrics (Figure 6C), dynamical TICA/MSM metrics (Figure 6c), and energy-based analyses (Appendix A.6, Table 7) (Also reported below). These metrics are also supported by the results shown in Figure A & B, which highlight the improved FES and torsion decorrelations respectively by EGInterpolator on selected examples.
>
> |Metric|EGInterpolator|GeoTDM|
> |-|-|-|
> |Bond Angle (↓)|**0.102**|0.607|
> |Bond Length (↓)|**0.074**|0.643|
> |Torsion BB (↓)|**0.305**|0.520|
> |Torsion SC (↓)|**0.223**|0.434|
> |Torsion All (↓)|**0.256**|0.469|
> |TICA₀ (↓)|**0.282**|0.425|
> |TICA₀₁ (↓)|**0.462**|0.634|
> |MSM (↓)|**0.262**|0.344|
>
> Finally, to explore the scalability of our method, we are extending our framework to larger biomolecular systems using Boltz1 [4] on the ATLAS dataset [3], training models to forward-simulate 25 ns at a 100 ps frame rate. In Section 5.8, we present initial trajectory visualizations and validation metrics relative to the naïve Boltz1 baseline, showing improvements in torsion-angle JSDs and other early indicators. We are actively pursuing more comprehensive evaluation and comparisons to ensemble and dynamics models on these larger systems.
>
> We hope these experiments clearly demonstrate the breadth and scalability of our approach, as well as its promise across molecular regimes ranging from small molecules to peptides and larger biomolecular systems.

---

> > ### Author Response · Authors · 2025-11-26
> >
> > > **Q2. Line 171 defines the so-called conformer distribution, which is a potentially misleading definition.**
> >
> > We thank the reviewer for mentioning this point. We fully agree that defining it as the conformer data distribution is more precise and we have revised the manuscript to reflect this point.
> >
> > > **Q3. In the MD finetuning stage, are the model parameters pretrained on the conformer dataset fixed or further optimized?**
> >
> > We thank the reviewer for pointing this out. The parameters pretrained on the conformer dataset are kept fixed during MD finetuning, and only the temporal module is optimized; we apologize for the lack of clarity and will make this explicit in the revision, as it is essential for the practical validity of Theorem 4.1.
> >
> > > **Q4. Lines 234–235 assume an extreme case, $\hat{p}^\text{md} = p^\text{md}$, to illustrate the advantage of the parametrization.**
> >
> > We thank the reviewer for raising this point. In fact, we did not aim to argue that our parameterization only works under the assumption that $\hat{p}^\text{md} = p^\text{md}$. Instead, we want to show that under our parameterization, in the extreme case where $\hat{p}^\text{md} = p^\text{md}$, we only need an indentity mapping for the temporal block $\mathbf{s}\_\phi^\text{tp}$ which is easily reachable. By constrast, other naive parameterizations that do not include $\hat{\mathbf{\epsilon}}^\text{md}$ as the input will require the temporal block $\mathbf{s}\_\phi^\text{tp}$ to learn the frame-wise independent conformer score $\hat{\mathbf{\epsilon}}^\text{md}$ again, which unnecessarily brings more training overhead. Empirically, although $\hat{p}^\text{md} = p^\text{md}$ does not hold, it is still much easier to let the temporal block $\mathbf{s}_\phi^\text{tp}$ learn the difference between $\hat{p}^\text{md}$ and $p^\text{md}$, instead of directly learning the MD distribution $p^\text{md}$, which exactly jutifies the advantage of our parameterization.
> >
> >
> > > **Q5. From the experimental results in Figure 3(A), the recall of the coverage and matching metrics on the QM9 dataset still falls short of SOTA performance**
> >
> > We thank the reviewer for the question. As noted in Section 5.1, we intentionally prioritized precision-based metrics during conformer pretraining rather than recall-oriented coverage metrics, as higher geometric precision provides a stronger structural prior for downstream trajectory modeling. Appendix A.1 (Table 3; Fig. 9) analyzes this trade-off and shows that the reduced recall is accompanied by improved fidelity in bond-length, bond-angle, and torsion distributions, which are more critical for MD generation priors. The recall–precision Pareto frontier naturally differs across models due to variations in architecture, data cleaning, and heavy-atom handling, and our choice reflects an emphasis on high-quality structural priors rather than maximal conformer diversity.
> >
> > > **Q6. I have concerns about the setup of the interpolation task.**
> >
> > We thank the reviewer for this thoughtful comment. Our interpolation setup follows the standard MSM-based evaluation used in prior work such as MDGen [1]. The goal is to assess whether the model can generate geometrically and metastate-consistent transition pathways. The 0.52 ns rollout length is a fixed evaluation horizon applied equally to our model and to the MSM-sampled MD oracle paths. As shown in Section 5.5, Appendix A.2, and Appendix E.6-7, our trajectories achieve competitive metastate-sequence and path-probability statistics relative to these oracles, indicating that the generated transitions are plausible within the MSM transition landscape. We will clarify in the revision that the interpolation task evaluates path quality rather than absolute physical timescales.
> >
> > > **Q7. A typo: line 458 states that the generated MD trajectory spans 0.52 ns, whereas Figure 5D labels it as 1 ns.**
> >
> > We appreciate the reviewer for catching this and have made the correction in the revision.
> >
> >
> > [1] Jing et al., Generative modeling of MD trajectories, arXiv (2024).
> > [2] Klein et al. Timewarp: Transferable acceleration of molecular dynamics by learning time-coarsened dynamics. arXiv:2302.01170 (2023).
> > [3] Vander Meersche et al., ATLAS: Protein flexibility from MD simulations, NAR (2024).
> > [4] Wohlwend et al., Boltz-1: Democratizing biomolecular interaction modeling, bioRxiv (2024).

---

> > > ### Comment · Reviewer_yR8j · 2025-11-27
> > >
> > > Thank you for the authors' response. While some minor clarifications have been addressed, the core weaknesses of the work remain unresolved.
> > >
> > > First, at the molecular scale of small molecules and even peptides, classical MD simulations based on empirical force fields (e.g., AMBER) can already achieve high computational efficiency. To convincingly demonstrate the effectiveness of EGInterpolator, there should be empirical evidence showing that the model can reach accuracy comparable to classical MD while offering higher efficiency. Yet, the results in Table 1 indicate that the model is still notably distant from the MD oracle in terms of most metrics. Moreover, no comparison of inference efficiency (e.g., wall-clock time) is provided. In my view, this suggests that the practical advantages of the model are unsubstantiated.
> > >
> > > Second, judging from Table 1 and the experiments on the Timewarp dataset in the supplementary material, GeoTDM performs poorly on bond angles, bond lengths, and torsions, to the extent that it essentially fails to generate physically plausible conformations. I believe using such a weak baseline does not convincingly demonstrate the strength of the proposed method. Moreover, the field already offers stronger and more relevant baselines, such as MDGen [1], AlphaFlow [2], UniSim [3], BioEmu [4], Timewarp [5]. The absence of comparison with these models further limits the credibility of the empirical evaluation.
> > >
> > > Plus, as the rebuttal phase has already lasted for two weeks, it is quite feasible to reproduce at least one of these baselines on your setting. The authors' failure to do so, or to offer a compelling justification for why it was not possible, is unsatisfactory and further amplifies my concerns regarding the rigor of the evaluation.
> > >
> > > After careful consideration, I remain skeptical of the soundness and completeness of this work, and therefore I decide to lower my score.
> > >
> > > **Reference**
> > >
> > > > [1] Jing, B., Stärk, H., Jaakkola, T., & Berger, B. (2024). Generative modeling of molecular dynamics trajectories. Advances in Neural Information Processing Systems, 37, 40534-40564.
> > >
> > > > [2] Jing, B., Berger, B., & Jaakkola, T. (2024). AlphaFold meets flow matching for generating protein ensembles. arXiv preprint arXiv:2402.04845.
> > >
> > > > [3] Yu, Z., Huang, W., & Liu, Y. UniSim: A Unified Simulator for Time-Coarsened Dynamics of Biomolecules. In Forty-second International Conference on Machine Learning.
> > >
> > > > [4] Lewis, S., Hempel, T., Jiménez-Luna, J., Gastegger, M., Xie, Y., Foong, A. Y., ... & Noé, F. (2025). Scalable emulation of protein equilibrium ensembles with generative deep learning. Science, 389(6761), eadv9817.
> > >
> > > > [5] Klein, L., Foong, A., Fjelde, T., Mlodozeniec, B., Brockschmidt, M., Nowozin, S., ... & Tomioka, R. (2023). Timewarp: Transferable acceleration of molecular dynamics by learning time-coarsened dynamics. Advances in Neural Information Processing Systems, 36, 52863-52883.

---

> > > > ### Author Response · Authors · 2025-12-03
> > > >
> > > > We once again appreciate the reviewer for the comments! We provide additional results referenced in our prior responses below:
> > > >
> > > > We include in Section 5.6, comparing \textsc{EGInterpolator} to a variant we call \textsc{EGInterpolator-Stack}. In this baseline, we remove both layer cascading and spatial–temporal interpolation, instead appending an equivalent number of temporally focused residual layers as a finetuned head atop the pretrained (and frozen) spatial model. As shown in Table 2, our cascaded, interpolated design delivers substantially better performance. These results empirically justify the added architectural complexity and support our theoretical motivation for interpolation.
> > > >
> > > > | Method                | Bond Angle M | Bond Angle Md | Bond Length M | Bond Length Md | Torsion M | Torsion Md | TICA₀,₁ M | TICA₀,₁ Md |
> > > > |-|-|-|-|-|-|-|-|-|
> > > > | EGInterpolator-S      | 0.325 | 0.330 | 0.330 | 0.321 | 0.414 | 0.419 | 0.673 | 0.672 |
> > > > | EGInterpolator-N      | 0.332 | 0.332 | 0.386 | 0.383 | 0.455 | 0.466 | 0.698 | 0.703 |
> > > > | **EGInterpolator**    | **0.173** | **0.153** | **0.142** | **0.112** | **0.377** | **0.388** | **0.650** | **0.644** |
> > > >
> > > > To further assess our method, we benchmark on the MDGen [1] tetrapeptide dataset. Using the procedure described in Appendix B.1.1, we subsample MDGen training trajectories into a peptide conformer dataset and combine it with a pruned TimeWarp conformer set to avoid data leakage. We then train a conformer model on this combined dataset, initializing from the GEOM-DRUGS pretrained weights. Our results highlight a key strength of \textsc{EGInterpolator}: it accurately models bond lengths and bond angles, which are critical for all-atom molecular dynamics. However, our approach underperforms MDGen in torsional metrics and downstream dynamical measures, suggesting that MDGen’s torsion-focused parameterization provides an advantage in capturing peptide rotational behavior.
> > > >
> > > > | Metrics | EGInterpolator | MDGen |
> > > > |-|-|-|
> > > > | Bond Angle | 0.092 | NA |
> > > > | Bond Length | 0.056 | NA |
> > > > | Torsion BB | 0.378 | 0.130 |
> > > > | Torsion SC | 0.189 | 0.093 |
> > > > | Torsion All | 0.265 | 0.109 |
> > > > | TICA₀ | 0.409 | 0.230 |
> > > > | TICA₀₁ | 0.568 | 0.316 |
> > > > | MSM | 0.312 | 0.235 |
> > > > [1] Jing, B., Stärk, H., Jaakkola, T., & Berger, B. (2024). Generative modeling of molecular dynamics trajectories. Advances in Neural Information Processing Systems, 37, 40534-40564

---

### Official Review · Reviewer_c4YK · 2025-10-26

**Soundness:** 3
**Presentation:** 3
**Contribution:** 2
**Rating:** 6
**Confidence:** 3

**Summary:**

This paper presents EGInterpolator, a framework for generating molecular dynamics (MD) trajectories by leveraging structure pretraining. The key idea is to first pretrain a diffusion model on large-scale conformer datasets, then introduce a temporal interpolator module trained on limited MD data to enforce temporal consistency.

**Strengths:**

1. The authors test their method across multiple tasks (unconditional generation, forward simulation, interpolation) and provide extensive metrics including JSD for various metrics. The ablation studies are quite thorough.
2. Using structure pretraining to leverage abundant conformer data is a sensible solution that makes practical sense.
3. The method outperforms baselines and performs impressive results on the interpolation.

**Weaknesses:**

1. The experiments are restricted to small organic molecules (QM9 has molecules with ≤9 heavy atoms). While the authors acknowledge this in limitations, it's unclear how well this approach would scale to larger, more practical systems like proteins or protein-ligand complexes. The method's utility for real-world drug discovery applications remains uncertain without demonstration on larger molecules (As far as I knew, there are also large molecules in GEOM dataset).
2.  In several metrics, the model performs worse than even short MD oracle trajectories. This suggests the generated dynamics may be too fast or not physically accurate enough for practical applications. The authors should discuss this gap more critically.
3.  Theorem 4.1 shows the interpolator induces an intermediate distribution, but why this particular interpolation strategy is optimal, and how the choice of α affects the bias-variance tradeoff.

**Questions:**

1. How much conformer data is actually needed? What if you pretrain on a smaller, more targeted set of conformers? This would help understand the data efficiency of your approach.
2. You train separate models for QM9 and Drugs. Have you tried training a single model on both datasets? What prevents the method from generalizing across different molecular systems in a unified way?
3. What about long-time stability - do generated trajectories eventually diverge or produce unphysical configurations?
4. Table 5 shows some degradation in later blocks for forward simulation. How many blocks can you roll out before quality becomes unacceptable? Is there a way to prevent this deterioration?

---

> ### Author Response · Authors · 2025-11-26
>
> We thank the reviewer for their detailed review, questions, and suggestions! We provide point-to-point response to the comments as follows.
>
> > **W1. The experiments are restricted to small organic molecules.**
>
> We thank the reviewer for this comment. First, as mentioned by the reviewer, although QM9 are small in heavy atom size, our experiments in the DRUGS datasets highlight our framework on classes of larger and more diverse molcules and chemistries. Namely DRUGS contians a large list of heavy atoms including *C, N, O, S, P, F, Cl, Br, I, B, Si* and molecules in the range of 30-40 heavy atoms.
>
> We also additionally evaluate our framework on tetrapeptide systems using the Timewarp datasets [1]. As conformer datasets for peptides are limited, we create our own peptide conformer dataset from the frames of the training trajectories, and this process is detailed in Appendix B.1.1. We generate 5 samples of 10 ns of dynamics at a 10 ps frame rate per peptide to compare to 50 ns of reference trajectories. We find that our method yields the strongest performance across distributional JSD metrics (Figure 6C), dynamical TICA/MSM metrics (Figure 6c), and energy-based analyses (Appendix A.6, Table 7) (Also reported below). These metrics are also supported by the results shown in Figure A & B, which highlight the improved FES and torsion decorrelations respectively by EGInterpolator on selected examples.
>
> |Metric|EGInterpolator|GeoTDM|
> |-|-|-|
> |Bond Angle (↓)|**0.102**|0.607|
> |Bond Length (↓)|**0.074**|0.643|
> |Torsion BB (↓)|**0.305**|0.520|
> |Torsion SC (↓)|**0.223**|0.434|
> |Torsion All (↓)|**0.256**|0.469|
> |TICA₀ (↓)|**0.282**|0.425|
> |TICA₀₁ (↓)|**0.462**|0.634|
> |MSM (↓)|**0.262**|0.344|
>
> Finally, to explore the scalability of our method, we are extending our framework to larger biomolecular systems using Boltz1 [3] on the ATLAS dataset [2], training models to forward-simulate 25 ns at a 100 ps frame rate. In Section 5.8, we present initial trajectory visualizations and validation metrics relative to the naïve Boltz1 baseline, showing improvements in torsion-angle JSDs and other early indicators. We are actively pursuing more comprehensive evaluation and comparisons to ensemble and dynamics models on these larger systems.
>
> We hope these experiments clearly demonstrate the breadth and scalability of our approach, as well as its promise across molecular regimes ranging from small molecules to peptides and larger biomolecular systems.
>
> > **W2. In several metrics, the model performs worse than even short MD oracle trajectories.**
>
> We appreciate the reviewer’s point and agree that our model underperforms MD oracles on certain fast-timescale dynamical metrics as rapid small-molecule motions are inherently difficult. At the same time, our method consistently outperforms all learned baselines, and on the interpolation task it achieves performance competitive with MD oracles, indicating that it captures the underlying transition pathways. Nonetheless, we appreciate the reviewer’s feedback and will make this limitation more explicit in the paper.
>
> > **W3. Theorem 4.1 shows the interpolator induces an intermediate distribution, but why this particular interpolation strategy is optimal, and how the choice of α affects the bias-variance tradeoff.**
>
> In fact, Theorem 4.1 conveys the following information: When we fit the score $\mathbf{\epsilon}^\text{tp}\_{\theta,\phi}$ toward the MD data distribution $p^\text{md}(\mathbf{x}^{[T]})$, we are essentially optimizing the additionally introduced parameters $\phi$ such that their induced score function $\mathbf{\epsilon}\_\phi$ models the interpolant $p^\text{md}(\mathbf{x}^{[T]})^\beta \hat{p}^\text{md}(\mathbf{x}^{[T]})^{1-\beta}$. Therefore, we want to highlight that the target distribution is always the MD data distribution $p^\text{md}(\mathbf{x}^{[T]})$ and we do not introduce any bias in the target distribution. Theorem 4.1 instead assures that by using our linear interpolation design, the new parameters $\phi$ only need to fit the $\beta$-reweighted interpolant as opposed to the highly complex MD distribution $p^\text{md}(\mathbf{x}^{[T]})$, while the final output score is still guaranteed to fit the MD distribution. Therefore, our iterpolation strategy effectively offloads the modeling complexity in capturing the complex MD distribution into an intermediate simpler distribution between the conformer distribution and the MD distribution, whose efficacy has also been verified in our experiments.

---

> > ### Author Response · Authors · 2025-11-26
> >
> > > **Q1. How much conformer data is actually needed?**
> >
> > We appreciate the reviewer’s question regarding data efficiency. Our tetrapeptide experiments provide a useful case study: when comparing (i) no pretraining to (ii) using only the limited tetrapeptide conformers, we observe that the latter yields the strongest downstream dynamics performance, indicating that meaningful gains can be achieved even with limited and targeted conformer data. Moreover, in our combined QM9–DRUGS model (Q2), broader conformer pretraining with DRUGS improves QM9 trajectory quality, but models trained with more restricted pretraining still perform reasonably well. Overall, while increased conformer diversity and scale improve trajectory generation, our results suggest that the approach remains data-efficient and can benefit substantially from smaller, domain-specific conformer sets when large datasets are not available.
> >
> > > **Q2. You train separate models for QM9 and Drugs. Have you tried training a single model on both datasets?**
> >
> > We thank reviewer for this suggestion. To further assess generality, we evaluate a single unified model trained jointly on two tasks (unconditional generation and forward simulation) and across two datasets (QM9 and DRUGS), including conformer pretraining across both. This model performs substantially better than all baselines and closely matches the performance of specialized models; notably, its accuracy on QM9 slightly exceeds that of the QM9-only model. We hypothesize that the DRUGS dataset, with its greater chemical complexity and diversity, provides richer training signal that enhances generalization to the simpler QM9 regime. Overall, these findings show that a cross-task, cross-dataset model can perform competitively, underscoring the versatility of our approach. This effort is detailed in Appendix A.5 and the results are reported below.
> >
> > |Model|Bond Angle|Bond Length|Torsion|TICA_0|TICA_01
> > |-|-|-|-|-|-|
> > |EGInterpolator-Both QM9|**0.231** **0.219**|**0.168** **0.158**|**0.348** **0.367**|**0.393** **0.390**|**0.623** **0.631**|
> > |EGInterpolator QM9|0.305 0.292|0.210 0.188|0.363 0.380|0.417 0.406|0.636 0.642|
> > |GeoTDM QM9|0.691 0.690|0.676 0.670|0.489 0.527|0.449 0.453|0.691 0.694|
> > |EGInterpolator-Both Drugs|0.212 0.197|0.216 0.195|0.417 0.434|0.488 0.506|0.681 0.679|
> > |EGInterpolator Drugs|**0.173** **0.153**|**0.142** **0.112**|**0.377** **0.388**|**0.454** **0.441**|**0.650** **0.644**|
> > |GeoTDM Drugs|0.640 0.645|0.643 0.645|0.498 0.503|0.531 0.550|0.712 0.720|
> >
> > > **Q3. What about long-time stability - do generated trajectories eventually diverge or produce unphysical configurations?**
> >
> > To further demonstrate long-timescale generation, in Appendix A.4 and A.6.2 we report results from producing 16 consecutive block rollouts with the DRUGS forward-simulation model, yielding 20.8 ns of continuous dynamics. We compare these to our original 4-block diffusion model, which was sampled in parallel five times to produce 26 ns of generated trajectories. The long-simulation model performs substantially better than all baselines across distributional, dynamical, and energetic metrics, even in this more challenging regime, and its quality remains reasonably close to that of the 4-block model. We also analyze error propagation over the 16 blocks and observe that while performance begins to degrade after block 8, it remains markedly closer to ground truth than GeoTDM—even within the 4-block setting.
> >
> > |Method|Bond Angle|Bond Length|Torsion|TICA₀|TICA₀,₁|
> > |-|-|-|-|-|-|
> > |GeoTDM|0.640 0.645|0.643 0.645|0.498 0.503|0.531 0.550|0.712 0.720|
> > |EGInterpolator|0.173 0.153|0.142 0.112|0.377 0.388|0.454 0.441|0.650 0.644|
> > |EGInterpolator-Long|0.180 0.155|0.147 0.116|0.404 0.411|0.484 0.484|0.685 0.680|
> >
> > > **Q4. Table 5 shows some degradation in later blocks for forward simulation. How many blocks can you roll out before quality becomes unacceptable? Is there a way to prevent this deterioration?**
> >
> > As noted in our response to Q3, our error-propagation analysis over 16 consecutive blocks shows that trajectory quality begins to degrade after roughly 8 blocks, though it remains substantially closer to ground truth than GeoTDM—even within the shorter 4-block regime. In this paper, we focus on models trained purely on trajectory data to isolate the contributions of interpolation and pretraining. However, future extensions could incorporate force- or energy-based supervision, inference-time guidance, or selection strategies such as beam search over candidate blocks (e.g., choosing segments with lower energies) to mitigate energetic and physical drift during long block-diffusion rollouts.
> >
> >
> > [1] Klein et al. Timewarp: Transferable acceleration of molecular dynamics by learning time-coarsened dynamics. arXiv:2302.01170 (2023).
> > [2] Vander Meersche et al., ATLAS: Protein flexibility from MD simulations, NAR (2024).
> > [3] Wohlwend et al., Boltz-1: Democratizing biomolecular interaction modeling, bioRxiv (2024).

---

> ### Author Response · Authors · 2025-12-03
>
> We once again appreciate the reviewer for the comments! We provide additional results referenced in our prior responses below:
>
> We include an ablation in Section 5.6, comparing \textsc{EGInterpolator} to a variant we call \textsc{EGInterpolator-Stack}. In this baseline, we remove both layer cascading and spatial–temporal interpolation, instead appending an equivalent number of temporally focused residual layers as a finetuned head atop the pretrained (and frozen) spatial model. As shown in Table 2, our cascaded, interpolated design delivers substantially better performance. These results empirically justify the added architectural complexity and support our theoretical motivation for interpolation.
>
> | Method                | Bond Angle M | Bond Angle Md | Bond Length M | Bond Length Md | Torsion M | Torsion Md | TICA₀,₁ M | TICA₀,₁ Md |
> |-|-|-|-|-|-|-|-|-|
> | EGInterpolator-S      | 0.325 | 0.330 | 0.330 | 0.321 | 0.414 | 0.419 | 0.673 | 0.672 |
> | EGInterpolator-N      | 0.332 | 0.332 | 0.386 | 0.383 | 0.455 | 0.466 | 0.698 | 0.703 |
> | **EGInterpolator**    | **0.173** | **0.153** | **0.142** | **0.112** | **0.377** | **0.388** | **0.650** | **0.644** |

---

### Official Review · Reviewer_YkBB · 2025-10-30

**Soundness:** 4
**Presentation:** 3
**Contribution:** 3
**Rating:** 4
**Confidence:** 4

**Summary:**

The paper proposes a two-stage framework for generating molecular dynamics (MD) trajectories using diffusion models. First, they train a large structure diffusion model on static molecular conformers (like QM9, GEOM-Drugs). This model learns how valid 3D molecular structures look like. Subsequently, they freeze it and train a smaller temporal interpolator on a limited set of MD trajectories. This temporal module learns to align the static structures in time, adding smooth and physically consistent motion between them. The combined system, called EGINTERPOLATOR, can generate realistic molecular trajectories even when there’s very little MD data available. It also keeps SE(3)-equivariance and can handle different generation modes (simulation, interpolation, etc.).

**Strengths:**

- The idea to separate spatial structure learning from temporal dynamics is elegant and practical. It reduces data requirements and improves generalization.
- The temporal interpolator design is well conceived — it works as a learned guidance or adapter that connects independent conformer frames into a coherent trajectory.
- Experiments are convincing, with good results on QM9 and GEOM-Drugs datasets. The model seems to generate smoother and more realistic bond and torsion distributions than baselines.
- It’s well grounded in symmetry (SE(3) equivariance), which is crucial for molecular data.
- Conceptually it’s similar to the trend in video diffusion models, but nicely adapted to the molecular domain.

**Weaknesses:**

- The physical validation is lacking — results are mostly on geometric statistics. No tests about energy conservation, temperature stability, or realistic MD physics.
- The method depends a lot on the pretrained conformer model. If that model is not good, the whole system might fail.
- It’s not clear how this would scale to bigger molecules (like proteins) or longer trajectories.
- The temporal module is still a bit of a black box. The paper doesn’t show much intuition about what it actually learns.
- Fine-tuning with small MD datasets could overfit, and the paper doesn’t really study that.
- Fundamentally, the architecture setup is still sequential when generating whereas BioEmu for example targets the equilibrium distribution directly.

**Questions:**

1) How stable are the generated trajectories over long time horizons? Do they drift away from realistic energy basins?
2) Could the temporal interpolator be trained or conditioned on energy or force information to improve physical consistency?
3) Could the method handle non-equilibrium or biased simulations?
4) What’s the computational cost compared to training a full end-to-end trajectory diffusion model?
5) Have you run experiments that in the limit sample from all meta-states from a single starting red point in appendix E6 and E7?

---

> ### Author Response · Authors · 2025-11-26
>
> We thank the reviewer for their detailed review, questions, and suggestions! We provide point-to-point response to the comments as follows.
>
> > **W1. The physical validation is lacking — results are mostly on geometric statistics. No tests about energy conservation, temperature stability, or realistic MD physics.**
>
> We agree that evaluating physical realism is crucial. In our current results, we already assess several proxies of physical fidelity by comparing against reference MD simulations and GeoTDM across both (1) structural and distributional accuracy—e.g., JSDs in Table 1, bond-length and torsion-angle histograms in Fig. 4A–B, and additional examples in App. Fig. 10 and Fig. 12—and (2) dynamic and transition fidelity—e.g., MSM state populations and autocorrelation decay (Fig. 4C–E; App. Fig. 11), as well as transition-path MSM analyses (Fig. 5 and App. Fig. 6).
>
> Beyond these metrics, we also include energy-based analyses in Appendix A.4 for both QM9 unconditional generation and DRUGS forward-simulation tasks. Table 5 reports the average Wasserstein-1 distance between predicted and ground-truth energy profiles, showing that our method matches the reference MD simulations substantially more closely than GeoTDM. In addition, Appendix E.5 (Table 12) provides example molecule comparisons of predicted energies from our model and GeoTDM versus MD reference across both QM9 and DRUGS, where we observe promising alignment. We will be sure to make these results more prominent and provide additional context in the revision!
>
> > **W2. The method depends a lot on the pretrained conformer model. If that model is not good, the whole system might fail.**
>
> We thank the reviewer for raising this concern. While our framework does rely on a pretrained conformer model to provide structural priors, in practice this dependence is not a limiting factor. First, static conformer datasets are far easier to obtain than time-resolved MD trajectories, and modern generative conformer models are generally quite reliable; as shown in Fig. 3A, the conformer backbone we use performs competitively with state-of-the-art generators.
>
> Second, our interpolator includes spatial update layers that refine structural predictions during trajectory generation. As demonstrated in the table below, the full interpolator improves bond lengths, bond angles, and torsions relative to the α = 1 setting (which disables temporal interpolation), indicating that the system can correct imperfections in the conformer prior rather than inherit them. (Section A.7.5)
>
> |Model|Bond Angle|Bond Length|Torsion|
> |-|-|-|-|
> |EGInterpolator Normal QM9|0.305 0.292|0.210 0.188|0.363 0.380|
> |EGInterpolator $\alpha=1$ QM9|0.398 0.391|0.618 0.613|0.358 0.372|
> |EGInterpolator Normal Drugs|0.173 0.153|0.142 0.112|0.377 0.388|
> |EGInterpolator $\alpha=1$ Drugs|0.435 0.445|0.580 0.091|0.378 0.382|
>
> Moreover, cross-domain pretraining can further improve robustness. We evaluate a single unified model trained jointly on two tasks (unconditional generation and forward simulation) and across two datasets (QM9 and DRUGS), with conformer pretraining performed on both. This unified model performs substantially better than all baselines and closely matches the performance of task- and dataset-specific models; notably, its accuracy on QM9 even exceeds that of the QM9-only variant. We hypothesize that the richer chemical diversity of DRUGS provides stronger training signal that transfers to the simpler QM9 regime. These results highlight that a cross-task, cross-dataset model can remain highly competitive, underscoring the versatility of our approach. This is highlighted in Appendix A.5 and reported below:
>
> |Model|Bond Angle|Bond Length|Torsion|TICA_0|TICA_01
> |-|-|-|-|-|-|
> |EGInterpolator-Both QM9|**0.231** **0.219**|**0.168** **0.158**|**0.348** **0.367**|**0.393** **0.390**|**0.623** **0.631**|
> |EGInterpolator QM9|0.305 0.292|0.210 0.188|0.363 0.380|0.417 0.406|0.636 0.642|
> |GeoTDM QM9|0.691 0.690|0.676 0.670|0.489 0.527|0.449 0.453|0.691 0.694|
> |EGInterpolator-Both Drugs|0.212 0.197|0.216 0.195|0.417 0.434|0.488 0.506|0.681 0.679|
> |EGInterpolator Drugs|**0.173** **0.153**|**0.142** **0.112**|**0.377** **0.388**|**0.454** **0.441**|**0.650** **0.644**|
> |GeoTDM Drugs|0.640 0.645|0.643 0.645|0.498 0.503|0.531 0.550|0.712 0.720|
>
> In another lens, our tetrapeptide experiments using the Timewarp datasets [1] demonstrate robustness when conformer data are scarce. Because peptide conformer datasets are limited, we in fact we create our own peptide conformer dataset from the frames of the training trajectories, and this process is detailed in Appendix B.1.1. Across 5×10 ns generated trajectories (10 ps frame rate) evaluated against 50 ns reference MD, this pretrained EGInterpolator scheme consistently performs best across distributional JSD metrics, TICA/MSM dynamical metrics, and energy-based analyses (Figure 6, Appendix A.6).

---

> > ### Author Response · Authors · 2025-11-26
> >
> > > **W3. It’s not clear how this would scale to bigger molecules (like proteins) or longer trajectories.**
> >
> > We thank the reviewer for this question and agree that scaling to larger molecules and longer trajectories is an important long-term goal. Our method is designed with this in mind. It makes no molecule-specific assumptions and represents each system as a 3D heterogeneous graph, allowing the same architecture to operate on small molecules, peptides, and larger biomolecules. The 3-hop local neighborhood (Appendix C.1) captures relevant torsional interactions while keeping computational cost near-linear in the number of atoms. To address longer rollouts, Section 5.4 introduces block rollouts, which segment the diffusion process and reduce memory usage. The alternating spatial–temporal attention has worst-case complexity (O(NT^2) + O(N^2 T)), but the (N^2) term is substantially mitigated by local neighborhoods. Together, these design choices enable the architecture to scale while maintaining fidelity.
> >
> > To further demonstrate long-timescale generation, in Appendix A.4 (Table 5) we report results from producing 16 consecutive block rollouts with the DRUGS forward-simulation model, yielding 20.8 ns of continuous dynamics. We compare these to our original 4-block diffusion model, which was sampled in parallel five times to produce 26 ns of generated trajectories. The long-simulation model performs substantially better than all baselines across distributional, dynamical, and energetic metrics, even in this more challenging regime, and its quality remains reasonably close to that of the 4-block model (Results reported below). We also analyze error propagation over the 16 blocks and observe that while performance begins to degrade after block 8, it remains markedly closer to ground truth than GeoTDM—even within the 4-block setting (Appendix A.6.2, Table 8).
> >
> > |Method|Bond Angle|Bond Length|Torsion|TICA₀|TICA₀,₁|
> > |-|-|-|-|-|-|
> > |GeoTDM|0.640 0.645|0.643 0.645|0.498 0.503|0.531 0.550|0.712 0.720|
> > |EGInterpolator|0.173 0.153|0.142 0.112|0.377 0.388|0.454 0.441|0.650 0.644|
> > |EGInterpolator-Long|0.180 0.155|0.147 0.116|0.404 0.411|0.484 0.484|0.685 0.680|
> >
> > Finally, to explore the scalability of our method, we are extending our framework to larger biomolecular systems using Boltz1 [3] on the ATLAS dataset [2], training models to forward-simulate 25 ns at a 100 ps frame rate. In Section 5.8, we present initial trajectory visualizations and validation metrics relative to the naïve Boltz1 baseline, showing improvements in torsion-angle JSDs and other early indicators. We are actively pursuing more comprehensive evaluation and comparisons to ensemble and dynamics models on these larger systems.
> >
> > We hope these experiments clearly demonstrate the breadth and scalability of our approach, as well as its promise across molecular regimes ranging from small molecules to peptides and larger biomolecular systems.
> >
> >
> > > **W4. The temporal module is still a bit of a black box. The paper doesn’t show much intuition about what it actually learns.**
> >
> > We thank the reviewer for raising this point. In addition to the results reported in W2, our analyses show that the temporal module has meaningful and interpretable behavior. Appendix A.5.4 (Table 7) demonstrates that it captures realistic torsional dynamics—without it, autocorrelation functions collapse to degenerate behavior. Appendix A.6.2 (Table 8) further shows that inference-time perturbations of the learned mixing parameter α increase conformer diversity with minimal precision loss, indicating that the module encodes structural variability beyond the pretrained backbone. Finally, Appendix A.6 (Fig. 8, 15, 16) reveals consistent layerwise mixing patterns: early blocks rely more on pretrained structure, while later blocks emphasize dynamic updates, matching our architectural intent. These results suggest that the temporal module learns coherent, interpretable corrections.
> >
> > > **W5. Fine-tuning with small MD datasets could overfit, and the paper doesn’t really study that.**
> >
> > We agree with this concern and our design mitigates overfitting in low–data regimes: only the temporal module is trained on MD trajectories while the pretrained conformer backbone is frozen, substantially limiting the number of trainable parameters. Our distributional and dynamic evaluations on diverse, unseen molecules further indicate that the model does not memorize trajectories, as the learned dynamics do not collapse. Inference-time perturbations of the learned mixing parameter α (App. A.6.2, Table 8) also produce coherent changes in conformer diversity, suggesting that the module captures new structural variability rather than overfitting. Additionally, the extended QM9 evaluation (App. A.5.2, Table 6) shows consistent improvements over baselines on 959 unseen molecules, further supporting robust generalization.

---

> > > ### Author Response · Authors · 2025-11-26
> > >
> > > > **W6. Fundamentally, the architecture setup is still sequential when generating whereas BioEmu for example targets the equilibrium distribution directly.**
> > >
> > > We view our approach complementary to BioEmu and equilibrium-ensemble generators. While these target the stationary distribution directly, our formulation uniquely enables time-resolved trajectory generation, capturing dynamical quantities like autocorrelations and transition paths. Moreover, our framework likely benefits from advances in ensemble generators: improved structure priors can be incorporated into our pretrained backbone to further enhance temporal modeling under limited physio-realistic MD data. We will clarify this complementary relationship in the revision.
> > >
> > > > **Q1. How stable are the generated trajectories over long time horizons? Do they drift away from realistic energy basins?**
> > >
> > > We thank the reviewer for this question. Our scope in this work is to generate time-resolved trajectories whose statistical properties match those of reference MD over practically relevant rollout horizons. Within the trajectory lengths evaluated in Sections 5.2–5.3, we do not observe systematic drift into unrealistic regions. The generated samples maintain ground-truth bond-length, bond-angle, and torsion distributions (Table 1; Fig. 4A–B; App. Fig. 10, 12), and reproduce key dynamical observables such as torsional autocorrelations, TICA components, and MSM state populations (Fig. 4C–E; Fig. 5; App. Fig. 6, 11).
> > >
> > > Appendix A.4 provides energy-based analyses using TorchANI2x. Our method achieves substantially lower W1 distances to ground-truth energy profiles than GeoTDM on both QM9 and DRUGS (Table 5), and block-wise forward-simulation results show minimal deterioration across rollout blocks. Molecule-level comparisons (App. Table 12) similarly demonstrate close alignment with MD reference energies. This is continued in our analysis and response to W3 on our long simualtion experiments.
> > >
> > > > **Q2. Could the temporal interpolator be trained or conditioned on energy or force information to improve physical consistency?**
> > >
> > > We thank the reviewer for this interesting suggestion. In principle, the temporal interpolator could indeed be trained on or conditioned with energy or force information, and incorporating such signals may further improve physical consistency. In this work, we focus on learning dynamics purely from trajectory data, but the framework is fully compatible with hybrid objectives that combine trajectory supervision with energy- or force-based terms—for example, weighting frames by unnormalized energies, aligning denoising predictions with force vector fields or NNP energy predictors, or applying energy-based guidance at inference time.
> > >
> > > > **Q3. Could the method handle non-equilibrium or biased simulations?**
> > >
> > > We thank the reviewer for this question. Our formulation does not assume stationarity or detailed balance; the temporal interpolator learns whatever transition statistics are present in the training data. Thus, the method can in principle model non-equilibrium or biased simulations provided such trajectories are available. In this work, we focus on equilibrium MD for controlled comparison, but extending the framework to non-equilibrium processes—or incorporating biased initial forward simulation states—is a natural direction for future work.
> > >
> > > > **Q4. What’s the computational cost compared to training a full end-to-end trajectory diffusion model?**
> > >
> > > A full end-to-end EGInterpolator requires roughly 48 hours on 8 RTX A400 GPU-hours to train in our setting, but still performs worse than the pretrained-structure variant.
> > >
> > > > **Q5. Have you run experiments that in the limit sample from all meta-states from a single starting red point in appendix E6 and E7?**
> > >
> > > We thank the reviewer for this thoughtful suggestion. Our analysis currently measures a related notion of metastate coverage in Figure 4, where we compare the MSM state-occupancy statistics of our generated trajectories against both the reference MD trajectories and an MD oracle. We find that our model achieves higher $R^2$ correspondence to the reference distribution, indicating that it reproduces the global metastate landscape more faithfully than baselines. In a similar vein, in our tetrapeptide experiments we can see good coverage and visitation to FES states defined by the leading TICA coeffiecients, as well as lower JSD to the reference in the MSM state probability distributions as compared to GeoTDM. We agree that analyzing long-horizon simulations to assess mixing across all metastates is an important direction and is an experiment we plan to look into.
> > >
> > > [1] Klein et al. Timewarp: Transferable acceleration of molecular dynamics by learning time-coarsened dynamics. arXiv:2302.01170 (2023).
> > > [2] Vander Meersche et al., ATLAS: Protein flexibility from MD simulations, NAR (2024).
> > > [3] Wohlwend et al., Boltz-1: Democratizing biomolecular interaction modeling, bioRxiv (2024).

---

> > > > ### Author Response · Authors · 2025-12-03
> > > >
> > > > We once again appreciate the reviewer for the comments! We provide additional results referenced in our prior responses below:
> > > >
> > > > We include  ablation in Section 5.6, comparing \textsc{EGInterpolator} to a variant we call \textsc{EGInterpolator-Stack}. In this baseline, we remove both layer cascading and spatial–temporal interpolation, instead appending an equivalent number of temporally focused residual layers as a finetuned head atop the pretrained (and frozen) spatial model. As shown in Table 2, our cascaded, interpolated design delivers substantially better performance. These results empirically justify the added architectural complexity and support our theoretical motivation for interpolation.
> > > >
> > > > | Method                | Bond Angle M | Bond Angle Md | Bond Length M | Bond Length Md | Torsion M | Torsion Md | TICA₀,₁ M | TICA₀,₁ Md |
> > > > |-|-|-|-|-|-|-|-|-|
> > > > | EGInterpolator-S      | 0.325 | 0.330 | 0.330 | 0.321 | 0.414 | 0.419 | 0.673 | 0.672 |
> > > > | EGInterpolator-N      | 0.332 | 0.332 | 0.386 | 0.383 | 0.455 | 0.466 | 0.698 | 0.703 |
> > > > | **EGInterpolator**    | **0.173** | **0.153** | **0.142** | **0.112** | **0.377** | **0.388** | **0.650** | **0.644** |

---

### Official Review · Reviewer_dsY1 · 2025-10-31

**Soundness:** 3
**Presentation:** 3
**Contribution:** 3
**Rating:** 8
**Confidence:** 4

**Summary:**

Continuing on previous work in the field authors propose a diffusion model for molecular dynamics (MD) trajectories of small molecules. Specifically, they propose a decomposed approach where they divide the task into modeling valid 3D structures (per frame conformers) as the first pre-training task based on larger conformer datasets, ensuring structural validity and generalizability, and then learning a temporal interpolator to realize valid MD trajectories. Introducing a structure pre-training task for a dynamics model seems well-motivated.

**Strengths:**

The paper is technically well-written and derivations seem correct. Broadly speaking the proposed decomposition is also well-motivated. I particularly like introducing a marginal distribution over 3D structures (conformers) as a pre-training task so as to ensure structural validity across trajectories and transfer across molecules. The approach is also in principle extendable to larger molecules such as (small) proteins.

The proposed temporal interpolator seems to capture some measures of temporal dynamics (autocorrelation) better than alternatives such as GeoTDM as shown in Figure 4 E/F.

**Weaknesses:**

The per-frame marginal distribution over conformers could be perhaps more cleanly separated as a learning task. As a pre-training task, the model could be estimated from larger conformer datasets and then separately fine-tuned based on subsampled frames from MD trajectories prior to learning any temporal dynamics. It seems unnecessary to further adjust this part. Indeed, a product distribution over the frames would serve as a proper tilting function for learning a light temporal interpolator (tilted analogously to reward guided sampling). It is unclear to me why authors adopted a more convoluted convex combination that is learned from trajectories (with some frozen layers).

While theorem 4.1 appears correct, it also highlights potential issues with the approach. The resulting intermediate target distribution seems undesirable with the 1/(1-alpha) exponent.

The primary comparison results in Table 1 pertain to per-frame metrics except TICA. For this reason, authors' own pre-trained per-frame model (or any other per-frame model) would do well for most of these metrics, requiring no temporal interpolator. It would be helpful to focus primarily on evaluation of dynamics since this extension is the key contribution in the paper, not conformer generation.

**Questions:**

Could you elaborate on the justification for the convex combination in comparison to a simpler approach that uses per-frame marginals (pre-trained with all the data, including sampled MD frames) to adjust interpolator scores? The architecture that the authors use for the interpolator, equivariant temporal attention network, already includes analogous alternating per-frame and temporal updates. The temporal interpolator could still take ${\hat\epsilon}^{md}$ estimates as input, ensuring that its role would remain similarly light, offering (only) temporal corrections, aligning well with the motivation start at lines 231. I understand that the authors do freeze pre-trained ES layers in their approach.

---

> ### Author Response · Authors · 2025-11-26
>
> We thank the reviewer for their detailed review, questions, and suggestions! We provide point-to-point response to the comments as follows.
>
> > **W1. The per-frame marginal distribution over conformers could be perhaps more cleanly separated as a learning task... . It is unclear to me why authors adopted a more convoluted convex combination that is learned from trajectories (with some frozen layers).**
>
> We thank the reviewer for their comment and apologize if we misunderstood their question.
>
> First, we clarify that in our approach the pretrained components—trained on conformer datasets—are frozen during MD finetuning. The MD-specific training therefore focuses on learning the temporal alignment in latent space, effectively disentangling the conformer and trajectory distributions. Our architectural design is motivated in part by the comparisons in Table 1 & 2: the naïvely trained EGInterpolator-N variant (with no pretrained spatial layers) already outperforms GeoTDM, despite GeoTDM using a very similar architecture but without our interpolation-based information-passing layers. With the addition of pretrained spatial layers, we observe the further gains reported in Table 1 & 2.
>
> Related to the reviewer’s question on subsampled trajectory frames, we highlight our tetrapeptide experiments in Section 5.7 using the Timewarp datasets [1]. Because peptide conformer datasets are sparse, we create our own peptide conformer dataset from the frames of the training trajectories, and this process is detailed in Appendix B.1.1. Evaluated on 5 generated trajectories of 10 ns each (at a 10 ps frame rate) against 50 ns reference MD, pretrained EGInterpolator scheme consistently performs best across distributional JSD metrics, TICA/MSM dynamical metrics, and energy-based analyses (Figure 6, Appendix A.6).
>
> We agree that more targeted ablations of architectural components would be useful, and we plan to run and report these results shortly.
>
> > **W2. While theorem 4.1 appears correct, it also highlights potential issues with the approach. The resulting intermediate target distribution seems undesirable with the 1/(1-alpha) exponent.**
>
> We thank the reviewer for raising this interesting point! In fact, Theorem 4.1 conveys the following information: When we fit the score $\mathbf{\epsilon}^\text{tp}\_{\theta,\phi}$ toward the MD data distribution $p^\text{md}(\mathbf{x}^{[T]})$, we are essentially optimizing the additionally introduced parameters $\phi$ such that their induced score function $\mathbf{\epsilon}\_\phi$ models the interpolant $p^\text{md}(\mathbf{x}^{[T]})^\beta \hat{p}^\text{md}(\mathbf{x}^{[T]})^{1-\beta}$. Therefore, we want to highlight that the target distribution is still the MD data distribution $p^\text{md}(\mathbf{x}^{[T]})$. Theorem 4.1 instead assures that by using our linear interpolation design, the new parameters $\phi$ only need to fit the interpolant as opposed to the highly complex MD distribution $p^\text{md}(\mathbf{x}^{[T]})$, while the final output score is still guaranteed to fit the MD distribution, without the $1/(1-\alpha)$ exponent.
>
> > **W3. It would be helpful to focus primarily on evaluation of dynamics since this extension is the key contribution in the paper, not conformer generation.**
>
> We thank the reviewer for this helpful suggestion! We agree that per-frame metrics cannot assess temporal fidelity and include them to establish structural and distributional validity before evaluating dynamics. Our evaluation focuses on temporal behavior through TICA components, torsional autocorrelation functions (Fig. 4E–G; App. Fig. 11), MSM state-population $R^2$ (Fig. 4C–D), and an ablation of the interpolator’s contribution to dynamic modeling (Appendix A.5.4). These metrics directly capture the effect of the temporal interpolator and show clear improvements over both the pretrained per-frame model and GeoTDM. In the revision, we highlight these temporal evaluations more prominently and clarify that per-frame results serve as prerequisites for meaningful dynamic comparisons.

---

> > ### Author Response · Authors · 2025-11-26
> >
> > > **Q1. Could you elaborate on the justification for the convex combination in comparison to a simpler approach that uses per-frame marginals (pre-trained with all the data, including sampled MD frames) to adjust interpolator scores?**
> >
> > As noted in our response to W1, our architectural design was motivated in part by the comparisons in Tables 1 and 2: the naïvely trained EGInterpolator-N variant (with no pretrained spatial layers) already outperforms GeoTDM. Building on this, our full model freezes pretrained spatial layers to encourage the desired disentangling of conformer and trajectory distributions during MD finetuning. We further explored the benefits of frame-wise conformer pretraining in our tetrapeptide experiments, where such initialization improved downstream dynamics generation.
> >
> > A final capability enabled by our interpolation mechanism is enhanced inference-time control. As shown in Section A.6.2 and Table 8, adjusting the learned α parameter during inference increases conformational diversity when EGInterpolator is sampled as a single-frame conformer generator, approaching the performance of GeoDiff [2] on the QM9 conformer task. Nonetheless, we agree that more targeted ablations would be beneficial and plan to carry these out in the coming days.
> >
> > [1] Klein et al. Timewarp: Transferable acceleration of molecular dynamics by learning time-coarsened dynamics. arXiv:2302.01170 (2023).
> > [2] Xu et al., GeoDiff: a Geometric Diffusion Model for Molecular Conformation Generation, ICLR (2022).

---

> > > ### Author Response · Authors · 2025-12-03
> > >
> > > We once again appreciate the reviewer for the comments -- their comments and concerns have made our work better! We provide additional results referenced in our prior responses below:
> > >
> > > > Q1. Could you elaborate on the justification for the convex combination in comparison to a simpler approach that uses per-frame marginals (pre-trained with all the data, including sampled MD frames) to adjust interpolator scores?
> > >
> > > We address this concern with the ablation in Section 5.6, comparing EGInterpolator to a variant we call EGInterpolator-Stack. In this baseline, we remove both layer cascading and spatial–temporal interpolation, instead appending an equivalent number of temporally focused residual layers as a fine-tuned head atop the pre-trained (and frozen) spatial model. As shown in Table 2, our cascaded, interpolated design delivers substantially better performance. These results empirically justify the added architectural complexity and support our theoretical motivation for interpolation.
> > >
> > > | Method                | Bond Angle M | Bond Angle Md | Bond Length M | Bond Length Md | Torsion M | Torsion Md | TICA₀,₁ M | TICA₀,₁ Md |
> > > |-|-|-|-|-|-|-|-|-|
> > > | EGInterpolator-S      | 0.325 | 0.330 | 0.330 | 0.321 | 0.414 | 0.419 | 0.673 | 0.672 |
> > > | EGInterpolator-N      | 0.332 | 0.332 | 0.386 | 0.383 | 0.455 | 0.466 | 0.698 | 0.703 |
> > > | **EGInterpolator**    | **0.173** | **0.153** | **0.142** | **0.112** | **0.377** | **0.388** | **0.650** | **0.644** |

---

### Official Review · Reviewer_eveP · 2025-11-02

**Soundness:** 2
**Presentation:** 2
**Contribution:** 1
**Rating:** 2
**Confidence:** 4

**Summary:**

The paper proposes to learn generative models of MD trajectories by fine-tuning conformer generative models. This is done by adding additional temporal layers to the conformer generation model and fine tuning on dynamics trajectories. The model is evaluated on simulations of small druglike molecules from QM9 and GEOM-DRUGS, where it roughly reproduces static and dynamic observables.

**Strengths:**

The approach is very sensible - developing full-trajectory models by fine-tuning static structure models is an approach that many later works will likely follow.

**Weaknesses:**

**Novelty**
* The authors' contribution amounts to the incorporation of conformer pretraining for MD trajectory generation, which in my opinion is not significant or non-obvious enough for a conference paper in the absence of compelling results.

**Significance**
* The task of MD trajectory generation for small molecules is of unclear utility. I am sympathetic that the authors are following prior precedent, where small molecule conformations have historically served as testbeds for modeling larger systems. However, as the AI for science field matures, it is important that the community stays focused on forward-looking applications.

**Experiments**
* The authors write "In contrast, our method generalizes more readily across arbitrary molecular systems," yet do not show experiments on peptides or proteins, which would allow proper comparisons with previous work.
* The result that the model outperforms AR baselines is not surprising, given prior work (MDGen).
* From the results shown in Figure 4, it appears that the model has a lot of trouble matching ground truth distributions of bond lengths, torsion angles, and autocorrelation decays.

**Method**
* The linear interpolation of the temporal module output seems rather contrived, especially the thereotical justification. There are no ablation studies showing why this additional complexity is necessary.

**Questions:**

No specific questions

---

> ### Author Response · Authors · 2025-11-26
>
> We appreciate the reviewer for the comments! We provide a point-to-point response as follows.
>
> > **W1. Novelty -- The authors' contribution amounts to the incorporation of conformer pretraining for MD trajectory generation, which in my opinion is not significant or non-obvious...**
>
> We thank the reviewer for raising this point and apologize for any lack of clarity. The core idea of the paper is a principled decomposition of MD generative modeling into (i) structural generation and (ii) temporal consistency enforcement, implemented through a pretrained diffusion backbone and a dedicated interpolator. Learning coherent MD trajectories introduces fundamentally different inductive biases than static conformer generation, and our method introduces new architectural components and a temporal-training objective specifically designed to enforce coherence across time. To our knowledge, no prior trajectory model leverages large-scale conformer pretraining to address MD data scarcity. In contrast, our method introduces:
> * a pretrained structure diffusion model that provides a transferable geometric prior,
> * a temporal interpolator operating in the pretrained latent space to enforce smoothness,
> * and a temporal loss that aligns independently generated structures into a coherent trajectory.
>
> Finally, the interpolation module is essential rather than cosmetic. Ablations in Appendix A.5.4 (Table 7) show that removal of the contribution of the interpolation module leads to degenerate torsion de-correlations and significantly worse dynamic coherence.
>
> > **W2. Significance -- The task of MD trajectory generation for small molecules is of unclear utility...**
>
> We appreciate the reviewer’s concern and agree that scaling to biomolecular systems is an important long-term goal. We began our investigation in the small-molecule MD domain because it remains the standard, well-controlled setting for developing and benchmarking generative dynamics models. Systems like QM9 and DRUGS de-correlate an order of magnitude faster (median 4.8 ps / 2 ps), demanding accurate handling of rapid, less-constrained motions. Chemically meaningful evaluation further requires precise modeling of bond lengths, angles, and torsions—capabilities our method explicitly targets. Moreover, QM9 and DRUGS span far broader atom types, functional groups, and topologies than short peptides, making them a good test of both accuracy and generalization.
>
> Nonetheless, in light of showcasing our method across multiple and different systems, we are excited to report the additional results, including tetrapeptides and proteins. Please refer to our response in W3 where we discuss our additional experiments.

---

> > ### Author Response · Authors · 2025-11-26
> >
> > > **W3. Experiments -- Experiments on peptides or proteins, which would allow proper comparisons with previous work.**
> >
> > Below we provide additional results illustrating the flexibility of our framework across diverse molecular systems. The experiments in Sections 5.3–5.5 focused on small molecules and demonstrated strong generalization to unseen organic compounds with varied functional groups and topologies.
> >
> > To further assess generality, we evaluate a single unified model trained jointly on two tasks (unconditional generation and forward simulation) and across two datasets (QM9 and DRUGS). This model performs substantially better than all baselines and closely matches the performance of specialized models; notably, its accuracy on QM9 slightly exceeds that of the QM9-only model. We hypothesize that the DRUGS dataset, with its greater chemical complexity and diversity, provides richer training signal that enhances generalization to the simpler QM9 regime. Overall, these findings show that a cross-task, cross-dataset model can perform competitively, underscoring the versatility of our approach. These results can be found in Section A.5 (Table 5) as well as below:
> >
> > |Model|Bond Angle|Bond Length|Torsion|TICA_0|TICA_01
> > |-|-|-|-|-|-|
> > |EGInterpolator-Both QM9|**0.231** **0.219**|**0.168** **0.158**|**0.348** **0.367**|**0.393** **0.390**|**0.623** **0.631**|
> > |EGInterpolator QM9|0.305 0.292|0.210 0.188|0.363 0.380|0.417 0.406|0.636 0.642|
> > |GeoTDM QM9|0.691 0.690|0.676 0.670|0.489 0.527|0.449 0.453|0.691 0.694|
> > |EGInterpolator-Both Drugs|0.212 0.197|0.216 0.195|0.417 0.434|0.488 0.506|0.681 0.679|
> > |EGInterpolator Drugs|**0.173** **0.153**|**0.142** **0.112**|**0.377** **0.388**|**0.454** **0.441**|**0.650** **0.644**|
> > |GeoTDM Drugs|0.640 0.645|0.643 0.645|0.498 0.503|0.531 0.550|0.712 0.720|
> >
> > We also evaluate our framework on tetrapeptide systems using the Timewarp datasets [1] (Section 5.7). As conformer datasets for peptides are limited, we create our own peptide conformer dataset from the frames of the training trajectories, and this process is detailed in Appendix B.1.1. We generate 5 samples of 10 ns of dynamics at a 10 ps frame rate per peptide to compare to 50 ns of reference trajectories. We find that our method yields the strongest performance across distributional JSD metrics (Figure 6C), dynamical TICA/MSM metrics (Figure 6c), and energy-based analyses (Appendix A.6, Table 7) (Also reported below). These metrics are also supported by the results shown in Figure A & B, which highlight the improved FES and torsion decorrelations respectively by EGInterpolator on selected examples.
> >
> > |Metric|EGInterpolator|GeoTDM|
> > |-|-|-|
> > |Bond Angle (↓)|**0.102**|0.607|
> > |Bond Length (↓)|**0.074**|0.643|
> > |Torsion BB (↓)|**0.305**|0.520|
> > |Torsion SC (↓)|**0.223**|0.434|
> > |Torsion All (↓)|**0.256**|0.469|
> > |TICA₀ (↓)|**0.282**|0.425|
> > |TICA₀₁ (↓)|**0.462**|0.634|
> > |MSM (↓)|**0.262**|0.344|
> >
> > Finally, to explore the scalability of our method, we are extending our framework to larger biomolecular systems using Boltz1 [2] on the ATLAS dataset [3], training models to forward-simulate 25 ns at a 100 ps frame rate. In Section 5.8 & Figure 6D, we present initial trajectory visualizations and validation metrics relative to the naïve Boltz1 baseline, showing improvements in torsion-angle JSDs and other early indicators. We are actively pursuing more comprehensive evaluation and comparisons to ensemble and dynamics models on these larger systems.
> >
> > We hope these experiments clearly demonstrate the breadth and scalability of our approach, as well as its promise across molecular regimes ranging from small molecules to peptides and larger biomolecular systems.
> >
> > > **W4. Experiments -- The result that the model outperforms AR baselines is not surprising, given prior work (MDGen).**
> >
> > We include autoregressive (AR) baselines to establish a broad set of all-atom comparators. Our model outperforms these AR baselines on both unconditional generation and forward simulation tasks on DRUGS. Beyond these direct comparisons, our analysis also examines capabilities that naïve AR models do not capture, like interpolation / transition-path sampling, where our method shows compelling performance compared with MD oracles (Section 5.5, Fig. 5; Section A.2, Fig. 6). We also provide additional results unique to our approach, such as improved conformer-generation ability from the trajectory-trained model when sampled in single-frame mode (Section A.6.2, Table 8).

---

> > > ### Author Response · Authors · 2025-11-26
> > >
> > > > **W5. Experiments -- From the results shown in Figure 4, it appears that the model has a lot of trouble matching ground truth distributions of bond lengths, torsion angles, and autocorrelation decays.**
> > >
> > > We apologize for the ambiguity in the original plots and have added a legend to Figure 4 for clarity. The displayed colors correspond to ground truth (MD) in red, GeoTDM in blue, and our model in green.
> > >
> > > In the bond-length histogram (Fig. 4A), our model’s distribution aligns closely with the ground truth, causing the two curves to visually overlap and appear as a single mode. The wider background distribution corresponds to GeoTDM’s higher variance. In the torsion-angle histogram (Fig. 4B), our method accurately captures the trimodal ground-truth pattern, whereas GeoTDM produces a degenerate distribution. Additional examples demonstrating this behavior appear in Figures 10 and 12 of the Appendix.
> > >
> > > Similarly, Figures 4E–G show the torsional autocorrelation decay for the ground truth, GeoTDM, and our method, with further examples in Figure 11. Across these cases, our model reproduces the characteristic decorrelation times observed in reference MD simulations, in contrast to the overly smooth or degenerate dynamics generated by GeoTDM. (Appendix values as well)
> > >
> > > Finally, we also observe that our trajectories yield MSM state populations with comparable or higher R² relative to the MD oracle, further supporting the fidelity of the generated dynamics for that example.
> > >
> > > > **W6. Method -- The linear interpolation of the temporal module output seems rather contrived, especially the thereotical justification. There are no ablation studies showing why this additional complexity is necessary.**
> > >
> > > We appreciate the reviewer’s comment. As an initial point of comparison, Section 5.6 (Table 2) shows that a naïvely trained version of our architecture—without any pretrained or frozen components—already outperforms GeoTDM and other baselines. Since GeoTDM is architecturally similar but lacks our interpolation layers, this suggests that the architectural design itself contributes meaningfully to performance. Adding pretrained spatial layers further amplifies these gains, yielding the improvements reported in Table 2 relative to the naïve variant.
> > >
> > > Nonetheless, we agree that additional head-to-head ablations would be valuable, and we plan to conduct and report these analyses promptly.
> > >
> > > [1] Klein et al. Timewarp: Transferable acceleration of molecular dynamics by learning time-coarsened dynamics. arXiv:2302.01170 (2023).
> > > [2] Vander Meersche et al., ATLAS: Protein flexibility from MD simulations, NAR (2024).
> > > [3] Wohlwend et al., Boltz-1: Democratizing biomolecular interaction modeling, bioRxiv (2024).

---

> ### Author Response · Authors · 2025-12-03
>
> We once again appreciate the reviewer for the comments! We provide additional results referenced in our prior responses below:
>
> > W6. Method -- The linear interpolation of the temporal module output seems rather contrived, especially the thereotical justification. There are no ablation studies showing why this additional complexity is necessary.
>
> We address this concern with the ablation in Section 5.6, comparing EGInterpolator to a variant we call \EGInterpolator-Stack. In this baseline, we remove both layer cascading and spatial–temporal interpolation, instead appending an equivalent number of temporally focused residual layers as a fine-tuned head atop the pre-trained (and frozen) spatial model. As shown in Table 2, our cascaded, interpolated design delivers substantially better performance. These results empirically justify the added architectural complexity and support our theoretical motivation for interpolation.
>
> | Method                | Bond Angle M | Bond Angle Md | Bond Length M | Bond Length Md | Torsion M | Torsion Md | TICA₀,₁ M | TICA₀,₁ Md |
> |-|-|-|-|-|-|-|-|-|
> | EGInterpolator-S      | 0.325 | 0.330 | 0.330 | 0.321 | 0.414 | 0.419 | 0.673 | 0.672 |
> | EGInterpolator-N      | 0.332 | 0.332 | 0.386 | 0.383 | 0.455 | 0.466 | 0.698 | 0.703 |
> | **EGInterpolator**    | **0.173** | **0.153** | **0.142** | **0.112** | **0.377** | **0.388** | **0.650** | **0.644** |

---

### Meta-Review · Area_Chair_8pVH · 2026-01-03

**Summary:**

This work proposes a framework to first pre-training the structure of molecules and then apply fine-tuning for MD trajectory generation. The authors train a diffusion-based structure generation model and introduce an interpolator to enforce consistency among generated structures. Through downstream finetuning on two tasks and two datasets, authors show the value of the structure pre-training method.

The reviewers raise lots of concerns and questions in the initial reviews, such as the design logic that the authors adopt a more convoluted convex combination that is learned from trajectories, the theoretical guidance, the significance of the experiments and also the performances comparison. Authors provided detailed responses towards the review comments, while some of the concerns and questions are resolved from my view, some remain uncleared.
After carefully reading the review and rebuttal, I suggest the authors to revise the rebuttal concerns by improving the paper, especially towards the large molecule systems and to better show the significance of the work from an empirically way.

**Reviewer Concerns:**

The reviewers raise lots of concerns and questions in the initial reviews, such as the design logic that the authors adopt a more convoluted convex combination that is learned from trajectories, the theoretical guidance, the significance of the experiments and also the performances comparison. Authors provided detailed responses towards the review comments, while some of the concerns and questions are resolved from my view, some remain uncleared.

Reviewer eveP raises multiple concerns about the design/experiments, and the novelty is also a big issue. Reviewer yR8j and YkBB also raise concerns about the effectiveness and the advantages of the method. Those questions are remained during rebuttal.

**Reviewer Scores:**

From the current review and the rebuttal, Reviewer eveP seems to be hard to change the score, or possibly change but still towards a negative view. Reviewer yR8j and Reviewer yR8j are also hard to change the score.

---

### Decision · Program_Chairs · 2026-01-26

Accept (Poster)